# Copper chelation redirects neutrophil function to enhance anti-GD2 antibody therapy in neuroblastoma

Jourdin R. C. Rouaen[1,2], Antonietta Salerno [1,2], Tyler Shai-Hee[1,2], Jayne E. Murray[1,2], Giulia Castrogiovanni[1], Charlotte McHenry[1,2], Toni Rose Jue [2], Vu Pham[1], Jessica Lilian Bell[1,2], Ensieh Poursani[1,2], Emanuele Valli [3], Riccardo Cazzoli [2], Naomi Damstra [4,5], Delia J. Nelson[5,6], Kofi L. P. Stevens[4,5], Jonathan Chee [4], Iveta Slapetova[7], Maria Kasherman[7], Renee Whan[7], Francis Lin[8], Blake J. Cochran[1], Nicodemus Tedla[1], Feyza Colakoglu Veli[9], Aysen Yuksel [10], Chelsea Mayoh [1,2], Federica Saletta[2], Daniele Mercatelli [11], Tatyana Chtanova[8,12], Arutha Kulasinghe [13], Daniel Catchpoole[10], Giuseppe Cirillo [14], Maté Biro[9], Holger N. Lode[15], Fabio Luciani [1,16], Michelle Haber [1,2], Juliet C. Gray[17], Toby N. Trahair [1,2,18] & Orazio Vittorio [1,2] ✉

Anti-disialoganglioside (GD2) antibody therapy has provided clinical benefit to patients with neuroblastoma however efficacy is likely impaired by the immunosuppressive tumor microenvironment. We have previously defined a link between intratumoral copper levels and immune evasion. Here, we report that adjuvant copper chelation potentiates anti-GD2 antibody therapy to confer durable tumor control in immunocompetent models of neuroblastoma. Mechanistic studies reveal copper chelation creates an immune-primed tumor microenvironment through enhanced infiltration and activity of Fc-receptor-bearing cells, specifically neutrophils which are emerging as key effectors of antibody therapy. Moreover, we report copper sequestration by neuroblastoma attenuates neutrophil function which can be successfully reversed using copper chelation to increase pro-inflammatory effector functions. Importantly, we repurpose the clinically approved copper chelating agent Cuprior as a non-toxic, efficacious immunomodulatory strategy. Collectively, our findings provide evidence for the clinical testing of Cuprior as an adjuvant to enhance the activity of anti-GD2 antibody therapy and improve outcomes for patients with neuroblastoma.

Neuroblastoma is a malignancy of the sympathetic nervous system and accounts for 15% of all childhood cancer-related deaths[1]. Despite intensive multi-modal therapy, the prognosis for high-risk patients remains poor. While the integration of anti-disialoganglioside (GD2) antibody (e.g., dinutuximab) into maintenance therapy has been demonstrated to substantially improve survival and is now a standard of care, nearly half of patients remain refractory or develop treatment resistance[2]. Moreover, most survivors exhibit an increased risk of late mortality, secondary malignancies, and other chronic health conditions[3,4]. Therefore, improving the long-term efficacy of

this promising immunotherapy remains an important unsolved challenge.

GD2 is a tumor antigen ubiquitously expressed by the majority of neuroblastomas with anti-GD2 therapies known to promote antibody-dependent cell-mediated cytotoxicity (ADCC) by Fc-expressing immune effectors[5]. Natural killer (NK) cells and macrophages are cited as prominent mediators of ADCC; however, neutrophils are emerging as key players in achieving effective responses in several cancer types, including neuroblastoma[6]. Neuroblastomas harbor an immunosuppressive tumor microenvironment and therefore combination strategies are now considered more likely to succeed in the restoration of antitumor immunity required for response to anti-GD2 therapy[7,8].

Elevated intracellular copper levels have been reported in several malignancies, suggesting copper homeostasis as a tumor dependency that may be exploited as a therapeutic target[9,10]. Concerning neuroblastoma, we have previously reported elevated levels of the high-affinity copper transporter 1 (CTR1) in neuroblastoma clinical samples and preclinical models, and recently demonstrated that intratumoral copper levels regulate the expression of the immune checkpoint molecule Programmed Death-Ligand 1 (PD-L1)[11,12]. Copper depletion using chelating agents including tetraethylenepentamine (TEPA) reduced tumoral PD-L1 expression, increased CD8[+] cytotoxic T and NK cell infiltration, and enhanced survival. These observations prompted us to evaluate copper chelation as a potential treatment strategy through widespread remodeling of the neuroblastoma tumor microenvironment.

In the current study, we report that copper chelation is an effective adjuvant strategy to potentiate anti-GD2 therapy in two preclinical immunocompetent models (*Th-MYCN*; NXS2) of neuroblastoma. We demonstrate that copper chelation favorably alters the infiltration and antitumor activity of both lymphoid and myeloid compartments, specifically neutrophils. We characterize a mechanism of tumor immune evasion in which copper sequestration drives neutrophil dysfunction. Importantly, we establish TETA (triethylenetetramine, marketed as Cuprior), an FDA-approved copper chelating agent for Wilson's disease, as an innovative immunomodulatory agent for repurposing as an adjuvant to anti-GD2 therapy. Together, our findings provide crucial evidence to support the clinical testing of this immune-based combination therapy for patients with neuroblastoma.

## Results

### Copper chelation potentiates anti-GD2 antibody therapy

To stimulate immune activation in *Th-MYCN* model, we primed the tumor microenvironment by administering the copper chelating agent TEPA daily for 1 week, followed by the addition of twice weekly doses of anti-GD2 antibody (Fig. 1a). After four cycles, TEPA and anti-GD2 therapy were both reduced to twice weekly administrations until the ethical endpoint (tumor diameter ≥10 mm) or a maximum treatment duration of 180 days was met. Importantly, this immunocombination strategy was well-tolerated with no adverse effects observed.

As per previous reports, anti-GD2 monotherapy induced modest antitumor activity, but this was substantially enhanced with the addition of TEPA, resulting in significantly extended survival ($p = 0.043$) and durable responses in approximately 30% of animals (Fig. 1b,c)[13,14]. To examine changes in the immune environment, we performed OPAL multiplex immunohistochemistry on tumors obtained after 14 days of treatment (Fig. 1d).

As a single agent, TEPA was observed to significantly increase NCR1[+] NK ($p = 0.046$), CD8[+] cytotoxic T ($p = 0.014$) and CD11b[+] myeloid ($p = 0.003$) cell compartments compared to the control isotype arm (Fig. 1e). Anti-GD2 therapy alone also significantly increased cytotoxic T ($p = 0.023$) and myeloid cell ($p = 0.0073$) infiltration with a trend towards increased NK infiltration compared to the control isotype arm. Nonetheless, the addition of copper chelation to anti-GD2 therapy

substantially enhanced the infiltration of NK ($p = 0.006$) and myeloid ($p = 0.0012$) immune subsets but did not synergize to cause further infiltration of cytotoxic T cells. NK cells are recognized as the primary effectors of ADCC in neuroblastoma, eliciting responses through Fc-receptor binding[15]. Unexpectedly, we observed that copper chelation therapy also increased the frequency of infiltrating myeloid cells, which have been associated with immunosuppressive activity in neuroblastoma but can also be engaged as potent effectors of ADCC[16].

Our immune-induction strategy demonstrates that copper chelation is an effective adjuvant to increase immune infiltration and enhance tumor control in combination with anti-GD2 antibody therapy.

### Copper chelation modulates cytokine levels to drive immune cell infiltration

The immunosuppressive tumor microenvironment is supported by soluble cytokines and chemokines, that exert pleiotropic effects on immune cells by regulating migration, infiltration, and effector activity to facilitate tumor progression[17]. To understand how these processes are altered during induction with TEPA, we performed multiplex cytokine profiling to analyze serum and tumor milieus after 1 week of treatment (Fig. 2a). Treatment was observed to modulate cytokine levels associated with immune cell recruitment (RANTES/CCL5, GM-CSF, KC/CXCL1) and pro-inflammatory effector functions (IFN-γ, IP-10/CXCL10, TNF, IL-10, IL-2, IL-6) (Fig. 2b). Of note, copper chelation significantly reduced levels of the immunosuppressive cytokine transforming growth factor-beta (TGF-β) in both serum ($p = 0.007$) and the tumor microenvironment ($p = 0.018$), aligning with our recent finding linking its expression to copper levels in a variety of cancer types, including neuroblastoma[18]. Interestingly, statistically significant changes were restricted to the tumor microenvironment except for KC and IFN-γ, suggesting that copper chelation can induce a "finely tuned" local shift in immune response.

Given the concomitant shift in cytokine profile and increase in myeloid cell infiltration with copper chelation, we sought to elucidate the composition of this immune compartment via flow cytometry. Results revealed CD11b[+]Ly6G[+] neutrophils as the major population, exhibiting a nearly six-fold increase in tumoral infiltration, followed by macrophages, monocytes, and dendritic cells (DCs) (Fig. 2c,d). Given the systemic and local upregulation of KC (a known neutrophil chemoattractant) with copper chelation therapy, we postulated that treatment could stimulate neutrophil trafficking toward the tumor site[19].

To assess this, we immunophenotyped the peripheral blood of control and TEPA-treated *Th-MYCN* animals and confirmed significant increases in both the abundance ($p = 0.029$) and relative percentages ($p = 0.029$) of circulating neutrophils (Fig. 2e). No other immune subsets or erythrocyte measurements were significantly affected (Supp. Fig. 1), with these results underscoring an exclusive relationship between copper chelation and tumoral neutrophil recruitment. Of note, neutrophils have been demonstrated to mediate ADCC against neuroblastoma in the presence of anti-GD2 antibody, illuminating a possible link to the enhanced tumor clearance conferred by immunocombination therapy in the *Th-MYCN* model (Fig. 1b,c)[20].

These data suggest that copper chelation favorably modulates cytokine levels to support the observed changes in immune cell infiltration, particularly neutrophils, thereby promoting an immune-permissive neuroblastoma tumor microenvironment.

### Copper chelation destabilizes the neuroblastoma tumor microenvironment

To further characterize changes occurring during priming of the tumor microenvironment, we constructed a tissue microarray of *Th-MYCN* tumors consisting of untreated, control and TEPA-treated tumors resected after 3 and 7 days of treatment. Utilizing the

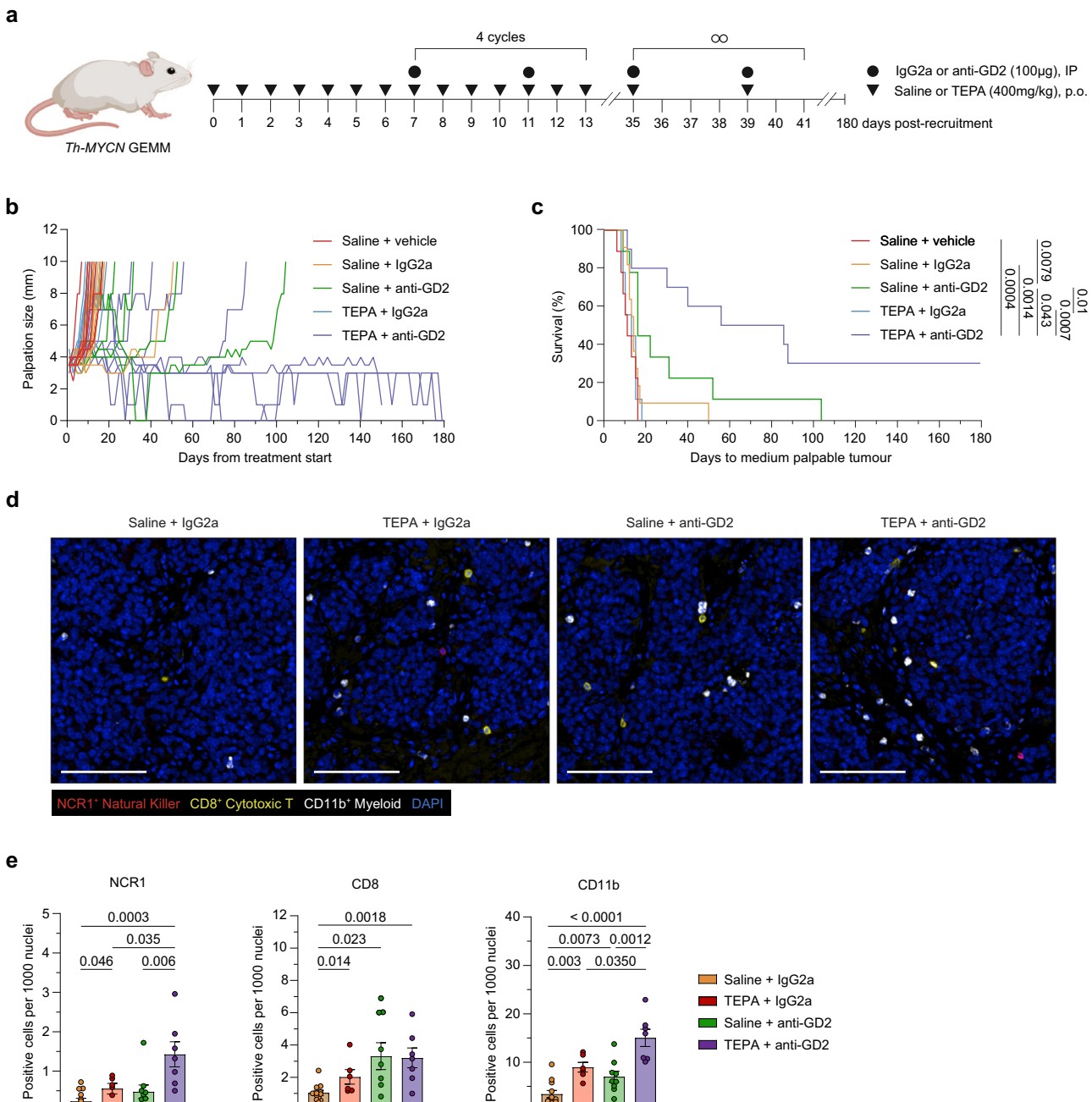

**Fig. 1 | Copper chelation potentiates antitumor activity of anti-GD2 immunotherapy. a** Experimental design and dosing strategy. Schematic created in BioRender. Vittorio, O. (2024). BioRender.com/c53h577. **b** Individual tumor kinetics of *Th-MYCN* mice. **c** Kaplan–Meier survival curves of *Th-MYCN* mice presented in (**b**). Statistical pairwise comparisons were calculated using a two-tailed Mantel–Cox log-rank test with *p* values displayed in the figure. For (**b**) and (**c**), data are *n* = 11 (Saline + IgG2a) or *n* = 10 (all other groups) biological replicates, one independent experiment. **d** Representative images of merged OPAL multiplex immuno-fluorescence spectra depicting the tumoral distribution of NCR1+ natural killer cells

(red), CD8+ cytotoxic T cells (yellow), CD11b+ myeloid (white) and DAPI nuclei stain (blue) in *Th-MYCN* neuroblastoma tumor tissue 14 days post-treatment. Scale bar, 100 μm. **e** Immune cell quantification of (**d**) as positive counts per 1000 nuclei. Significance was calculated using a two-tailed Mann–Whitney *U* test with *p* values displayed in figure. For (**e**), Data are mean ± SEM *n* = 4 (Saline + IgG2a) or *n* = 3 (all other groups) biological replicates with a minimum of two technical replicates, one independent experiment. Abbreviations: IP intraperitoneal, p.o. orally. Source data are provided as a Source Data file.

NanoString GeoMx Digital Spatial Profiling (DSP) platform, regions of interest (ROIs; *n* = 100 total) were selected using a ratio of pan-cytokeratin (PanCK) and pan-leukocyte marker CD45 to assign conditional assignment of low/high immune infiltration for each region (Supp. Fig. 12). We observed that copper chelation therapy with TEPA was strongly associated with immune infiltration, as determined by CD45-positive staining of tumor cores (Fig. 3a). When comparing low/high infiltration assignments, TEPA-treated tumors were observed to

exhibit a time-dependent increase in immune infiltration (Fig. 3b). Next, we performed a differential gene expression analysis comparing 3- and 7-day TEPA-treated low-infiltrated versus TEPA-treated high-infiltrated ROIs to highlight crosstalk between tumor and immune cell signals. Interestingly, we observed the upregulation of *Mgp* and *Mprip* together with *Znrf1* which is suggestive of neuronal differentiation, a key feature associated with a favorable clinical prognosis in neuro-blastoma (Fig. 3c)[21,22]. Enrichment analyses revealed significant

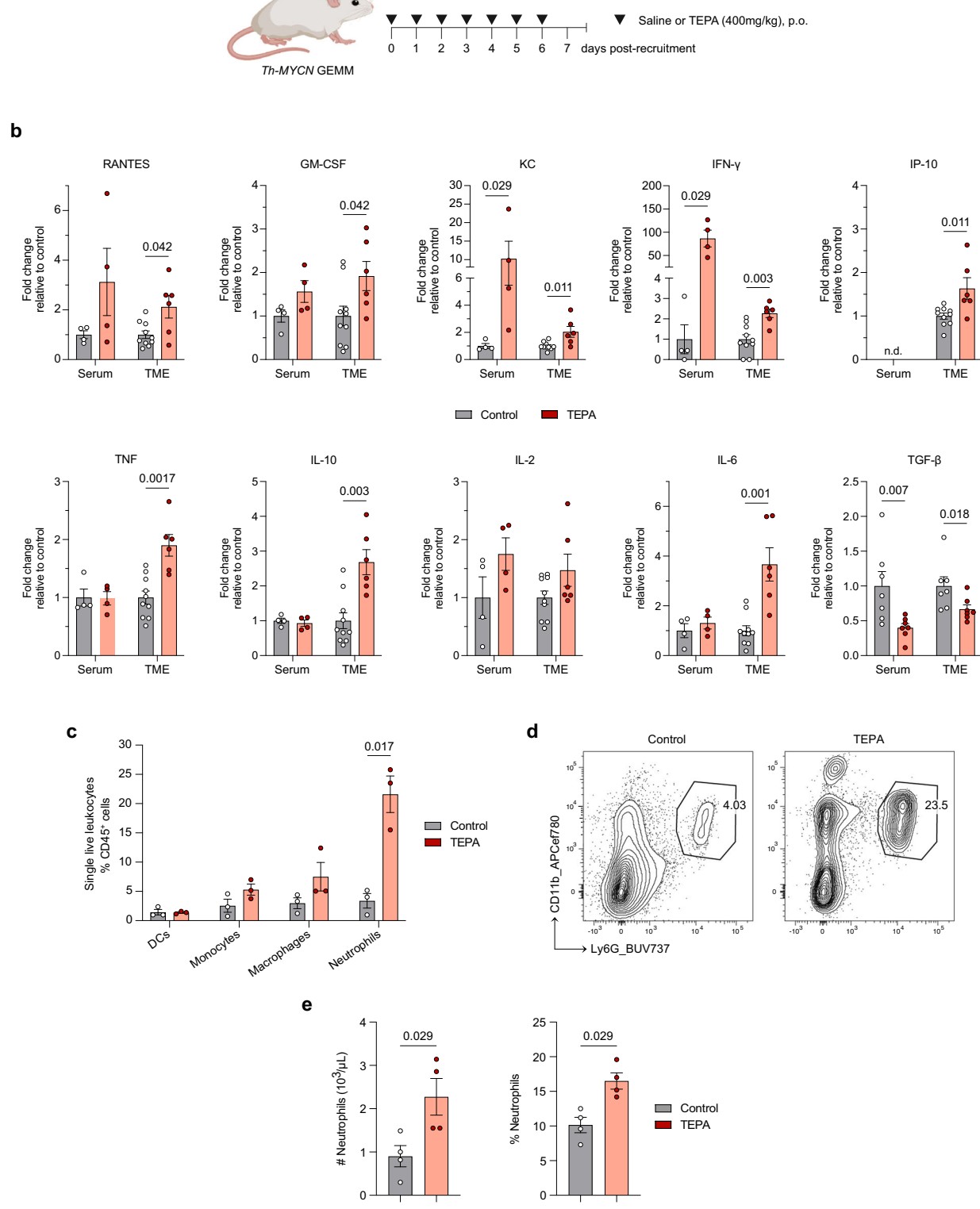

changes in numerous pathways with copper chelation including upregulation of pro-inflammatory pathways involved in interferon (IFN) and tumor necrosis factor (TNF) signaling, the p53 pathway involved in tumor suppression, and downregulation of MYC targets responsible for oncogenic signaling (Fig. 3d,e). We observed increased angiogenic signaling along with significant upregulation of the hemoglobin family genes *Hbb-b2*, *Hba-a1*, and *Hbb-b1* (Fig. 3c),

alluding to increased tumoral vascularity which may facilitate intra-tumoral immune infiltration (Fig. 3d). Interestingly, pathways were also enriched for hypoxia and neutrophil degranulation activity, demonstrating an interplay known to occur in inflamed environments and further emphasizing the role of this immune population in the tumor microenvironment (Fig. 3d,e)[23]. Notably, this enrichment analysis showed remarkable overlap with a recently published dataset by He

**Fig. 2 | Copper chelation promotes an immune-permissive tumor micro-environment. a** Experimental design and dosing strategy of *Th-MYCN* model with peripheral blood and tumors obtained after 1 week of treatment. Schematic created in BioRender. Vittorio, O. (2024). BioRender.com/c53h577. **b** Cytokine levels in sera and tumoral lysates obtained from control and TEPA-treated mice. Serum data (excluding TGF-β) are presented as mean ± SEM, *n* = 4 (both groups) biological replicates, one independent experiment. Tumor microenvironment (TME) data (excluding TGF-β) are presented as mean ± SEM, *n* = 10 (Control) and *n* = 6 (TEPA) biological replicates, one independent experiment. TGF-β serum and TME data are presented as mean ± SEM, *n* = 7 (both groups) biological replicates, two independent experiments. Significance was calculated using a two-tailed Mann–Whitney *U* test with *p*-values displayed in the figure. **c** Flow cytometric analysis of myeloid subset frequencies from tumors after 1 week of treatment. Data are presented as mean ± SEM, *n* = 3 (both groups) biological replicates, one independent experiment. Significance was calculated using a two-tailed *t*-test with Welch's correction with *p*-value displayed in the figure. **d** Representative flow cytometry plots of CD11b⁺Ly6G⁺ neutrophil frequency of tumors plotted in (**c**). **e** Count and percentage of circulating neutrophils obtained from control and TEPA-treated mice. Data are presented as mean ± SEM, *n* = 4 (both groups) biological replicates, one independent experiment. Significance was calculated using a two-tailed Mann–Whitney *U* test with *p*-values displayed in the figure. Abbreviations: n.d. no data available, p.o. orally, TME tumor microenvironment. Source data are provided as a Source Data file.

et al., which compared enriched pathways in low-risk versus high-risk neuroblastoma clinical cohorts[24].

Taken together, these results indicate that copper chelation can shift neuroblastoma tumors towards a less aggressive phenotype by destabilizing the tumor microenvironment.

## Copper chelation enhances neutrophil infiltration in the neuroblastoma tumor microenvironment

To characterize functional changes at a single-cell resolution, we analyzed tumors treated for 7 days with saline or TEPA using the BD Rhapsody system. Fresh tumor sections were dissociated using a protocol that ensured high viability of both tumor and immune compartments for subsequent enrichment by flow cytometric sorting (Fig. 4a). After quality control, dimensionality reduction, and clustering, we obtained 13,555 tumor cells (defined using *Mycn*⁺/*Ptprc* [CD45]⁻ gene expression; Fig. 4b). Importantly, tumor cells were highly responsive to copper chelation therapy with a significant decrease in gene expression of the metallothioneins *Mt1* and *Mt2* as surrogate markers for intracellular copper levels (both $p < 0.0001$) (Fig. 4c)[25]. Unexpectedly, the tumoral oncogene *Mycn* and its associated targets were also significantly downregulated ($p < 0.0001$), which is known to decrease tumor cell proliferation and increase immunogenicity and immune cell infiltration (Fig. 4c,d)[26,27].

The immune compartment consisted of 12,127 immune cells (defined using *Ptprc* [CD45]⁺ expression as *Cd3*-expressing T cells also express low levels of *Mycn*; Fig. 4e) with a diverse repertoire of 13 unique clusters identified in both treatment arms encompassing both lymphoid and myeloid lineages (Fig. 4e–g). This data supports previously reported tumor-associated immune subsets in the *Th-MYCN* model and clinical neuroblastoma samples as reviewed by Wienke and colleagues[28]; however, proportions may vary owing to different analytical techniques and tumor stage[29,30] (Fig. 4f; Supp. Fig. 2a). Of note, copper chelation resulted in increased numbers of CD8⁺ effector T cells yet revealed a marginal decrease in NK cells, the latter contrasting with our previous report[12]. Given animals commenced copper chelation therapy immediately after weaning (tumor diameter ≤2 mm) in the former study, we posit that the recruitment at a larger tumor size herein (tumor diameter ≥ 3–4 mm) may account for this discrepancy. To address this, we performed flow cytometry to determine the frequency of infiltrating NK cells in *Th-MYCN* tumors and observed a marked but non-significant increase with TEPA treatment (Supp. Fig. 2b). Taken together, this suggests that NK infiltration may occur as a secondary effect during the destabilization of the immunosuppressive tumor microenvironment. We also observed a decrease in the infiltration of CD4⁺ naive T cells with TEPA, which may lead to lower levels of immunosuppressive regulatory T cells induced by exposure to TGF-β[31]. Nonetheless, neutrophils still exhibited a profound five-fold increase with treatment which is consistent with our tumoral flow cytometric analysis (Fig. 4e,f; Fig. 2c, d). To assign this neutrophil cluster, we used the recently identified gene markers

*S100a8/a9* which together form the heterodimer calprotectin involved in neutrophil recruitment and activation alongside previously established markers (Fig. 4g)[29,32–34].

Collectively, these data demonstrate the impact of copper chelation in the neuroblastoma tumor microenvironment, including tumoral downregulation of *Mycn* and its targets and increased neutrophil infiltration.

## Copper chelation reinvigorates the anti-tumor immune response via neutrophil signaling

Next, we sought to assess the impact of copper chelation on the immune compartment by mapping biological processes in the treatment versus control arms. We observed enrichment for gene sets related to CD8⁺ T cell expansion, myeloid-associated hypoxia and autophagy responses, lymphocyte and myeloid migration, differentiation, and activation as well as cellular cytokine responses to IFN-γ and the TNF family (Fig. 5a). Further, bar plots and top pathways produced by gene set enrichment analysis for each immune cluster with TEPA treatment indicate metabolic alterations associated with activation of the innate immune response (Supp. Figs. 3,4).

The anti-tumor immune response involves the complex coordination of multiple cell types across both innate and adaptive cell subsets. To understand how these processes may be facilitated, we used Cell-ChatDB to examine changes in cell-cell communication between immune and tumor clusters in the control and TEPA-treated datasets[35]. We observed an overall increase in signaling networks occurring with copper chelation, dictated by the presence and or increased strength of cell-cell interactions (Fig. 5b). Furthermore, copper chelation treatment caused neutrophils to supersede tumor cells when comparing incoming interaction strengths, alluding to the dampening of tumoral-induced immunosuppression (Fig. 5c; Supp. Fig. 5). Given the increase in tumor-infiltrating neutrophils with TEPA treatment, we examined ligand-receptor expression occurring between this cluster and other subtypes and found this was largely driven by the Galectin-9 (*Lgals9*) and TNF (*Tnf*) axes which are known to support CD8⁺ cytotoxic T and NK cell-mediated cytotoxicity (Supp. Fig. 6)[36,37].

We therefore posit copper chelation skews the immunosuppressive tumor microenvironment and reinvigorates the antitumor immune response via enhanced neutrophil signaling.

## Mobilization of copper enhances the antitumor activity of neutrophils

Given the critical importance of copper for effective immune function, we sought to investigate changes in copper metabolism within the immune compartment using our single-cell transcriptomics data[38,39]. Comparing treatment groups, we did not observe changes in a selected set of copper-related genes which may be attributed to the tight homeostatic regulation of copper in non-malignant cells (Fig. 6a)[40]. However, we did observe neutrophil-restricted co-expression of *Slc31a1* (encoding Ctr1), *Atp7a*, and *Steap4*, the major genes responsible for modulating intracellular copper import and export in mammals[41].

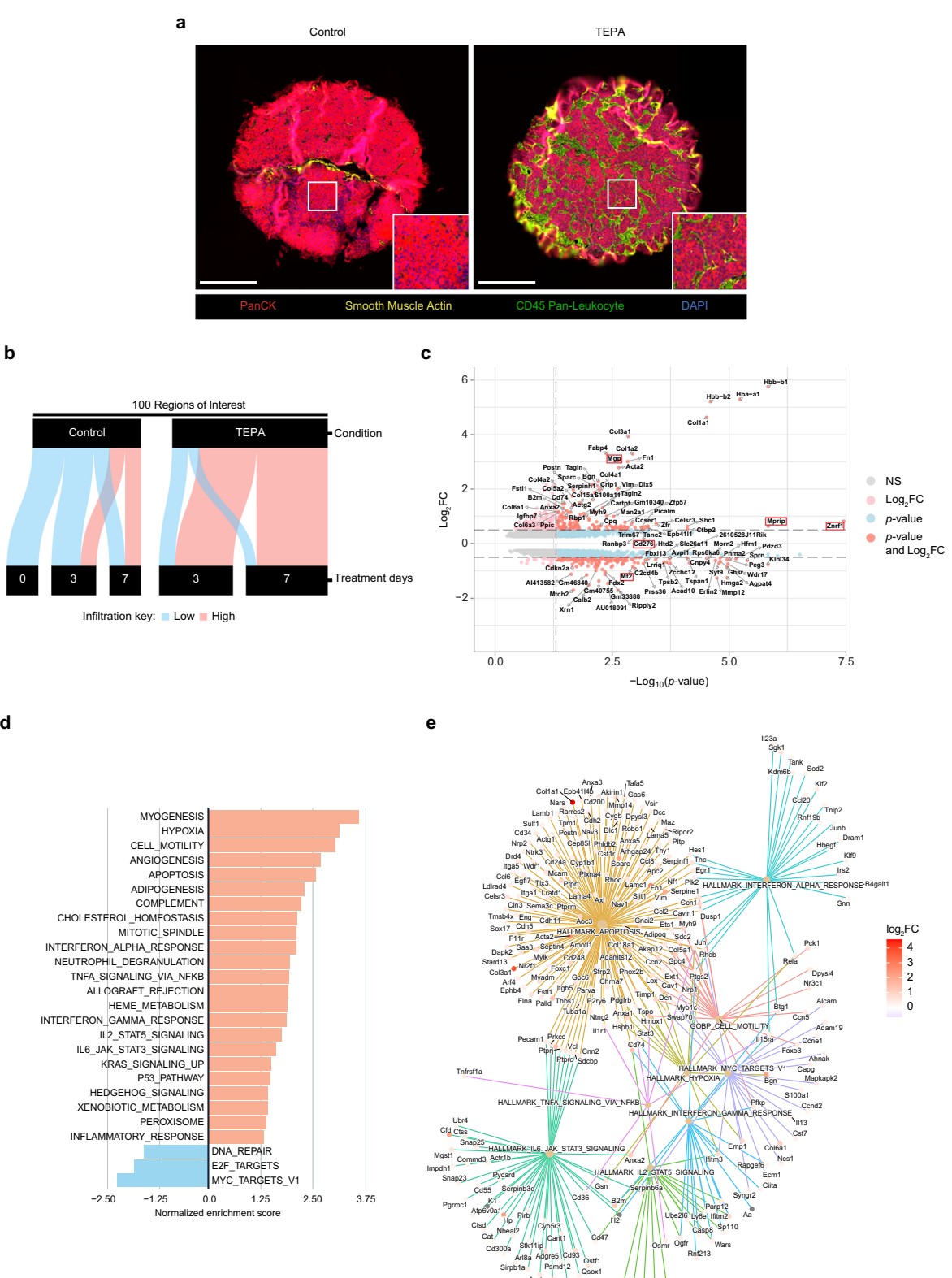

Consistent with our previous observations, copper chelation upregulated genes associated with neutrophil migration and extravasation with the notable downregulation of *Ccr7* (Fig. 6b). *Ccr7* is involved in neutrophil migration to lymph nodes where their accumulation has been implicated in solid tumor progression and metastases[42,43]. Mirroring the M1/M2 spectrum used for macrophage polarization states, an N1/N2 classification has recently emerged for neutrophils to assign anti- and pro-tumorigenic functions, respectively[44,45]. Using published datasets, we curated N1 and N2 gene signatures which revealed that copper chelation strongly promotes the polarization of tumor-infiltrating neutrophils towards a pro-inflammatory N1 phenotype (Fig. 6c,d). This was unsurprising given

**Fig. 3 | Copper chelation reinvigorates antitumor immunity via pro-inflammatory signaling. a** Representative images of a tissue microarray containing control and TEPA-treated *Th-MYCN* tumor cores (*n* = 10/treatment, biological replicates in technical duplicate, total *n* = 20 cores; one independent experiment) and resected after 0, 3 or 7 days for NanoString GeoMx Digital Spatial Profiling, stained with fluorescently conjugated antibodies to PanCK (red), smooth muscle actin (yellow), CD45 (green) with DAPI nuclei stain (blue). Scale bar, 300 μm. One independent experiment. **b** Sankey diagram of *Th-MYCN* tumor cores depicting distribution of CD45-infiltrated low/high regions of interest, plotted against treatment group and duration. **c** Volcano plot of genes upregulated and downregulated in TEPA-treated high versus low regions of interest. False discovery rate (FDR) was adjusted using the two-tailed Benjamini–Hochberg procedure. Thresholds: *p* < 0.1; |log$_2$FC| > 0.5. **d** Gene set enrichment analysis for TEPA-treated high-infiltrated tumor regions compared to low-infiltrated regions presented as a bar plot. **e** Network representation of selected pathways in (**d**) displaying differentially expressed genes as branches. The magnitude of change is reported as log$_2$FC using colored nodes. Abbreviations: FC fold change, PanCK pancytokeratin.

N2 polarization is chiefly mediated by TGF-β which is significantly reduced in both serum and the tumor microenvironment with TEPA treatment (Fig. 2b)[44]. Of note, treatment enriched the expression of the interferon response gene *Isg15*, a marker associated with a mature neutrophil phenotype that can be secreted to promote NK proliferation, DC maturation, and T cell secretion of IFN-γ[46,47]. This was further confirmed through gene set enrichment analysis with copper chelation inducing significant upregulation of the curated N1 anti-tumor phenotype and IFN-γ response pathways (Fig. 6e). IFN-γ stimulation has been demonstrated to enhance N1-associated properties, production of reactive oxygen species, direct and antibody-dependent cellular cytotoxicity, as well as T cell recruitment and activation[48,49].

Together, this suggests that copper chelation promotes the infiltration and polarization of N1 neutrophils in the neuroblastoma tumor microenvironment.

Systemic copper deficiency has been associated with reduced numbers of circulating neutrophils and depressed effector function, which can be rapidly reversed with copper supplementation[50,51]. As neuroblastoma cells exhibit elevated copper content to drive tumor progression (e.g., proliferation, angiogenesis, metastasis), we reasoned this sequestration may create a copper-depleted microenvironment to suppress immune cell function, particularly neutrophils. We therefore postulated that copper chelating agents such as TEPA can redirect the flow of copper ions from the "rich" tumor cells to the "poor" immune cells to slow tumor progression and reinvigorate anti-tumor immunity.

To test this hypothesis, we first validated single-cell findings by transducing the human *MYCN*-amplified neuroblastoma cell line SK-N-BE(2)-C with an *MT1X*-tGFP construct. This allowed us to qualitatively confirm that fluorescent signal (proportional to MT1X expression, a surrogate marker for intracellular copper) was indeed reduced in cells after 24 h of TEPA treatment (Fig. 6f). This suggests that–within a closed in vitro system–tumoral copper is released extracellularly and therefore the copper concentration of media can be assayed to infer copper biodistribution.

To this end, we devised a staggered co-culture system allowing us to accurately trace copper ion flow during a simulated neuroblastoma-neutrophil interaction using the SK-N-BE(2)-C cell line. Conditioned media was collected, and copper concentration was assayed before and after the addition of naive circulating neutrophils isolated from healthy donors. We noted copper concentration was marginally decreased in media isolated from untreated neuroblastoma cells relative to control media which indicates tumor cells sequester available copper to support their proliferation (Fig. 6g, gray bars). Expanding on this observation, SK-N-BE(2)-C cells monitored over 48 h continued to sequester copper from media to support their growth (Supp. Fig. 7a,b). In line with our hypothesis, we observed a repletion of copper in media after TEPA treatment which was subsequently taken up by neutrophils after incubation (*p* = 0.0024) (Fig. 6g, green bars). This phenomenon did not occur in untreated neuroblastoma cells, indicating that copper chelation therapy modulates levels of copper in the tumor microenvironment to facilitate copper uptake by neutrophils.

In clinical support, elevated expression of the copper export protein ATP7A (encoded by *ATP7A*) was significantly associated with a T cell-infiltrated tumor microenvironment in a wide range of solid human pediatric malignancies (*p* = 8.9e−05), and was also associated with the improved survival of patients with neuroblastoma (*p* = 7.4e−03) (Supp. Fig. 7c,d). Together, these data indicate that redistribution of copper within the tumor microenvironment is highly advantageous and is likely expedited with the use of copper chelation therapy.

To understand how this copper uptake influences neutrophil activity, we performed a series of functional studies using isolated human neutrophils. Following incubation in conditioned media, qPCR analysis of neutrophils revealed that the expression of genes associated with intracellular copper (*MT1X*), migration (*S100A8*), and pro-inflammatory activity (*ISG15*) were upregulated only in media from TEPA-treated neuroblastoma cells (Fig. 6h)[46,52]. Consistent with these results, we observed a marginal increase in migratory capacity towards SK-N-BE(2)-C cells when pre-treated with TEPA, supporting our previous in vivo observations (Fig. 6i). Additionally, we observed increased ADCC activity against GD2+/*MYCN*-amplified Kelly cells opsonized with anti-GD2 antibody when pre-treated with TEPA (*p* < 0.0001) (Fig. 6j).

These findings demonstrate that copper chelation can stimulate the recruitment of pro-inflammatory neutrophils and enhance N1 effector functions associated with an anti-tumor response.

## TETA plus anti-GD2 antibody immunocombination therapy offers a curative strategy for neuroblastoma

While the *Th-MYCN* model is considered the standard for the preclinical study of *MYCN*-amplified neuroblastoma, manual palpation is used to determine tumor burden. To enable improved monitoring of tumor growth kinetics, we utilized a syngeneic model generated by the subcutaneous injection of NXS2 cells into immunocompetent A/J mice[53]. This model has been widely used to study anti-GD2-directed treatments including combination therapies[54,55]. Having demonstrated the capabilities of copper chelation therapy, we sought to evaluate the feasibility of repurposing TETA (triethylenetetramine; an analog of TEPA marketed as Cuprior), an FDA-approved copper chelating agent for the treatment of Wilson's Disease, a genetic disorder resulting in excess copper accumulation in the body.

Using the NXS2 model, animals commenced treatment (saline control or TETA, 400 mg/kg) 1 week after inoculation, with copper chelation observed to slow tumor growth as a single agent (Fig. 7a). Subsequently, dissociated tumors were evaluated for neutrophil infiltration via flow cytometry with TETA similarly observed to increase the frequency of CD11b+Ly6G+ neutrophils (*p* = 0.029) (Fig. 7b). Moreover, treatment with TETA similarly increased neutrophil infiltration in the AB1-HA/BALB/c syngeneic model of mesothelioma (*p* = 0.001) but not in the AE17-OVA/C57BL/6 model, which is known to exhibit dysfunctional neutrophil trafficking (Supp. Fig. 8a–c)[56]. This indicates that TETA can similarly stimulate the infiltration of neutrophils, making it a suitable adjuvant for anti-GD2 antibody therapy.

In patients with Wilson's disease, copper chelation therapy can generate bursts of free copper levels which can often induce toxic side effects[57]. To identify potential adverse effects in a tumor-bearing context, peripheral blood was obtained from NXS2-inoculated mice following week-long TETA treatment and subjected to blood

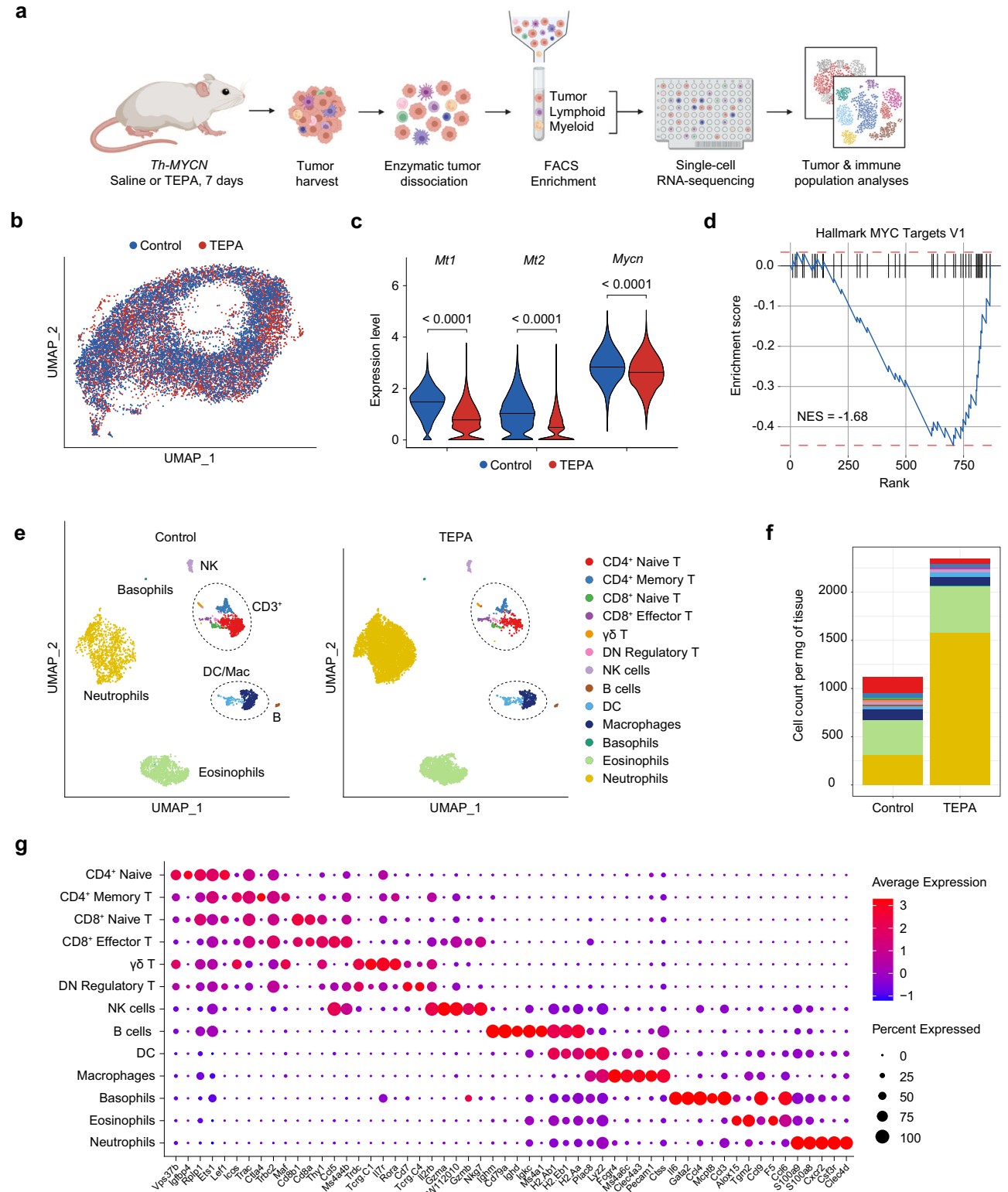

**Fig. 4 | The neuroblastoma tumor microenvironment is sensitive to copper chelation therapy and promotes neutrophil infiltration. a** Experimental design and tumor processing workflow for single-cell RNA sequencing. Schematic created in BioRender. Vittorio, O. (2024). BioRender.com/j88j488. **b** Uniform manifold approximation and projection (UMAP) representation of integrated samples in the tumoral compartment (13,544 cells), colored by treatment group. **c** Violin plots of gene expression levels associated with intracellular copper levels (*Mt1*, *Mt2*) and neuroblastoma oncogene *Mycn*, split by treatment group. Significance was

calculated using two-tailed differential expression analysis using the MAST algorithm after batch correction with *p*-values displayed in figure. Horizontal line indicates data median. **d** Gene set enrichment analysis plot for HALLMARK_MYC_TARGETS_V1 using the fgsea package. **e** Split UMAP representation of immune cell compartment (12,127 cells) according to treatment arm and colored by annotated immune subsets. **f** Bar plot of the proportion of immune cell subsets shown in (**e**). **g** Dot plot of gene expression markers used to classify the immune subsets defined in (**e**).

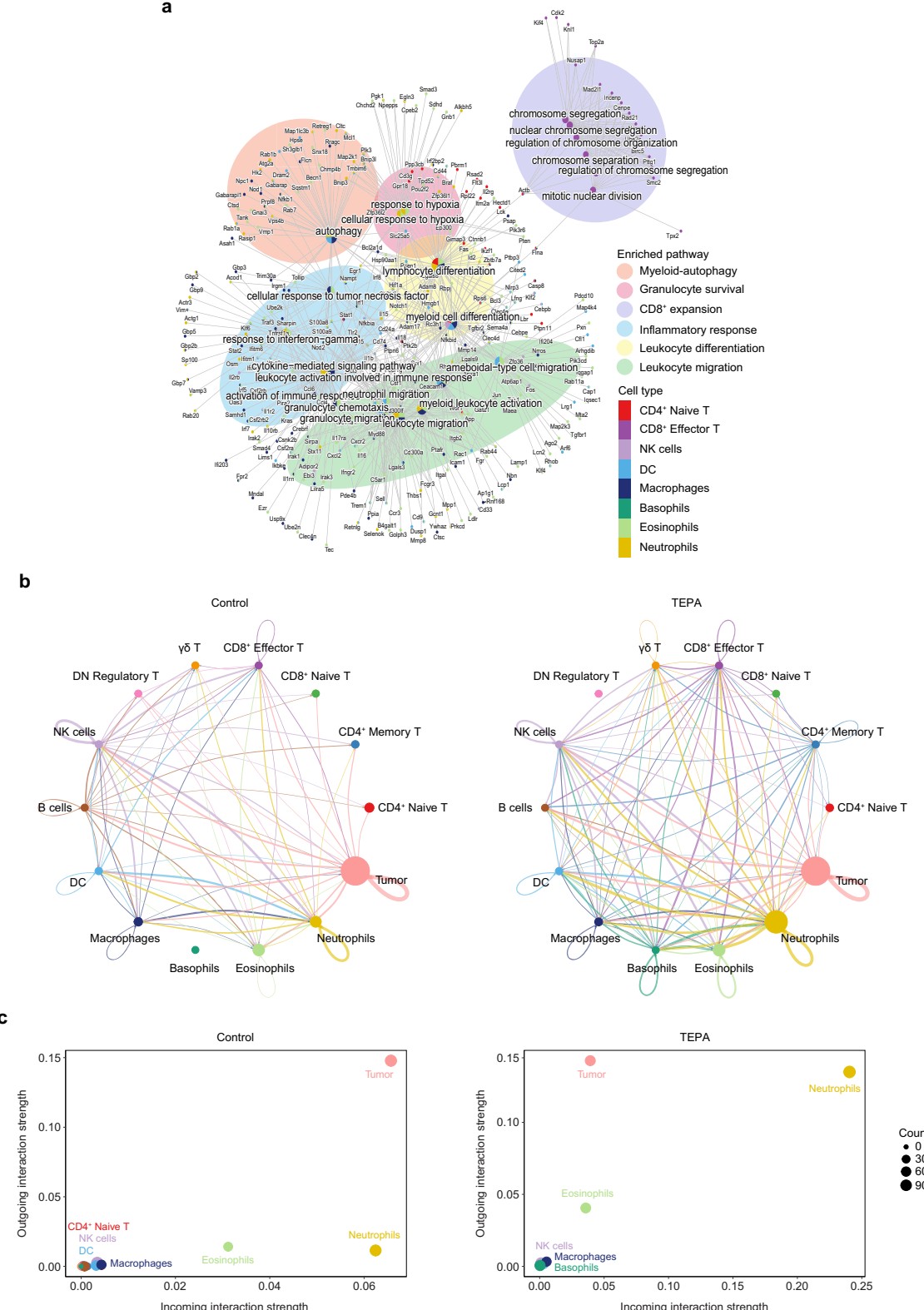

**Fig. 5 | Neutrophils supersede tumorigenic signaling to drive reinvigoration of antitumor immunity. a** Overrepresentation analysis of pathways relatively enriched in TEPA-treated immune cell clusters compared to control, presented as nodes with associated genes as branches using single-cell RNA sequencing. **b** Circle plot of the aggregated cell-cell communication networks in control and TEPA-treated single-cell samples. Edge width is proportional to the number of ligand-receptor interactions between cell types. **c** Scatter plots comparing the outgoing and incoming interaction strengths between control and TEPA-treated samples.

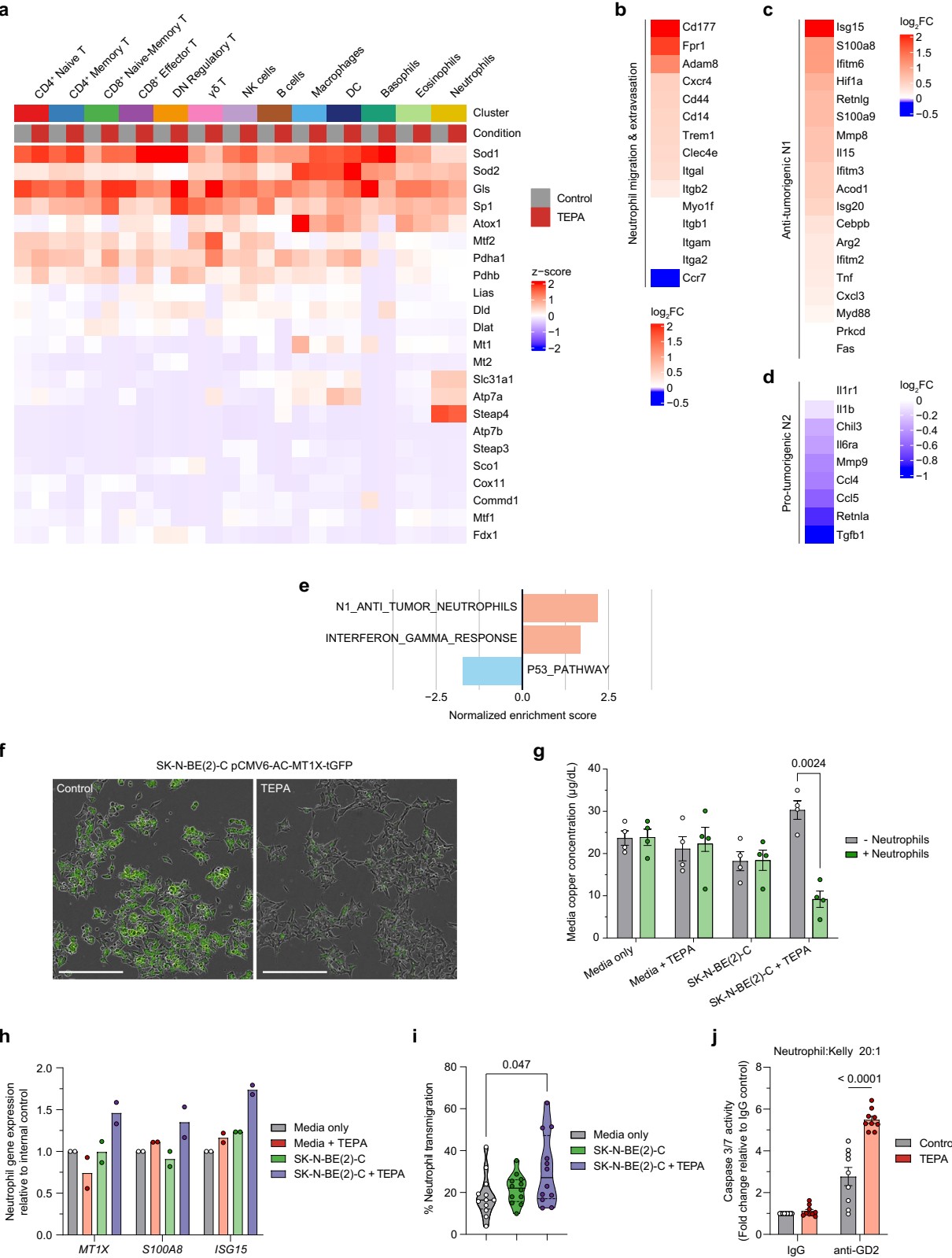

chemistry analysis. TETA treatment did not impact any analyte concentrations associated with hepatic, renal, or overall systemic disorders compared to the control (Supp. Fig. 9). This is particularly important when considering the liver as the systemic reservoir of copper. Overall, these results validate TETA as a non-toxic and efficacious alternative to TEPA to favorably remodel the neuroblastoma tumor microenvironment.

This encouraging data prompted us to investigate the effect of TETA and anti-GD2 antibody as a combination therapy in the NXS2 model. The treatment schedule was similar to that utilized in the TEPA/ *Th-MYCN* model; however, treatment was ceased after four cycles of combination therapy to evaluate relapse rates (Fig. 7c). To validate model sensitivity, we confirmed GD2 expression on the NXS2 cell line by flow cytometry prior to inoculation (Supp. Fig. 8d).

**Fig. 6 | Copper chelation facilitates infiltration and N1-polarization of neutrophils via copper mobilization to exert an anti-tumor response. a** Heatmap comparing the average expression of genes associated with copper metabolism across treatment arms within immune cell clusters. **b** Heatmap of $log_2FC$ in expression between treatment arms for genes associated with migration and extravasation within the neutrophils cluster. Heatmap of $log_2FC$ in expression for genes associated with (**c**) N1 anti-tumorigenic or (**d**) N2 pro-tumorigenic neutrophil phenotypes from control vs. TEPA-treated *Th-MYCN* tumors within the neutrophils cluster. Data presented in (**a–d**) were obtained from single-cell RNA sequencing with relevant cell values averaged and scaled. **e** Gene set enrichment analysis shows top pathways relatively enriched in TEPA-treated neutrophils. The "N1_ANTI-TUMOR NEUTROPHILS" signature was constructed using the N1-associated genes listed in (**b**). **f** IncuCyte cell imaging of neuroblastoma cell line SK-N-BE(2)-C transfected with a plasmid encoding tGFP-tagged MT1X protein following 24 h of TEPA treatment (10× objective). Representative image obtained from one independent experiment. Scale bar, 100 µm. **g** Concentration of copper in conditioned media before and after 30 min incubation with naive neutrophils isolated from healthy donors. Data are presented as mean ± SEM, $n = 4$/condition, biological replicates (healthy donors), three independent experiments. Significance was calculated using a two-tailed paired *t*-test with *p* value displayed in the figure. **h** qRT-

PCR analysis for the expression of genes in human neutrophils associated with intracellular copper (*MT1X*), migration (*S100A8*) and pro-inflammatory activation (*ISG15*) obtained after 30 min incubation in conditioned media as per (**e**). Data are presented as mean, $n = 2$ biological replicates (healthy donors), one independent experiment. **i** Transwell migration assay of neutrophils towards untreated or TEPA-treated SK-N-BE(2)-C cells. Migrated neutrophils were counted using flow cytometry and percentage transmigration was calculated relative to input cells. Data are presented in a violin plot, $n = 2$ biological replicates (healthy donors) in triplicate, two independent experiments. Data minima and maxima values are as indicated, the median (solid line), and the first and third quartiles (dotted horizontal lines). Significance was calculated using an ordinary one-way ANOVA with Tukey's post-hoc test with *p*-value displayed in the figure. **j** Antibody-dependent cytotoxicity assay against the Kelly neuroblastoma cell line using neutrophils isolated from healthy donors in the presence of anti-GD2 antibody (1 µg/ml) with caspase 3/7 activity quantified after 8 h. Data are presented as mean ± SEM, $n = 3$/condition, biological replicates (healthy donors) in triplicate, one independent experiment. Significance was calculated using a two-tailed Mann–Whitney *U* test with *p*-value displayed in figure. Abbreviations: FC fold change, tGFP turbo green fluorescent protein. Source data are provided as a Source Data file.

Upon commencement of the treatment schedule, mice experienced slight weight loss attributed to the introduction of daily gavage as a stressor. Once accustomed, all treatments were well-tolerated including the immunocombination arm with no adverse events reported (Fig. 7d). As a highly aggressive model of neuroblastoma, control arms exhibited rapid tumor expansion with the immunocombination arm observed to effectively restrain tumor growth (Fig. 7e). Control arms (Saline + vehicle; Saline + IgG2a) were highly similar in terms of tumor growth and survival which suggests the isotype antibody did not elicit an immunogenic effect. It is noted that treatment with TETA + IgG2a sufficiently reduced tumor burden in a single animal which succumbed to a rapid relapse following cessation of treatment on day 42, potentially due to tumor escape. In contrast to the *Th-MYCN* model (Fig. 1b,c), anti-GD2 therapy alone did not mediate a substantial anti-tumor effect though the addition of TETA produced a remarkable anti-tumor effect (median survival: 29 days vs. 19 days in Saline + anti-GD2, $p = 0.0091$) leading to durable eradication in ~40% of animals (Fig. 7f). Notably, these animals did not exhibit any signs of relapse following cessation of treatment up to the experimental endpoint of 90 days. Overall, the addition of TETA to anti-GD2 therapy significantly extended survival when compared to the respective monotherapy arms.

To examine changes in the immune compartment ($NCR1^+$ NK cells, $CD8^+$ cytotoxic T cells, and $CD11b^+$ myeloid cells) occurring with treatment, we performed OPAL multiplex immunohistochemistry in tumors resected 14 days post-treatment (Fig. 7g). Across all subsets examined, TETA and anti-GD2 monotherapies were comparable and significantly promoted immune infiltration when compared to the control (Fig. 7h). The addition of TETA to anti-GD2 therapy predominantly enhanced myeloid infiltration ($p = 0.023$), with no observed changes occurring in NK or cytotoxic T cell frequencies (Fig. 7h). Remarkably, the combination group exhibited exceptional tumor control, alluding to the TETA-mediated reinvigoration of effector functions associated with an anti-tumor immune response.

Collectively, our results reinforce the critical role of copper as a modulator of the neuroblastoma tumor microenvironment. We have demonstrated the ability of copper chelating agents to successfully circumvent immune evasion phenotypes and elicit a robust anti-tumor immune response. Moreover, we have confirmed that TETA is a highly effective, non-toxic, and specific copper chelating agent. Study findings provide evidence for repurposing the clinically approved copper chelating agent Cuprior as an immunomodulatory agent to potentiate anti-GD2 immunotherapy and improve responses in patients with neuroblastoma.

## Discussion

Anti-GD2 immunotherapy has improved the survival of patients with high-risk neuroblastoma with efficacy likely hampered by the immunosuppressive tumor microenvironment[8]. Current efforts are therefore focused on the characterization and therapeutic targeting of the tumor microenvironment to improve patient responses. Here, we report that copper chelation therapy induces a marked increase in the infiltration of pro-inflammatory neutrophils and reprograms the neuroblastoma tumor microenvironment to reinvigorate anti-tumor immunity. These findings establish the rationale for using copper chelation as an immune-priming strategy to potentiate the effects of anti-GD2 immunotherapy.

Leveraging spatial and single-cell transcriptomics, cytokine profiling, and multiplex immunohistochemistry, we demonstrate that copper chelation therapy can induce the downregulation of *Mycn* expression and its targets whilst simultaneously enhancing the infiltration and activation of both lymphoid and myeloid immune lineages. A key driver of high-risk disease, *MYCN* amplification has been strongly associated with tumor immune escape and adverse prognosis relative to low-risk disease[58]. Our results are consistent with a recent study showing that pharmacological targeting of *MYCN* enhances activation of interferon pathways and IP-10/CXCL10 expression to promote T cell recruitment and activation[26]. In parallel, another study reported that T cell infiltration was accompanied by NK cell infiltration, suggesting a coordinated recruitment, and was associated with a favorable prognosis[59]. Together, our data demonstrates that copper chelation can remodel the tumor microenvironment to restore immune recruitment and stimulate effector functions.

Although myeloid cells have been historically associated with an immunosuppressive phenotype in neuroblastoma, recent investigations into subset diversity have revealed a heterogeneous range of microenvironment-dependent populations (reviewed in ref. 60). Herein we report the in silico and in vivo identification of major myeloid-associated cell subsets including monocytes (differentiating into macrophages, dendritic cells) and granulocytes (differentiating into neutrophils, basophils, and eosinophils), which exhibited enhanced activation consistent with an anti-tumor immune response with short-term copper chelation therapy. Of strong interest, treatment was found to polarize neutrophils towards an N1 pro-inflammatory phenotype which was subsequently validated using functional assays in human neutrophils.

Despite their abundance in both mice and humans, neutrophils are underrepresented in single-cell RNA-sequencing datasets owing to relatively low mRNA content resulting in fewer transcripts[61]. This

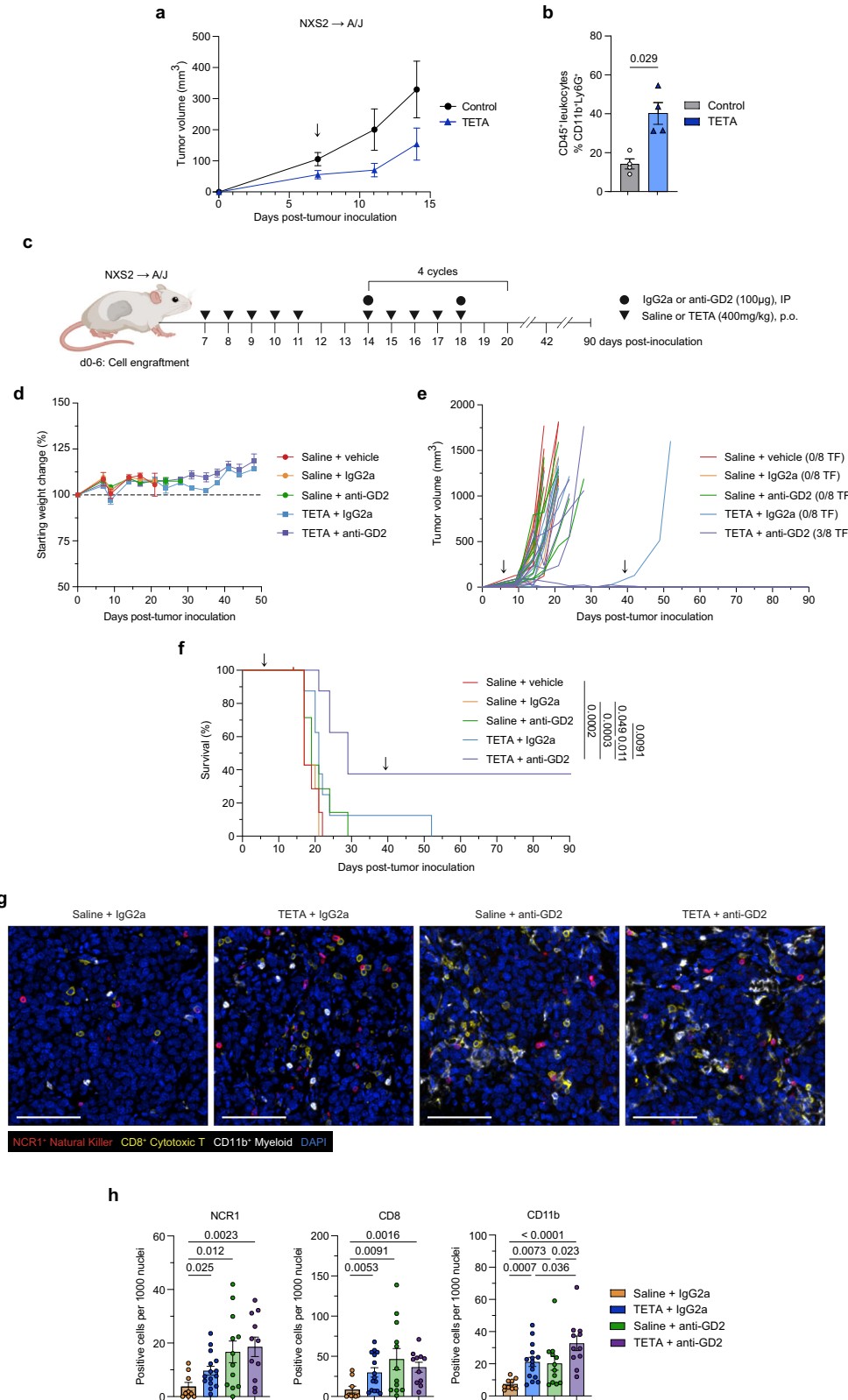

technical hurdle reflects a known advantage of the BD Rhapsody system in capturing an exceptionally high number of mRNA molecules per cell, leading to unprecedented insight into neutrophil heterogeneity and function[62]. Although the increase in CD11b[+] cells by immunohistochemistry was explicitly defined as neutrophils as per single-cell transcriptomics and flow cytometry, future profiling efforts should seek to elucidate their diversity and plasticity within the neuroblastoma tumor microenvironment as per other solid cancers[63,64].

As copper deficiency is a cause of reversible neutropenia, we posit that neuroblastomas mirror this occurrence within the local tumor microenvironment to negatively regulate immune function. In support of this hypothesis, we report that copper chelation can facilitate copper transfer between neuroblastoma cells and neutrophils in vitro and

**Fig. 7 | Copper chelating agent TETA synergizes with anti-GD2 therapy to mediate antitumor activity in the syngeneic NXS2 model of neuroblastoma. a** Tumor growth kinetics in a syngeneic model of neuroblastoma involving the subcutaneous inoculation of A/J mice with NXS2 cells. Animals commenced treatment 1 week after inoculation (black arrow) and were treated by oral gavage with saline (control) or TETA (400 mg/kg/day) for 7 days before blood and tumor collection. Data are presented as mean ± SEM, $n = 6$ (Control) and $n = 7$ (TETA) biological replicates, two independent experiments. **b** Flow cytometric analysis of neutrophil frequencies in NXS2 tumors after 1 week of TETA treatment. Data are presented as mean ± SEM, $n = 4$ (both groups) biological replicates, one independent experiment. Significance was calculated using a two-tailed Mann–Whitney $U$ test with $p$-value displayed in the figure. **c** Experimental design of the syngeneic NXS2 → A/J preclinical model and immunocombination dosing strategy. Schematic created in BioRender. Vittorio, O. (2024). BioRender.com/c53h577. For (**d**–**f**), arrows indicate the treatment period. **d** Relative weight change in tumor-bearing mice measured from date of inoculation. **e** Tumor growth kinetics of individual

tumors measured from date of inoculation. **f** Kaplan–Meier survival curves of tumor-bearing mice measured from date of inoculation. Statistical pairwise comparisons were calculated using a two-tailed Mantel–Cox log-rank test with $p$-values displayed in figure. For (**d**–**f**), data are presented as mean ± SEM, $n = 8$ (all groups) biological replicates, one independent experiment. **g** Representative images of merged OPAL multiplex immunofluorescence spectra depicting the tumoral distribution of NCR1+ natural killer cells (red), CD8+ cytotoxic T cells (yellow), CD11b+ myeloid (white) and DAPI nuclei stain (blue) in NXS2 neuroblastoma tumor tissue 14 days post-treatment. Scale bar, 100 μm. **h** Immune cell quantification of (**g**) as positive counts per 1000 nuclei. Data presented as mean ± SEM, $n = 3$ (Saline + IgG2a); $n = 4$, (Saline + anti-GD2, TETA + anti-GD2), $n = 5$ (TETA + IgG2a) biological replicates with three technical replicates, one independent experiment. Significance was calculated using a two-tailed Mann–Whitney $U$ test with $p$ values displayed in the figure. Abbreviations: IP intraperitoneal, p.o. oral gavage, TF tumor-free. Source data are provided as a Source Data file.

enhance effector function. This work presents a biological phenomenon for neutrophils and future studies should explore the applicability of this immunosuppressive mechanism in other solid malignancies that may benefit from copper chelation as a treatment adjuvant[9].

In neuroblastoma, neutrophils are recognized as a key effector of GD2 antibody therapy, mediating the eradication of opsonized cells via Fc gamma receptor IIa (FcγRIIa) binding[65]. In vitro mechanistic studies have recently revealed that neutrophil-mediated ADCC occurs predominantly via trogocytosis with cytotoxic activity enhanced by stimulation with G-CSF[65,66]. The role of neutrophils in neuroblastoma has also been evidenced clinically with superior responses to anti-GD2 antibody therapy obtained following the addition of GM-CSF to the treatment regimen, and was further enhanced in patients who possessed a polymorphic variant of FcγRIIa[67,68]. Although there are recognized distinctions between mouse and human neutrophils, we have demonstrated that copper chelation can similarly promote neutrophil activity in both organisms to potentiate anti-GD2 antibody efficacy in neuroblastoma[69,70].

Virtually all patients with neuroblastoma experience substantial treatment-associated acute toxicities with survivors often reporting late effects, prompting the search for less toxic anti-cancer agents. Drug repurposing is becoming an increasingly attractive approach to reduce the time, resources, and risks associated with de novo drug research and development[71]. We present compelling evidence to support the clinical repurposing of TETA, a non-toxic copper chelating agent indicated for Wilson's disease, to potentiate anti-GD2 antibody therapy. For patients with Wilson's disease, treatment with TETA is lifelong and therefore extensive clinical and safety data is available[72]. The preclinical TETA dose was determined to be clinically relevant in pediatric patients using the body surface area normalization method[73]. Clinical data reports that TETA is efficacious in pediatric patients, features a low occurrence of documented side effects (often reversible after dose reduction or discontinuation), and is available as an oral formulation for ease of administration[74,75].

In conclusion, we report that copper chelation is a selective and non-toxic strategy to disrupt the immunosuppressive neuroblastoma tumor microenvironment and extend the benefits of anti-GD2 antibody therapy. This work provides a strong rationale for clinical testing of this immune-based combination therapy in patients with neuroblastoma.

## Methods
### In vivo studies
For neuroblastoma models, experimental procedures were approved by the University of New South Wales Animal Care and Ethics Committee (Approval numbers: 20/25B, 21/96B, and 18/97B), and performed in accordance with the 1985 Animal Research Act (New South

Wales, Australia) and the National Health and Medical Research Council 2013 Australian Code of Practice for Care and Use of Animals for Scientific Purposes. All animals were housed in a specific pathogen-free facility, with a maintained temperature of 22–24 °C on a 12 h day/night cycle. Mice were housed in Ventirack cages (Tecniplast, Italy), provided food and water ad libitum, and received environmental enrichment.

For in vivo treatment: the copper chelating agents tetraethylenepentamine pentahydrochloride (TEPA [$C_8H_{23}N_5 \cdot 5HCl$], Sigma, USA; #357683) and triethylenetetramine tetrahydrochloride (TETA [$C_6H_{18}N_4 \cdot 4HCl$], Sigma, USA; #161969) were freshly dissolved in medical-grade saline and administered by oral gavage at 400 mg/kg. The anti-GD2 monoclonal antibody (clone 14G2a, #BE0318) and IgG2a isotype control (clone IgG2a, #BE0085) were obtained from BioXCell (USA) and freshly diluted in medical-grade saline and administered intraperitoneally in a 100 μg bolus.

The *Th-MYCN* (Tg(Th-MYCN)41Waw, 129/SvJ *Ter* backcross) model of neuroblastoma was kindly provided by Prof Michelle Haber (Children's Cancer Institute, Australia) and approved for use by the Institutional Biosafety Committee. *Th-MYCN* mice were maintained onsite and genotyped with only homozygous mice used experimentally. For tumor characterization studies, male and female animals were recruited when a small tumor (3–4 mm in diameter) was palpated and were treated for 7 days with saline or TEPA (400 mg/kg) before tumor collection and sectioning for downstream applications.

For immunotherapy combination studies, male and female mice were recruited as above and randomly assigned to the following treatment groups: Saline + saline vehicle; Saline + IgG2a; Saline + anti-GD2; TEPA + IgG2a; TEPA + anti-GD2. This recruitment size was selected to obtain adequate tumor material to study the synergy between copper chelation and immunotherapy. Regarding survival experiments, animals ($n = 5$/group/sex) were weighed and palpated regularly for progression, regression or relapse and were sacrificed when tumor diameter reached ≥10 mm as the maximal burden.

Female A/J (A/JOzarc) mice were obtained from the Ozgene Animal Resources Centre (Perth, Australia) and the NXS2 cell line was kindly provided by Prof. Holger Lode (University of Greifswald, Germany). Animals aged 6–7 weeks were inoculated subcutaneously with $1.5 \times 10^6$ NXS2 cells (derived from a hybrid of the C1300 neuroblastoma cell line and dorsal root ganglion cells) in a 1:1 mix of serum-free Dulbecco's Modified Eagle Media (DMEM; Gibco, USA; #11995065) and Matrigel (Corning, USA; #354234). Tumors were engrafted for 7 days (reaching 50–100 mm³) before commencing treatment with saline or TEPA (400 mg/kg) for 7 days. This dose was selected based on prior optimization studies to ensure adequate tumor material was available to study the synergy between copper chelation and immunotherapy.

Mice were assigned to treatment groups to achieve approximately equal average initial tumor sizes to mitigate bias. Animals were

weighed and tumor volumes measured twice weekly using digital callipers (calculated as $0.5 \times$ length $\times$ width$^2$). For immunotherapy combination studies, mice were assigned to the following treatment groups: Saline + saline vehicle; Saline + IgG2a; Saline + anti-GD2; TEPA + IgG2a; TEPA + anti-GD2. Animals were sacrificed once tumor volume reached ≥1000 mm$^3$ as the maximal burden in survival experiments.

For mesothelioma models, experimental procedures were approved by the Harry Perkins Institute of Medical Research Animal Ethics Committee (Approval number: AE271) and performed in accordance with guidelines of the and the National Health and Medical Research Council 2013 Australian Code of Practice for Care and Use of Animals for Scientific Purposes. Animals were maintained under standard, specific pathogen-free housing conditions.

Female BALB/c (BALB/cOzarc) and C57BL/6 (C57BL/6JOzarc) mice were bred and maintained at the Ozgene Animal Resources Centre (Perth, Australia) or Harry Perkins Institute of Medical Research (Murdoch and Nedlands, Australia). Animals aged 8–10 weeks were inoculated subcutaneously with $5 \times 10^5$ mesothelioma cell lines, AB1-HA (Balb/c) or AE17-OVA (C57BL/6), both generated from intraperitoneal exposure to crocidolite asbestos. Cell lines were grown in Roswell Park Memorial Institute (RPMI)−1640 supplemented with 20 mM HEPES, 50 µM 2-mercaptoethanol, 100 U/ml of penicillin, 50 µg/ml of gentamicin, 10% fetal bovine serum (FBS), and 50 mg/ml G418 sulfate (Thermo Fisher Scientific, USA; #10131035). Tumors were grown for 1 week until tumors reached 8–10 mm$^2$ (maximal burden: 1500 mm$^3$) then treated with saline or TETA for 7 days (400 mg/kg) via oral gavage and tumor kinetics measured as described above followed by tumor collection for flow cytometric analysis.

We confirm that all animals were euthanized by exposure to $CO_2$ upon reaching experimental endpoints and no animals exceeded the maximal tumor burden.

## OPAL multiplex immunohistochemistry (IHC)

Tumor sections were formalin-fixed and paraffin-embedded (FFPE) by the Katharina Gaus Light Microscopy Facility (KGLMF) at the University of New South Wales. Tumors were sectioned at 4 µm and in preparation for staining, slides were baked for 1 h at 58 °C before deparaffinization and rehydration using a Gemini AS Automated Slide Stainer (Epredia, USA). Chromogen-based IHC analysis was performed using the BOND-RX automated staining system (Leica Biosystems, USA). Spleens obtained from tumor-bearing *Th-MYCN* mice were used as a control for single antibody and OPAL multiplex optimization. The following antibodies were all obtained from Abcam, UK: rabbit monoclonal NCR1 (clone EPR23097-35, 1:500, #ab233558, Lots GR3347976-1, GR3400659-6, EDTA pH 8-9 antigen retrieval), rabbit monoclonal CD8a (clone EPR20305, 1:1000, #ab209775, Lots GR3194554-6, GR3334378-1, GR3275780-5, EDTA pH 8-9 antigen retrieval), and rabbit monoclonal CD11b (clone EPR1344, 1:20000, #ab133357, Lots GR3209213-12, GR3345111-10, citrate pH 6 antigen retrieval). Immunofluorescent signal was visualized using the OPAL 7-color Automation IHC kit (Akoya Biosciences, USA; #NEL871001KT) using TSA dyes 650, 570, and 520 respectively, and counterstained with spectral DAPI. Optimization also included the sequence of antibodies which was determined to obtain the same dynamic ranges between each fluorophore to avoid signal "cross-talk" known as the umbrella effect[76]. Labeled slides were imaged using the Vectra Polaris system (Akoya Biosciences, USA) using auto-exposure at 20× magnification. Whole slides were imaged using Phenochart software v1.1.0 (Akoya Biosciences, USA) via multispectral field scans. Acquired images were unmixed using Inform v2.5.1 (Akoya Biosciences, USA) and subsequently stitched together in HALO suite v3.6 (Indica Labs, USA) to produce a whole-slide multispectral TIFF image. Nuclei segmentation was performed using HALO artificial intelligence which leverages machine learning to train the software. Cell phenotyping analysis was performed using the HighPlex FL v4.2.5 module using defined thresholds for nuclear detection and minimum fluorescence intensity. The classifier was trained on all slides and necrotic areas were excluded prior to analysis. Tumor sections (necrotic areas excluded) were selected at random and were subjected to analysis and cells were classified as positive if fluorescence intensity exceeded a predefined threshold. Cell density was determined as the number of positive cells per 1000 defined nuclei per tumoral section.

## Peripheral and tumoral cytokine profiling

Peripheral blood was obtained from control and TEPA-treated *Th-MYCN* animals after 1 week and allowed to clot at room temperature for 15 min. Samples were centrifuged at $2000 \times g$ for 10 min at 4 °C to obtain serum. Serum cytokine levels were measured using the Quantitative Mouse Cytokine Antibody Array (Abcam, USA; #ab197465) as per manufacturer's instructions.

Frozen tumor sections were homogenized in RIPA lysis buffer supplemented with 1× Protease and Phosphatase Inhibitors (Roche, USA; #04693159001) using the TissueRuptor II (Qiagen, Germany; #9002755). To standardize cytokine concentrations, total extracted protein was calculated using the Pierce bicinchoninic acid (BCA) Protein Assay Kit (Thermo Fisher Scientific, USA; #23225) as per the manufacturer's instructions. Tumor cytokine levels were measured using the 36-Plex Mouse ProcartaPlex Panel 1A (Thermo Fisher Scientific, USA; #EPX360-26092-901) as per manufacturer's instructions. A Luminex MAGPIX System (Luminex Corporation, USA) was calibrated with MAGPIX Calibration and Performance Verification Kits (Millipore, USA) and data acquired using xPONENT software (Luminex Corporation, USA). Acquired data was analyzed using Multiplex Analyst software v5.1 (Merck, Germany) as the Median Fluorescent Intensity (MFI) with spline curve-fitting for calculating analyte concentrations in samples. Transforming growth factor-beta (TGF-β) levels were determined using an enzyme-linked immunosorbent assay kit (Invitrogen, USA; #BMS608-4) according to the manufacturer's instructions. Samples were diluted 1:10 with assay diluent before running with resulting concentrations normalized to extracted protein.

## Flow cytometry immunophenotyping

For neuroblastoma tumors, fresh sections were roughly minced and incubated in a tumor digestion mix consisting of DMEM supplemented with 25 µg/mL DNase I (Sigma-Aldrich, USA; #11284932001) and 20 µg/mL Collagenase IV (Worthington Biochemical, USA; #LS004186) for 1 h on an orbital shaker at 37 °C at 130RPM. A single-cell dissociation was achieved by passing the mixture through a 70µm MACS SmartStrainer (Miltenyi Biotec, Germany; #130-110-916). Cells were pelleted at $330 \times g$ for 5 min and resuspended in room temperature ACK Lysis Buffer to remove contaminating erythrocytes. Cells were stained in in fluorescence-activated cell sorting (FACS) Buffer (1× Phosphate buffered saline [PBS]/1% FBS/0.5 mM EDTA) for the following surface antibodies: CD45-BV510 (clone 30-F11, 1:250, BD Biosciences, USA; #563891, Lot 169522), CD11b-APC-ef780 (clone M1/70, 1:400, Thermo Fisher Scientific, USA; #47-0112-82, Lot 2272759), Biotin-Ly6G (clone 1A8, 1:300, BioLegend, USA; #127604, Lot B161712) and conjugated in-house with Streptavidin-BUV737 (clone IM7, BD Biosciences, USA; #612775, Lot 0233661), MHC-II-BV711 (clone M5/114.15.2, 1:250, BD Biosciences, USA; #563414, Lot 1146367), CD64-AF647 (clone X54-5/7.1, 1:100, BD Biosciences, USA; # 558539, Lot 0030911), CD11c-BV421 (clone HL3, 1:300, BD Biosciences, USA; #562782, Lot 3319725), and Ly6C-PE-Cy7 (clone AL-21, 1:300, BD Biosciences, USA; #560593, Lot 0058963). Zombie UV dye (1:500, BioLegend, USA; #423108) was used as a cell viability marker. Sample acquisition was performed using BD FACSAria III (BD Biosciences, USA) and analyzed using FlowJo v10 (TreeStar, USA). A representative gating strategy is presented in Supplementary Fig. 10a.

For mesothelioma tumors, fresh sections were roughly minced and incubated in a tumor digestion mix consisting of PBS/2% FBS

supplemented with 100 μg/mL DNase I (Worthington Biochemical, USA; #LS006331) and 1.5 mg/mL Collagenase IV (Sigma-Aldrich, USA; #C4-22-1G) for 1 h at 37 °C at 180RPM. Cells were strained and pelleted as above before resuspension in FACS buffer and stained for the following surface antibodies: CD45-Spark Violet (clone 30-F11, 1:250, Biolegend; #103179, Lot 7005686), CD3-FITC (clone 17A2, 1:500, Cytek Biosciences; #35-0032-U100, Lot C0032100223353), CD11b-Spark YG (clone M1/70, 1:400, Biolegend; #101281, Lot 8368394) and Ly6G-PerCP (clone 1A8, 1:1000, Biolegend, #127653; Lot 38873). ViaDye Red (1:1000, Cytek Biosciences; #R7-60008, Lot F-100322-02) was used as a viability marker. Samples were analyzed on a Cytek 5L Aurora (Cytek Biosciences, USA) with 200,000 events collected per sample. Analyses were completed on FlowJo v10 (TreeStar, USA). A representative gating strategy is presented in Supplementary Fig. 10b.

NXS2 cells were stained in FACS buffer as above and stained for surface GD2-BV650 (clone 14.G2a, 1:150, BD Biosciences; #563705, Lot 0219086) with 7-AAD used as a cell viability marker. Sample acquisition was performed using BD FACSAria III (BD Biosciences, USA) and analyzed using FlowJo v10 (TreeStar, USA). A representative gating strategy is presented in Supplementary Fig. 8d.

## Tissue microarray construction and digital spatial profiling (DSP) hybridization
A tissue microarray (TMA) was prepared using 20 *Th-MYCN* tumor samples (10 control, 10 TEPA-treated) in duplicate, cored at 1 mm. Coring sites were chosen at random with areas exhibiting necrosis excluded prior by an experienced histopathologist using a hematoxylin-eosin stain. The formalin-fixed paraffin-embedded TMA block was sectioned at 4 μm and transferred to a Bond Plus slide (Leica Biosystems, USA; #S21.2113.A) and were processed by the NanoString GeoMx DSP Technology Access Program. In brief, slides were hybridized with the GeoMx Mouse Whole Transcriptome Atlas (-18,000 targets) followed by immunofluorescent staining with pan-cytokeratin (PanCK; clone AE1/AE3, Thermo Fisher Scientific, USA; #53-9003-82) for identification of tumor cells, smooth muscle actin (SMA, clone 1A4, Abcam, UK; #ab184675) for extracellular matrix, CD45 (clone D3F8Q, Cell Signaling Technology, USA; #35154) for all hematopoietic cells and DNA GeoMx Nuclear Stain (NanoString, USA; #121303303) for cell nuclei. Post-staining, slides were loaded onto the NanoString GeoMx instrument and scanned. For each tumor core, geometric regions of interest (ROIs) were selected ($n = 120$) and were binarily defined as having high or low immune infiltration, determined by the respective presence or absence of CD45 staining (Supp. Fig. 12).

## Digital spatial profiling data processing
Segments and probes quality control was performed using the Bioconductor package GeomxTools. Of note, twenty ROIs were found to exhibit areas of necrosis and were subsequently excluded. The Seurat package was used to perform the following downstream analyses. Principal Component Analysis (PCA) was used to reduce the dimensionality of the dataset, and 50 PCs were used to retain >85% of variability. Unsupervised clustering did not reveal any pattern of variation independent from treatment and infiltration. Differential gene expression analysis used Seurat's FindMarkers function and assumed negative binomial distribution of the data, with three conditions evaluated within groups of ROIs: treatment versus control; low-infiltration vs. high-infiltration (treated ROIs); treatment vs. control (infiltrated ROIs). Gene set enrichment analysis (GSEA) was performed on the log$_2$-transformed and scaled gene expression matrix of highly-infiltrated vs. poorly-infiltrated TEPA-treated ROIs. The mouse hallmark gene sets from the Molecular Signatures Database (MsigDB) was used, along with selected pathways of interest obtained from Reactome and Gene Ontology (GO) databases (GO:0048870, R-MMU-6798695). Enrichment scores were calculated to evaluate the enrichment of the gene sets within the gene expression profiles of each cell type. Gene expression profiles were ranked based on the magnitude of change of genes significantly differentially expressed between control and treatment. To visualize the results, bar plots were generated per cell type, displaying the top significantly enriched pathways, each associated with a specific negative enrichment score value. To further explore the enriched pathways, network plots were created to highlight the top ten genes associated with each gene set. Non-relevant or less informative pathways were excluded from visualization to emphasize the most relevant findings.

Statistical tests were implemented according to established methods. Multiple testing corrections using the Benjamini–Hochberg procedure (two-tailed) were applied to adjust *p*-values for multiple comparisons. The significance thresholds for determining differentially expressed genes and enriched pathways were determined based on adjusted *p*-values and correspond to *p*-adj < 0.1.

## Single-cell RNA sequencing (scRNA-seq)
*Th-MYCN* tumor sections were dissociated into single-cell suspensions as previously described above except for the use of commercial Stain Buffer (BD Biosciences, USA; #554656) in lieu of in-house FACS buffer for ACK neutralization and flow cytometric staining. To reduce non-specific antibody staining of IgG receptors, $1 \times 10^6$ cells were aliquoted and pre-incubated with Mouse BD Fc CD16/CD32 Block (BD Pharmingen, USA; #553142, Lot 8130843). Cells were incubated with CD3ε-FITC (clone 145-2C11, 1:100, Thermo Fisher Scientific, USA; #11-0031-82, Lot 231864), NK1.1-PE (clone PK136, 1:200, Thermo Fisher Scientific, USA; #12-5941-82, Lot 2142869) and CD11b-BV421 (clone M1/70, 1:200, BioLegend, USA; #101251, Lot B322058). Tumor cells were selected for using previously optimized gating strategy using the absence of described markers. Approximately 50,000 single cells of each subset (CD3-NK1.1-CD11b- tumor cells, CD3$^+$/NK1.1$^+$ lymphocytes and NK cells, and CD11b$^+$ myeloid cells) were sorted into a single tube containing fetal bovine serum (Gibco, USA; #10100-147) using the BD FACSAria III (BD Biosciences, USA). A representative gating strategy is presented in Supplementary Fig. 11a.

The BD Rhapsody system (BD Biosciences, USA) was used to capture the transcriptomic data of approximately 25,000 total cells applied per cartridge (1× Control; 1× TEPA-treated). Whole transcriptome libraries were constructed following the BD Rhapsody single-cell whole transcriptome analysis (WTA) workflow according to the manufacturer's instructions. Libraries were quantified using a High Sensitivity DNA chip (Agilent, USA; #5067-4626) on a Bioanalyzer 2200 and the Qubit High Sensitivity double-stranded DNA Assay Kit (Thermo Fisher Scientific, USA; #Q32851). The resulting DNA libraries were sequenced on an Illumina NovaSeq 6000 S4 2 × 150 bp kit to yield an average of 80,000 reads per cell.

Raw sequencing data was converted into gene expression profiles for individual cells using the BD Rhapsody WTA pipeline provided on the Seven Bridges Platform (Seven Bridges Genomics, USA). The pipeline involves the removal of low-quality reads, read alignment, gene expression quantification, and data normalization. To reduce bias during dimensionality reduction, downstream analyses were conducted separately for tumor and immune cell compartments. To isolate the tumor compartment, cells expressing *Mycn* > 0 and *Ptprc* = 0 were retained, excluding cells that express at least one lymphocyte-associated biomarker in the dataset (*Cd3d, Cd3e, Cd3g, Cd8a, Cd8b1*). This yielded a total of 13,560 cells across control and treated groups in the tumor compartment. To isolate the immune compartment, cells expressing *Ptprc* > 0 and *Mycn* = 0 were retained, followed by the addition of *Mycn* > 0 lymphocytes previously excluded from the tumor compartment. This yielded a total of 22,182 cells across control and treated groups in the immune compartment.

In the immune cell compartment, 12,127 cells passed filtering conditions with the following thresholds: 200 <nFeature_RNA > 5500 (number of transcripts); nCount_RNA < 3000 (number of counts);

percent.mt <25 (percentage of mitochondrial counts); and percent.ribo <20 (percentage of ribosomal counts). In the tumor cell compartment, 13,544 cells successfully passed the following filtering conditions: nFeature_RNA > 200; percent.mt <25, and percent.ribo <15. Library-specific thresholds were manually assessed after exploring the empirical distribution of these variables in the 2D space. The CCA Integration method implemented in Seurat was used to perform integration-based anchoring to correct for batch effect. Raw counts were normalized through natural-log transformation.

Separate Uniform manifold approximation and projection (UMAP) plots were generated for the tumor and immune cell compartments. Principal Component (PC) Analysis was used to reduce the dimensionality of the integrated dataset. For the immune cell compartment, 60 PCs were used to retain >70% of variability. Cells are then clustered and sub-clustered via the Louvain algorithm, with a resolution ranging between 0.2 and 0.3. For the tumor cell compartment, 80 PCs were used to retain >86% of variability. Cells were then clustered via the Louvain algorithm, with a resolution of 0.3.

Clustering annotation for the immune cell subsets was initially performed using the scType platform and subsequently curated manually based on the different gene markers identified by the MAST algorithm after batch correction[77]. It was not possible to accurately assign a cell type to 2/15 identified immune clusters due to the minimal number of cells present ($n < 50$) and were therefore excluded from analysis. Final cell annotations were performed using relevant markers well-established in the literature[78].

Pathway enrichment analysis was performed to gain insights into the biological processes and pathways associated with each independent cell cluster. The analysis consisted of two main steps: Gene Set Enrichment Analysis (GSEA) using the fgsea function and Over-Representation Analysis (ORA) using the clusterProfiler package. The mouse hallmark gene sets from the Molecular Signatures Database (MsigDB) were used, along with selected pathways of interest obtained from Reactome, Gene Ontology (GO), and WikiPathways (WP) databases (GO:0048870, R-MMU-6798695; WP3941, WP4466, WP412). Additionally, custom-made signatures were included due to their unavailability in the public databases: copper-related genes[79], N1 anti-tumor phenotype and N2 pro-tumor phenotype[80–83].

To visualize GSEA results, the same approach for the GeoMX DSP data was employed as above. For the ORA analysis, the GO database was used to identify over-represented gene sets within each cell type. Significantly enriched GO terms associated with biological processes and molecular functions were identified using a two-tailed Fisher's exact test. To visualize the results of the ORA analysis, a selection of enriched pathways was made for each cell type. A network plot was generated, with each node representing a pathway and color-coded according to the corresponding cell type. Additionally, enriched genes within each pathway were identified and displayed within the network plot, allowing for a comprehensive view of the genes associated with each enriched pathway, per cell type.

Multiple testing correction using the Benjamini–Hochberg procedure was applied to adjust values from multiple comparisons, as per the GeoMX DSP data. Inference and analysis of cell-cell communication networks were performed with CellChat (v2), using the CellChatDB (v2) as the reference library which contains ~3300 validated ligand-receptor interactions. The number of inferred signaling networks was narrowed down to 16 using a truncated mean of 25% i.e., for a given certain cell group, the average gene expression of a ligand-receptor pair is set to zero if the percentage of expressed cells in that cell group is ≤25%.

## MindRay Hematological analysis

To quantitate the circulating number and frequencies of immune subsets, peripheral blood was obtained from control and TEPA-treated *Th-MYCN* animals after 1 week of treatment. Samples were collected in K2-EDTA tubes (Greiner, Germany; #450532) and analyzed immediately using the Mindray BC-5150 Auto Hematology Analyzer (Mindray, China) according to the manufacturer's instructions.

## Generation of the SK-N-BE(2)-C pCMV6-Ac-GFP cell line

The neuroblastoma cell line SK-N-BE(2)-C was obtained from the American Type Culture Collection with working stocks centrally managed by the Children's Cancer Institute Cell Bank. Both master and working stocks were validated using short tandem repeat profiling and routinely verified as *Mycoplasma* negative. Cells were cultured in 10% FBS/DMEM (see above), incubated under standard conditions (37 °C, 5% CO$_2$, 95% humidity) and passaged routinely upon reaching a confluency of approximately 80%. The neuroblastoma cell line SK-N-BE(2)-C was stably transfected with plasmid pCMV6-Ac-GFP containing the transcript encoding the human MT1X protein (NCBI Reference Sequence: NM_005952.4) with a c-terminal TurboGFP tag (tGFP) (Origene, USA; #RG207116). Cultures were kept under positive selection using Geneticin Selective Antibiotic (G418 [Thermo Fisher Scientific, USA; #10131035]) at 1 mg/mL. Cells were plated and treated with 6 mM TEPA for 24 h and imaged using the IncuCyte Live-Cell Analysis system (Essen BioScience, USA). Merged images were taken in phase contrast and green fluorescence channels (auto-exposure) using a 10× objective.

## Staggered co-culture copper transfer assay

Media alone or containing $0.2 \times 10^6$ SK-N-BE(2)-C cells were seeded in 2% FBS/DMEM before overnight treatment with 1 mM TEPA. The resulting conditioned media was collected immediately after neutrophils were ready for incubation. A portion of the conditioned media was collected as pre-incubation control, designated as: - neutrophil.

Peripheral blood from consenting male healthy donors (University of New South Wales Human Research Ethics Project Approval Numbers: iRECS0865, HC180299) was collected by venipuncture into K2-EDTA tubes (BD Biosciences, USA; #366643). Erythrocytes were sedimented using a 50% volume of Dextran solution (6% Dextran [Merck, USA; #09184-50G-F]; 0.9% NaCl in ddH$_2$O) for 30 min. A Percoll gradient was prepared using 90% Percoll solution (Percoll [Merck, USA; #P4937-100ML] in 10× PBS (No Ca$^{2+}$/Mg$^{2+}$) to form bottom (~55% Percoll solution), middle (~68%) and top (~81%) layers in 1 × PBS (No Ca$^{2+}$/Mg$^{2+}$).

The bottom layer was added to a fresh Falcon tube followed by the middle layer so as not to disturb the interface. The top layer of separated blood containing lymphocytes was collected and centrifuged at $350 \times g$ for 20 min (speed 9 for both acceleration and brake) at 20 °C. Pelleted lymphocytes were gently resuspended in the top Percoll layer which was then layered on top of the previously prepared gradient. The resulting preparations were centrifuged at $700 \times g$ for 20 min (speed 0 for both acceleration and brake) at 20 °C. The resulting Percoll gradient yielded a top lymphocyte layer and a bottom neutrophil layer. The neutrophil layer was obtained and resuspended in 2% FBS/DMEM and centrifuged at $250 \times g$ for 6 min (speed 5 acceleration, speed 9 brake) at 20 °C. The supernatant was aspirated, and neutrophils were resuspended in 2% FBS/DMEM for counting.

$0.25 \times 10^6$ neutrophils were aliquoted and spun in a microfuge at $250 \times g$ for 6 min at 20 °C. Supernatant was aspirated and pellets were gently resuspended in the respective conditioned media for 30 min at room temperature before centrifuging again at $1000 \times g$ at 20 °C for 5 min. Cell-free conditioned media (designated as: + neutrophil) was transferred to fresh Eppendorf tubes for copper concentration analysis.

The concentration of copper in media samples was quantitatively determined using the QuantiChrom Copper Assay Kit (Universal Biologicals, UK; #DICU-250) according to the manufacturer's instructions. Absorbance was determined using a Benchmark Plus Plate Reader with Microplate Manager v5.2.1 (Bio-Rad, USA) at a wavelength of 356 nm.

## Neutrophil gene analysis

$5 \times 10^6$ human neutrophils were resuspended in conditioned media as above and incubated for 1 h at room temperature, centrifuged at $1000 \times g$ at 20 °C and washed twice with PBS. Neutrophil RNA was extracted in TRI Reagent (Invitrogen, USA; #AM9738) according to manufacturer's instructions. A total of 210 µg of RNA was converted to cDNA, synthesized using mixed oligo(dT) (Promega, USA; #C1101) and random hexamer primers (Promega, USA; #C1181) with using MMLV RTase RNase H Minus (Promega, USA; #M5301) according to manufacturer's instructions. Quantitative PCR analysis was performed using SsoAdvanced Universal SYBR Green Supermix (Bio-Rad, USA; #1725272) and the QuantStudio 5 Real-Time PCR System (Thermo Fisher Scientific, USA; #A34322). Primers used are as follows: *ISG15* (F: AAGAGGCAGCGA ACTCATCT and R: AGCTTCAGCTCTGACACCG), *MT1X* (F: GCTTCTCCT TGCCTCGAAA and R: GCAGCAGCTCTTCTTGCAG), *S100A8* (F: AAGG GGAATTTCCATGCCGT and R: ACGTCTGCACCCTTTTTCCT). Quantifications were normalized using internal controls: *ACTB* (F: AGAAAAT CTGGCACCACACC and R: AGAGGCGTACAGGGATAGCA) for *ISG15* and *S100A8*, and *GUSB* (F: TGGTGCGTAGGGACAAGAAC and R: CCAAGG ATTTGGTGTGAGCG) for *MT1X*[84].

## Neutrophil migration assay

Media alone or containing $4 \times 10^4$ SK-N-BE(2)-C cells were seeded in 2% FBS/DMEM before overnight treatment with 1 mM TEPA in the bottom chamber of 24-well Transwell with 5 µm pore polycarbonate membrane (Corning, USA; #3421). Resuspension media was produced similarly except for the addition of TEPA to establish a gradient. Neutrophils were isolated from the peripheral blood of healthy donors as described above with $4 \times 10^5$ cells per chamber suspended in respective resuspension media and incubated at 37 °C for 4 h. After incubation, all cells were collected from the bottom chamber and stained with CD11b-BV41 (clone M1/70, 1:100, BioLegend; #101251, Lot B322058) and SPHERO AccuCount Particles (Spherotech, USA; #QACBP-70-10, Lot AR01) used to determine the absolute number of migrated cells using flow cytometry. Transmigration was calculated as: (# cells migrated/400,000) × 100%. A representative gating strategy is presented in Supplementary Fig. 11b.

## Neutrophil ADCC

$5 \times 10^3$ Kelly cells were seeded in a 96-well plate (Corning, USA; #3599) in 2% FBS/DMEM before overnight treatment with 1 mM TEPA. To induce ADCC, the anti-GD2 monoclonal antibody (clone 14G2a, [BioXCell, USA; #BE0318]) and IgG isotype control (clone IgG2a [BioXCell, USA; #BE0085]) were added to wells (1 µg/ml) 2 h before the addition of neutrophils in an effector:target ratio of 20:1 for 8 h. Neutrophils were isolated from the peripheral blood of healthy donors as detailed above. Incucyte caspase-3/7 green dye (1:4000) (Sartorius, USA; #4440) was added and wells imaged using the IncuCyte Live-Cell Analysis system (Essen BioScience, USA). ImageJ was used to quantify caspase-3/7 activity, with values normalized against the IgG control. No caspase activity was detected in matched conditions containing neutrophils or neuroblastoma cells only, indicating specific neutrophil-mediated ADCC against neuroblastoma cells.

## Neuroblastoma growth kinetics

$3 \times 10^3$ SK-N-BE(2)-C cells were seeded in a 24-well plate (Corning, USA; #3524) in 10% FBS/DMEM and counted daily over 48 h with matching media sampled for copper concentration using the QuantiChrom Copper Assay Kit (Universal Biologicals, UK; #DICU-250) according to the manufacturer's instructions.

## Clinical expression of ATP7A

Relative expression levels of *ATP7A* in solid human pediatric cancers were plotted according to Immune Paediatric Signature Score (IPASS) status (cold vs. T cell-infiltrated) as previously published[85].

## R2 Kaplan–Meier analysis

A survival analysis of neuroblastoma patients was performed using the publicly available R2: Genomics Analysis and Visualization Platform (http://r2.amc.nl), screened by *ATP7A* gene expression. Overall survival data were obtained from the Tumor Neuroblastoma Bell ($n = 95$ patients) dataset, generated by way of bulk RNA sequencing.

## VetScan blood chemistry analysis

Peripheral blood was obtained from animals and collected in lithium heparin tubes (Greiner, Germany; #450536) and immediately analyzed using the VetScan VS2 Chemistry Analyzer (Zoetis, USA) using the Comprehensive Diagnostic Profile Rotor (Zoetis, USA; #500-0038) according to the manufacturer's instructions.

## Statistical analysis

In vivo and in vitro data visualization and statistical analyses were performed using GraphPad Prism v10 (Dotmatics, UK) with data presented as the means ± standard error of the mean (SEM). Differences between the two groups were determined with Mann–Whitney $U$ tests (unpaired or paired where specified), paired $t$-tests or unpaired $t$-tests with Welch's correction where specified. Kaplan–Meier survival curves were analyzed with two-tailed Mantel–Cox log-rank tests. An ordinary one-way ANOVA with Tukey's post-hoc test was used identify differences between three treatment groups. A two-way ANOVA with Sidak's multiple comparisons test was used to identify differences between two models with two treatments. A $p$ value < 0.05 was considered statistically significant for all experiments and is noted within figures and the main text where relevant.

## Reporting summary

Further information on research design is available in the Nature Portfolio Reporting Summary linked to this article.

# Data availability

Raw and processed single-cell RNA sequencing data used in this study are available in the NCBI Gene Expression Omnibus (GEO) database repository with the dataset identifier GSE281843. Raw and processed ng GeoMx Digital Spatial Profiling data used in this study are available in the NCBI Gene Expression Omnibus (GEO) database repository with the dataset identifier GSE281844. The remaining data are available within the Article, Supplementary Information or from the corresponding author upon request. Source Data are provided with this paper.

# Code availability

The R scripts generated during this study are available at the following GitHub repository: https://github.com/antosalerno/TEPA_code_v2.0.

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

## Acknowledgements

We acknowledge the Ramaciotti Centre for Genomics and the Mark Wainwright Analytic Facility (both at the University of New South Wales) for their expert technical support. We thank Stephanie Alfred, Erin Mosmann, and Prakruti Sirigiri for their assistance with the *Th-MYCN* preclinical model. This work was supported by grants to: O.V. from Cure Cancer Australia (CCAF2023); the National Health and Medical Research Council (GNT2012567); O.V. and J.C. from iCare Dust Diseases Board Grants Program (ICARE2023); O.V. and J.C. from Cancer Council Western Australia (CCWA1343).

## Author contributions

Conceptualization: Jourdin R.C. Rouaen, Orazio Vittorio, Toby N. Trahair. Methodology: Jourdin R.C. Rouaen, Antonietta Salerno, Tyler Shai-Hee, Jayne E. Murray, Toni-Rose Jue, Jessica L. Bell, Emanuele Valli, Riccardo Cazzoli, Delia J. Nelson, Jonathan Chee, Iveta Slapetova, Maria Kasherman, Renee Whan, Blake J. Cochran, Nicodemus Tedla, Feyza Colakoglu Veli, Aysen Yuksel, Federica Saletta, Holger N. Lode, Michelle Haber. Validation and investigation: Jourdin R.C. Rouaen, Antonietta

Salerno, Tyler Shai-Hee, Giulia Castrogiovanni, Charlotte McHenry, Jessica L. Bell, Ensieh Poursani, Naomi Damstra, Kofi L.P. Stevens, Jonathan Chee, Francis Lin, Blake J. Cochran, Nicodemus Tedla, Feyza Colakoglu Veli, Chelsea Mayoh. Formal analysis: Jourdin R.C. Rouaen, Antonietta Salerno, Tyler Shai-Hee, Jayne E. Murray, Charlotte McHenry, Vu Pham, Naomi Damstra, Kofi L.P. Stevens, Jonathan Chee, Francis Lin, Feyza Colakoglu Veli, Daniele Mercatelli. Data curation: Jourdin R.C. Rouaen, Orazio Vittorio. Writing, original draft: Jourdin R.C. Rouaen, Orazio Vittorio, Antonietta Salerno. Writing, review, and editing: Jourdin R.C. Rouaen, Orazio Vittorio, Antonietta Salerno, Juliet C. Gray, Toby N. Trahair, Jayne E. Murray, Jessica L. Bell, Blake J. Cochran, Chelsea Mayoh, Tatyana Chtanova, Nicodemus Tedla, Maté Biro. Supervision: Jourdin R.C. Rouaen, Orazio Vittorio, Toby N. Trahair, Jayne E. Murray, Daniele Mercatelli, Delia J. Nelson, Tatyana Chtanova, Arutha Kulasinghe, Daniel Catchpoole, Giuseppe Cirillo, Maté Biro, Fabio Luciani, Juliet C. Gray.

## Competing interests

The authors declare no competing interests.

## Additional information

[1]School of Biomedical Sciences, UNSW Medicine & Health, UNSW Sydney, Sydney, NSW, Australia. [2]Children's Cancer Institute, Lowy Cancer Research Centre, UNSW Sydney, Sydney, NSW, Australia. [3]Department of Experimental Oncology, IEO, European Institute of Oncology IRCCS, Milano, Italy. [4]Institute for Respiratory Health, National Centre for Asbestos Related Diseases, University of Western Australia, Perth, WA, Australia. [5]Curtin Medical School, Curtin Health Innovation Research Institute, Faculty of Health Sciences, Curtin University, Bentley, WA, Australia. [6]Curtin Health Innovation Research Institute, Bentley, WA, Australia. [7]Katharina Gaus Light Microscopy Facility, University of New South Wales, Sydney, NSW, Australia. [8]School of Biotechnology and Biomolecular Sciences, Faculty of Science, UNSW Sydney, Sydney, NSW, Australia. [9]EMBL Australia, Single Molecule Science Node, School of Biomedical Sciences, UNSW Sydney, Sydney, NSW, Australia. [10]Tumour Bank, Children's Hospital at Westmead, Westmead, NSW, Australia. [11]Department of Pharmacy and Biotechnology, University of Bologna, Bologna, Italy. [12]Garvan Institute of Medical Research, Sydney, NSW, Australia. [13]Frazer Institute, University of Queensland, Brisbane, QLD, Australia. [14]Department of Pharmacy Health and Nutritional Science, University of Calabria, Rende, Italy. [15]Department of Pediatric Hematology-Oncology, University Medicine Greifswald, Greifswald, Germany. [16]Kirby Institute for Infection and Immunity, Faculty of Medicine and Health, UNSW Sydney, Sydney, NSW, Australia. [17]Centre for Cancer Immunology, University of Southampton, Southampton, UK. [18]Kids Cancer Centre, Sydney Children's Hospital, Sydney, NSW, Australia. ✉e-mail: orazio.vittorio@unsw.edu.au

