## [Transparent Peer Review file · Nature Communications]

Copper chelation redirects neutrophil function to enhance anti-GD2 antibody therapy in neuroblastoma

Corresponding Author: Professor Orazio Vittorio

Version 0:

Reviewer comments:

Reviewer #1

(Remarks to the Author)

Rouaen et al. present evidence that adjuvant copper chelation using trientine analogs potentiates efficacy of anti-GD2 antibody therapy in immunocompetent murine models of neuroblastoma through altered infiltration of lymphoid and myeloid cells. Specific focus is placed on the N1 (anti-tumor) neutrophil response, which is an understudied and likely critical aspect in the efficacy of anti-GD2 antibody therapy in patients. There is compelling evidence worthy of publication, and the overall project is highly translatable. The presentation of the manuscript is good, with a logical and well-written text. Some points for revision include:

Figure 1D, 2C – Images are somewhat blurry in my version; consider improving quality of Multiplex OPAL IHC images.

Figure 1D, E - It would be nice to see the TEPA + IgG control for these studies. While these were separately studied in Figure 2 in a 7-day treatment experiment without the IgG, it is important to see them in context with the other treatments in the same experiment.

FIGURE 1B, the authors should comment here or in the discussion why they do not see any anti-tumor/survival effect of the TEPA +IgG control, given their Cancer Research article shows a statistically significant improvement in survival in vivo comparing control to TEPA alone in same mouse model. Do they think the control IgG had some negative effect on TEPA treatment?

Figure 1E- In the authors' previous Cancer Research article, TEPA treatment significantly increased CD8 T cell tumor infiltration but this is not recapitulated in the current model. Is there a reason for this inconsistency? Difference here is the inclusion of anti-GD2 antibody, which has a strong mono-agent effect on CD8 infiltration. See further comment below.

In the text (lines 140-141), the authors state that levels of TGF-B are reduced in TEPA-treated tumors. However, the stats in Figure 2B do not support that this absolutely occurs and the reduction is marginal compared to the increases observed in other cytokines. It might be better to exclude this from the results text or modify to sound less conclusive.

In Figure 2C, TEPA treatment now significantly increases CD8 T cell tumor infiltration but this was not the case in Figure 1D,E. This result better reflects the authors' previous work. One explanation is that the anti-GD2 antibody caused a dramatic increase on its own that cannot be furthered by TEPA? The authors should draw this dissimilarity out in the text regarding Figure 2C-D. Other than this, panels 2C and 2D are somewhat redundant with 1D and 1E (and previously published work). It might be advantageous to just show the TEPA only controls in Figure 1. Based on the arms of the in vivo study presented, a TEPA and IgG treatment arm was included. Are samples available for IF?

Figure 3A, Smooth Muscle Actin appears greatly enhanced in TEPA-treated tumors compared to control. The authors do not touch on this marker in the text related to the figure. Is this change expected in the model (does it represent vasculature?) and how does it relate to the other findings in that figure?

Figure 3B, why are days 3 & 7 combined for control tumors but separated for treated tumors? Should the reader assume

tumors were identical at these points or was data combined for control purposes? Also, in Figure 3B, the True Infiltration signal seems robust even in a control tumor but in Figure 3A, almost no CD45 staining is observed – is that due to photo quality and/or magnification or sampling differences? Can the authors provide a subfigure with increased magnification in an area of infiltration for both control and treated tumors so that the results for Figure 3A/B are better aligned?

Figure 3C. The murine hemoglobin genes are the most highly upregulated in the + infiltrated tumor sections, along with smooth muscle actin (which is in keeping with figure 3A IF). Is there increased vascularity in those TEPA treated “infiltrated” cores that may help explain the increased WBC (as well as RBCs/Hb genes) in those sites? Was bleeding into the tissue noted by IHC at any of those “infiltrated” sites (to suggest necrosis or breakdown of the vascularity where WBC more easily “seep” into the tissues following TEPA treatment vs recruitment?). Or does smooth muscle actin represent something else?

Figure 4E, would these findings not refute the IF data from Figures 2 (and prior Ca Res work) where TEPA treatment increased tumor recruitment of CD8 T cells and NK cells? Do the authors have any explanation for why TEPA did not alter CD8/NK cell recruitment in sequencing data of tumors? It is also interesting that TEPA treatment results in a reduced CD4 T cell compartment (Figure 4E, F) but this is not addressed by authors. Please also compare Figure 4 results to Figure 1 TEPA + IgG alone arm (once you add it) and TEPA/Anti-GD2 results (which, the combination, is similar to antiGD2 alone in increasing CD8/NK recruitment).

Figure 6D, not sure of model-specific levels but this percentage seems low for circulating neutrophils. Can the authors provide details on the blood processing and/or show percentages of other immune cells as a supplemental component?

Could the authors use the human derived NB in vitro models in Figure 6 to provide any human validation of the murine modeling? Some considerations... Does TEPA treatment enhance migration of human neutrophils towards neuroblastoma cells using a migration assay? Do human cells mimic the cytokine secretion observed in Figure 2B following TEPA treatment?

In lines 299-323, the authors allude that neutrophil uptake of copper following neuroblastoma TEPA treatment should enhance neutrophil functionality. To support this, do human neutrophils demonstrate a more N1-like phenotype and/or ADCC potential following co-culture with TEPA treated human neuroblastoma cells?

The overall meaning of the sentence in lines 321-323 is not clear to me. Also, could the authors speak more to the secreted soluble factors that are mentioned?

For Figure 7B, should the Y-axis label be CD45+ leukocytes? I don't think the authors are looking for CD11b/Ly6G in the lymphocyte population as neutrophils are of myeloid lineage.

Visually the infiltration of NK/CD8/myeloid cells in 1D, 2C and 7i does not appear drastically different (albeit images are blurry) but the axes are drastically different in scale for subsequent quantification. The models are different so this cannot be compared head-to-head but the quantitation seems inconsistent between figures.

In discussion, the authors should consider including clinical literature for trientine in pediatric patients as studies are available and document potential adverse events. It is likely that toxicity can be controlled through proper dosing. (Mayr et al J Pediat Gastroenterol Nutr 2021)

Neutrophils are the predominant white blood cell subset in humans (~70% of circulating leukocytes) but are less common in mice (~30%). There are also several known phenotypic and functional differences between human and murine neutrophils (Reviewed by Eruslanov et al, Trends in Cancer 2017, i.e., human granulocytes do not express Ly-6G, differences in FC receptor expression and granule content). This should be addressed and/or highlighted as a potential limitation of the study. In addition to studies proposed above, do the authors have any more results that highlight their findings are recapitulated in human tissues?

Reviewer #2

(Remarks to the Author)

Summary:

The authors present a novel therapeutic combination of a copper chelating agent (TETA or TEPA) together with anti-GD2 for treatment of neuroblastoma. Using preclinical modeling, they present high-throughput testing using spatial profiling (GeoMx), single cell RNA sequencing via BD Rhapsody, and OPAL multiplex IHC. While they show some in vivo response testing, more information regarding the findings from this data is needed. In addition, even though the high-throughput testing is valuable, simplified techniques as in vitro killing and flow cytometry to validate the high-throughput findings is necessary. Their data is compelling, but could be improved upon with revisions with the provision of a few additional details (with info that they likely already have) as detailed below:

Minor Requests:

-Fig 3: More detail needs to be included regarding the method and results from this intriguing data. It's unclear what the ROIs were. For C-E, are those data from 3 or 7 days post treatment. Can you comment on the increased myogenesis and hypoxia seen in Fig E? Also, is increased angiogenesis helpful? Traditionally, wouldn't angiogenesis be associated with metastatic

potential for the tumor and tumor growth?

-Fig 6b: the authors claim that tumor-infiltrating neutrophils exhibited a pro-inflammatory N1 phenotype independent of copper chelation therapy...while there are some differences in gene expression in control vs. TEPA treated samples (Ifitm6, Hif1a, Isg25, Stat3), there aren't appreciable differences in "N2" gene expression signatures in control vs. TEPA treated samples. Can the authors provide a reason for the lack of changes in gene expression observed in the N2 phenotype genes identified?

-While grouping TINs into "N1" and "N2" categories might have been more acceptable in the neutrophil field in the past, recent neutrophil scRNA-seq studies published in the last two years have defined neutrophil heterogeneity more granularly (similar to M1/M2 descriptions). There may be even more differences between control vs. TEPA treated samples at various neutrophil subset levels if the authors expanded their neutrophil subclassifications beyond just "N1" and "N2". Furthermore, it appears as though neutrophils make up a large amount of the total tumor (Fig 4e and 4f)--was the neutrophil cluster subsetted out for downstream analysis?

-Is Figure 4f normalized to tumor weight? The figure caption points out that 4f is showing cell proportions; however, it is showing cell counts. What size (area/volume/weight) were the tumors at the time of harvest for scRNAseq?

-The authors stated in the methods section that treatment with saline or TEPA was started when mouse tumors were 3-4 mm³. This is a tiny tumor, even for neuroblastoma. It would be helpful to also provide tumor growth curves, as well as response data, for Figure 1.

-NXS2 is a notoriously "easy-to-cure" neuroblastoma model when grown in A/J mice. To my knowledge, the NXS2 model is not considered a model of high-risk neuroblastoma, and patients without high-risk disease often respond favorably to treatment (whereas those with high-risk disease have disease that is more difficult to treat.

While Figure 7a does indeed show that TETA treatment slows tumor growth, it doesn't appear to demonstrate (if observed) tumor clearance. Fig 7a shows 4 mice/group, which is not a lot of power. Have the authors repeated that study, or was that just a study of 1 repeat? Can the authors provide response data for that study (tumor free vs tumor bearing)?

Fig 7f shows that mice treated with TETA+anti-GD2 become tumor free around D32, but Fig 7g shows mice dying with that same treatment starting at D24. Could it be that most mice die, and only a handful of mice (<50%) become tumor free? Again, it would be helpful to show individual mouse curves here and response data. If less than 50% of mice become tumor free in this "easy-to-cure" NBL model system, is this worth pursuing in the SIOPEN-BEACON2 trial as the current standard of care therapy for patients with the difficult to treat High-Risk Neuroblastoma have a response rate >60%?

-In the discussion, on lines 461, the authors state that they reported on macrophages, DCs, granulocytes, etc. exhibit enhanced activation (etc.). Please mention here that this is Sup. Fig. 1, and that this was done in silico as no data in vitro or in vivo supports these findings.

-In lines 485 to 486 of the Discussion, the authors overstate that neutrophils are the "most" potent effector population of anti-GD2 therapy, here referring to a paper from 1991 and 2000. The paper from 2000, in fact, shows that while neutrophils can effectors with anti-GD2 therapy (at the right E:T ratios), they can also encourage tumor growth. I don't know that either of these papers compared neutrophils to NK cells, but such study would be needed to make that statement that neutrophils are the "most" potent effector as NK cells are incredibly potent effectors of anti-GD2 treatment in vitro. This statement should be softened quite a bit, or even removed.

Major Requests:

-The authors make the claim that "mobilization of copper enhances anti-tumor activity of neutrophils"; the evidence included in this manuscript to support this statement include increased neutrophil counts in tumors of TEPA treated mice and GSEA analysis showing "N1 anti-tumor neutrophils" in TEPA treated mice. The addition of in vitro killing assays would bolster the claim that the neutrophils are functionally "anti-tumoral" in addition to exhibiting gene signatures that suggest they are.

-The authors use high-tech approaches to identify neutrophil infiltration with TETA + anti-GD2. Oftentimes, neutrophil infiltration is "assumed" via the presence of CD11b+ myeloid cells. Have the authors verified that these in fact are neutrophils via flow cytometry in either model system? CD11b can be on NK cells too (besides its expression on a number of myeloid populations, including macrophages). In the TH-MYCN model, macrophages have been shown to one of the major immune cell subsets present within tumors that develop in these mice as shown by Costa et al. (which the authors acknowledge may be due to the analytical technique and tumor stage). Besides spontaneously developing NBL in TH-MYCN mice, it's been shown that syngeneic mice bearing TH-MYCN tumors, 9464D, show a strong role for both macrophages and NK cells in the antitumor response. Yet, the authors show in Fig 4 that the major population, even in untreated mice, is neutrophils and that the macrophage population is minimal compared to neutrophils. Especially since this differs from what has been reported previously (Costa et al.), it's crucial to validate the findings observed in Fig 4 with flow cytometry.

Furthermore, as the authors point out, it's possible that dependent upon the timing of the treatment, neutrophil-driven responses may seem more apparent than other immune cells (like NKs), but in the long term the neutrophils may not be, in fact, the major antitumor effector. The authors assess some of their neutrophil infiltrate at early time points (D7 for eg.). It would be good to confirm the actual number of neutrophils at this time point by flow, as mentioned above, but also to see if they persist at later time points.

-To determine if neutrophils truly are the key mediators of the antitumor response, and not NK cells, it would help to show this by depletion studies during treatment. While neutrophils may be hard to deplete, NK cells are not. It would strengthen the role for neutrophil-mediated response if the authors deplete NK cells in the NXS2/A/J model while giving full treatment. If mice respond, even in the face of NK cell (and possibly also CD8 T cell) depletion, this would strengthen the role that neutrophils may be playing.

-It might be worthwhile to pursue this treatment in a separate model of high-risk disease (such as 9464D syngeneic to C57Bl/6 mice) to confirm efficacy, and justify pursuit of clinical trial testing in SIOPEN-BEACON2. This data is too premature to make that jump.

Reviewer #3

(Remarks to the Author)

Comments:

The authors present a novel therapeutic combination of a copper chelating agent (TETA or TEPA) together with anti-GD2 for treatment of neuroblastoma. Using preclinical modeling, they present high-throughput testing using spatial profiling (GeoMx), single cell RNA sequencing via BD Rhapsody, and OPAL multiplex IHC. While they show some in vivo response testing, more information regarding the findings from this data is needed. In addition, even though the high-throughput testing is valuable, simplified techniques such as in vitro killing and flow cytometry to validate the high-throughput findings are necessary. Their data is compelling, but could be improved upon with revisions with the provision of a few additional details (with info that they likely already have) as detailed below:

Minor Requests:

Fig 3: More detail needs to be included regarding the method and results from this intriguing data. It's unclear what the ROIs were. For C-E, are those data from 3 or 7 days post treatment. Can you comment on the increased myogenesis and hypoxia seen in Fig E? Also, is increased angiogenesis helpful? Traditionally, wouldn't angiogenesis be associated with metastatic potential for the tumor and tumor growth?

Fig 6b: the authors claim that tumor-infiltrating neutrophils exhibited a pro-inflammatory N1 phenotype independent of copper chelation therapy...while there are some differences in gene expression in control vs. TEPA treated samples (Ifitm6, Hif1a, Isg25, Stat3), there aren't appreciable differences in "N2" gene expression signatures in control vs. TEPA treated samples. Can the authors provide a reason for the lack of changes in gene expression observed in the N2 phenotype genes identified?

While grouping TINs into "N1" and "N2" categories might have been more acceptable in the neutrophil field in the past, recent neutrophil scRNA-seq studies published in the last two years have defined neutrophil heterogeneity more granularly (similar to M1/M2 descriptions). There may be even more differences between control vs. TEPA treated samples at various neutrophil subset levels if the authors expanded their neutrophil subclassifications beyond just "N1" and "N2". Furthermore, it appears as though neutrophils make up a large amount of the total tumor (Fig 4e and 4f)--was the neutrophil cluster subsetted out for downstream analysis?

Is Figure 4f normalized to tumor weight? The figure caption points out that 4f is showing cell proportions; however, it is showing cell counts. What size (area/volume/weight) were the tumors at the time of harvest for scRNAseq?

The authors stated in the methods section that treatment with saline or TEPA was started when mouse tumors were 3-4 mm³. This is a tiny tumor, even for neuroblastoma. It would be helpful to also provide tumor growth curves, as well as response data, for Figure 1.

NXS2 is a notoriously "easy-to-cure" neuroblastoma model when grown in A/J mice. To my knowledge, the NXS2 model is not considered a model of high-risk neuroblastoma, and patients without high-risk disease often respond favorably to treatment (whereas those with high-risk disease have disease that is more difficult to treat).

While Figure 7a does indeed show that TETA treatment slows tumor growth, it doesn't appear to demonstrate (if observed) tumor clearance. Fig 7a shows 4 mice/group, which is not a lot of power. Have the authors repeated that study, or was that just a study of 1 repeat? Can the authors provide response data for that study (tumor free vs tumor bearing)?

Fig 7f shows that mice treated with TETA+anti-GD2 become tumor free around D32, but Fig 7g shows mice dying with that same treatment starting at D24. Could it be that most mice die, and only a handful of mice (<50%) become tumor free?

Again, it would be helpful to show individual mouse curves here and response data. If less than 50% of mice become tumor free in this "easy-to-cure" NBL model system, is this worth pursuing in the SIOPEN-BEACON2 trial as the current standard of care therapy for patients with the difficult to treat High-Risk Neuroblastoma have a response rate >60%?

In the discussion, on lines 461, the authors state that they reported on macrophages, DCs, granulocytes, etc. exhibit enhanced activation (etc.). Please mention here that this is Sup. Fig. 1, and that this was done in silico as no data in vitro or in vivo supports these findings.

In lines 485 to 486 of the Discussion, the authors overstate that neutrophils are the "most" potent effector population of anti-GD2 therapy, here referring to a paper from 1991 and 2000. The paper from 2000, in fact, shows that while neutrophils can effectors with anti-GD2 therapy (at the right E:T ratios), they can also encourage tumor growth. I don't know that either of these papers compared neutrophils to NK cells, but such study would be needed to make that statement that neutrophils are the "most" potent effector as NK cells are incredibly potent effectors of anti-GD2 treatment in vitro. This statement should be softened quite a bit, or even removed.

Major Requests:

The authors make the claim that “mobilization of copper enhances anti-tumor activity of neutrophils”; the evidence included in this manuscript to support this statement include increased neutrophil counts in tumors of TEPA treated mice and GSEA analysis showing “N1 anti-tumor neutrophils” in TEPA treated mice. The addition of in vitro killing assays would bolster the claim that the neutrophils are functionally “anti-tumoral” in addition to exhibiting gene signatures that suggest they are. The authors use high-tech approaches to identify neutrophil infiltration with TETA + anti-GD2. Oftentimes, neutrophil infiltration is “assumed” via the presence of CD11b+ myeloid cells. Have the authors verified that these in fact are neutrophils via flow cytometry in either model system? CD11b can be on NK cells too (besides its expression on a number of myeloid populations, including macrophages). In the TH-MYCN model, macrophages have been shown to one of the major immune cell subsets present within tumors that develop in these mice as shown by Costa et al. (which the authors acknowledge may be due to the analytical technique and tumor stage). Besides spontaneously developing NBL in TH-MYCN mice, we have found in syngeneic mice bearing TH-MYCN tumors, 9464D, that both macrophages and NK cells play major roles in the antitumor response. Yet, the authors show in Fig 4 that the major population, even in untreated mice, is neutrophils and that the macrophage population is minimal compared to neutrophils. Especially since this differs from what has been reported previously (Costa et al.), it’s crucial to validate the findings observed in Fig 4 with flow cytometry. Furthermore, as the authors point out, it’s possible that dependent upon the timing of the treatment, neutrophil-driven responses may seem more apparent than other immune cells (like NKs), but in the long term the neutrophils may not be, in fact, the major antitumor effector. The authors assess some of their neutrophil infiltrate at early time points (D7 for eg.). It would be good to confirm the actual number of neutrophils at this time point by flow, as mentioned above, but also to see if they persist at later time points.

To determine if neutrophils truly are the key mediators of the antitumor response, and not NK cells, it would help to show this by depletion studies during treatment. While neutrophils may be hard to deplete, NK cells are not. It would strengthen the role for neutrophil-mediated response if the authors deplete NK cells in the NXS2/A/J model while giving full treatment. If mice respond, even in the face of NK cell (and possibly also CD8 T cell) depletion, this would strengthen the role that neutrophils may be playing.

It might be worthwhile to pursue this treatment in a separate model of high-risk disease (such as 9464D syngeneic to C57Bl/6 mice) to confirm efficacy, and justify pursuit of clinical trial testing in SIOPEN-BEACON2. This data is too premature to make that jump.

Reviewer #4

(Remarks to the Author)

Title: Copper chelation redirects neutrophil function to enhance anti-GD2 antibody therapy in neuroblastoma

Manuscript Type: Research Paper

Manuscript ID: 488552

Author: Jourdin R.C. Rouaen et al.

This MS aims at determining whether the combination of a copper chelator (TEPA or TETA) can potentiate the anti-tumor activity of anti-GD2 antibody (Ab) therapy in neuroblastoma (NB). The hypothesis of a possible improved efficacy of a treatment regimen, combining TEPA with anti-GD2, comes from their previously published results. In those cases, they demonstrated that copper intra-tumoral levels regulate the expression of PD-L1 and copper depletion, by means of a copper chelator, reduces the expression of PD-L1 in NB cells and definitively results in increased infiltration of cytotoxic T lymphocytes and NK cells.

In the present work, AAs claim to demonstrate that the copper chelation enhances anti-GD2 Ab efficacy through increasing neutrophils infiltration in NB. The topic of the study is interesting, also in view of future translational application, since the drug repurposing of TETA that demonstrates to be safe and easily administrable per OS, seems to be rationale e feasible. The study has been performed using highly advanced technology and overall is well performed and executed.

Nevertheless, some major concerns need to be addressed before the MS can be accepted for publication in a so prestigious journal like Nature Communications.

Major points

- 1) Why did the AAs decide to use different schedules of treatment for the two mouse models (Th-MYCN GEMM vs NXS2) used? In the first case, animal are subjected to a long-lasting treatment until the ethical end-point is reached, while in the second, treatment is interrupted at day 42.
- 2) Why did the AAs decide to use different drugs (TEPA and TETA) for the two mouse models, even if they are analogs?
- 3) Figure 1 d and e lacks of an experimental control. The evaluation of immune cells infiltration in tumor mass of mice treated with TEPA + IgG needs to be included. Moreover, we are surprised of the total ineffectiveness of TEPA treatment in the Th-MYCN GEMM model, compared to the NXS2 one (Fig 7). Since TEPA determines a tumor microenvironment shift from immune suppressive to immune permissive, and due to the fact that treatment lasts until the ethical end-point, we would expect to see at least a slight effectiveness. How can the AAs explain this difference between the models used?
- 4) Most of the evidences brought by the AAs are based on gene-set and transcriptomics analyses, which confirm the hypothesis of the AAs. However, in our opinion, a mechanistic demonstration of the real role of the infiltrated neutrophils is necessary. The AAs advise that recruited and activated neutrophils are responsible for the improved efficacy of the combination therapy, but a real demonstration is lacking.
- 5) Further, did the AAs think about the possibility that an ADCP mechanism can be also involved? Please, address this point.
- 6) We also think that the cytofluorimetric analyses on peripheral blood, bone marrow and tumor masses collected from control and treated mice are mandatory and they need to be showed, to evaluate immune cell subpopulations, with particular attention for myeloid derived suppressor cell markers, which in part overlap with neutrophil and macrophages ones.

Minor points

- 1) Line 93: this sentence looks redundant, since it has been already reported in the introduction section.
- 2) Line 361: the sentence "Similarly to TEPA, TETA slowed tumor growth....." is not correct because, unless some AAs mistakes reported in Figure 1, TEPA is completely ineffective in the transgenic model.

Version 1:

Reviewer comments:

Reviewer #1

(Remarks to the Author)

Reviewer #1 anti-GD2 in cancer (Remarks to the Author):

Overall, the authors have addressed the majority of my concerns with the original manuscript with additional experiments and explanations. Please see below my comments to my revision requests (authors' detailed answers are taken out to abbreviate this review) with a few remaining points noted by [*] that still need to be addressed:

1. Figure 1D, 2C – Images are somewhat blurry in my version; consider improving quality of Multiplex OPAL IHC images.

- Addressed

2. Figure 1D, E - It would be nice to see the TEPA + IgG control for these studies. While these were separately studied in Figure 2 in a 7-day treatment experiment without the IgG, it is important to see them in context with the other treatments in the same experiment.

- The IgG control was appropriately added. * OF NOTE: Figure 1D contains TETA instead of TEPA in labeling – is this correct? If so, why (as the rest of the figure is TEPA)?

3. FIGURE 1B, the authors should comment here or in the discussion why they do not see any anti-tumor/survival effect of the TEPA + IgG control, given their Cancer Research article shows a statistically significant improvement in survival in vivo comparing control to TEPA alone in same mouse model. Do they think the control IgG had some negative effect on TEPA treatment?

- Addressed. While it seems like a very small difference in tumor size, I can appreciate that this small size can make a large impact in the outcome of an in vivo study.

4. Figure 1E- In the authors' previous Cancer Research article, TEPA treatment significantly increased CD8 T cell tumor infiltration but this is not recapitulated in the current model. Is there a reason for this inconsistency? Difference here is the inclusion of anti-GD2 antibody, which has a strong mono-agent effect on CD8 infiltration. See further comment below.

- Comment thoroughly addressed.

5. In the text (lines 140-141), the authors state that levels of TGF- β are reduced in TEPA-treated tumors. However, the stats in Figure 2B do not support that this absolutely occurs and the reduction is marginal compared to the increases observed in other cytokines. It might be better to exclude this from the results text or modify to sound less conclusive.

- Comment addressed by additional experimentation. * OF NOTE: the treatment schemas presented in Figure 2A of original manuscript and Figure 2A of revised manuscript differ by 1 additional day of treatment (mouse schema at top of figures) so not sure that it was appropriate to combine the results for statistical purposes if indeed the treatment durations differed. Please clarify if this was error in the cartoons and if all animals were treated with same duration of treatment regimen or justify why it is ok to combine?

6. In Figure 2C, TEPA treatment now significantly increases CD8 T cell tumor infiltration but this was not the case in Figure 1D,E. This result better reflects the authors' previous work. One explanation is that the anti-GD2 antibody caused a dramatic increase on its own that cannot be furthered by TEPA? The authors should draw this dissimilarity out in the text regarding Figure 2C- D. Other than this, panels 2C and 2D are somewhat redundant with 1D and 1E (and previously published work). It might be advantageous to just show the TEPA only controls in Figure 1. Based on the arms of the in vivo study presented, a TEPA and IgG treatment arm was included. Are samples available for IF?

- Comment thoroughly addressed with additional data to support conclusions. redundant figures removed.

7. Figure 3A, Smooth Muscle Actin appears greatly enhanced in TEPA-treated tumors compared to control. The authors do not touch on this marker in the text related to the figure. Is this change expected in the model (does it represent vasculature?) and how does it relate to the other findings in that figure?

- * Partially addressed. I still think it would be useful to the manuscript reader to address the changes in SMA in the main

text. It is still not entirely clear what SMA is meant to represent in the TMA and why it was recommended by nano string as part of the standard GeoMx panel.

8. Figure 3B, why are days 3 & 7 combined for control tumors but separated for treated tumors? Should the reader assume tumors were identical at these points or was data combined for control purposes? Also, in Figure 3B, the True Infiltration signal seems robust even in a control tumor but in Figure 3A, almost no CD45 staining is observed – is that due to photo quality and/or magnification or sampling differences? Can the authors provide a subfigure with increased magnification in an area of infiltration for both control and treated tumors so that the results for Figure 3A/B are better aligned?

- Addressed.

9. Figure 3C. The murine hemoglobin genes are the most highly upregulated in the + infiltrated tumor sections, along with smooth muscle actin (which is in keeping with figure 3A IF). Is there increased vascularity in those TEPA treated “infiltrated” cores that may help explain the increased WBC (as well as RBCs/Hb genes) in those sites? Was bleeding into the tissue noted by IHC at any of those “infiltrated” sites (to suggest necrosis or breakdown of the vascularity where WBC more easily “seep” into the tissues following TEPA treatment vs recruitment?). Or does smooth muscle actin represent something else?

- Addressed. See above regarding SMA.

10. Figure 4E, would these findings not refute the IF data from Figures 2 (and prior Ca Res work) where TEPA treatment increased tumor recruitment of CD8 T cells and NK cells? Do the authors have any explanation for why TEPA did not alter CD8/NK cell recruitment in sequencing data of tumors? It is also interesting that TEPA treatment results in a reduced CD4 T cell compartment (Figure 4E, F) but this is not addressed by authors. Please also compare Figure 4 results to Figure 1 TEPA + IgG alone arm (once you add it) and TEPA/Anti-GD2 results (which, the combination, is similar to antiGD2 alone in increasing CD8/NK recruitment).

- Addressed. Authors acknowledged discrepancies in findings and presented logic for differences with experiments to address. Adjusted conclusion to focus on neutrophils due to caveats in experiments.

11. Figure 6D, not sure of model-specific levels but this percentage seems low for circulating neutrophils. Can the authors provide details on the blood processing and/or show percentages of other immune cells as a supplemental component?

- Addressed.

12. Could the authors use the human derived NB in vitro models in Figure 6 to provide any human validation of the murine modeling? Some considerations... Does TEPA treatment enhance migration of human neutrophils towards neuroblastoma cells using a migration assay? Do human cells mimic the cytokine secretion observed in Figure 2B following TEPA treatment?

* Would be ideal to have an additional biological (and technical replicates) here but the authors explicitly state the low n's in the legend. Maybe move to supplemental if not going to perform a more robust characterization.

b) We performed a migration assay which demonstrated increased migration towards BE2C cells when treated with TEPA, as shown in Fig. 6i.

* Y-axis of figure has a typo – should be transmigration.

c) We performed an ADCC killing assay which showed enhanced killing in the presence of TEPA/anti-GD2 antibody against GD2+/MYCN-amplified Kelly cells, as shown in Fig. 6j.

- Comment was addressed using human derived in vitro models with similar outcomes.

13. In lines 299-323, the authors allude that neutrophil uptake of copper following neuroblastoma TEPA treatment should enhance neutrophil functionality. To support this, do human neutrophils demonstrate a more N1-like phenotype and/or ADCC potential following co-culture with TEPA treated human neuroblastoma cells?

- Addressed. * On the experiment design, can the authors confirm that the increased caspase-3/7 signal was generated by dying tumor cells and not dying neutrophils. The results fit within the story as a whole, which is great, but was the experiment controlled by assaying each of the components separately since caspase-3/7 dye is not specific to tumors or neutrophils.

14. The overall meaning of the sentence in lines 321-323 is not clear to me. Also, could the authors speak more to the secreted soluble factors that are mentioned?

- Addressed with revised wording

15. For Figure 7B, should the Y-axis label be CD45+ leukocytes? I don't think the authors are looking for CD11b/Ly6G in the lymphocyte population as neutrophils are of myeloid lineage.

- Addressed

16. Visually the infiltration of NK/CD8/myeloid cells in 1D, 2C and 7i does not appear drastically different (albeit images are blurry) but the axes are drastically different in scale for subsequent quantification. The models are different so this cannot be compared head-to-head but the quantitation seems inconsistent between figures.

- Addressed.

17. In discussion, the authors should consider including clinical literature for trientine in pediatric patients as studies are available and document potential adverse events. It is likely that toxicity can be controlled through proper dosing. (Mayr et al J Pediat Gastroenterol Nutr 2021)

- Addressed.

18. Neutrophils are the predominant white blood cell subset in humans (~70% of circulating leukocytes) but are less common in mice (~30%). There are also several known phenotypic and functional differences between human and murine neutrophils (Reviewed by Eruslanov et al, Trends in Cancer 2017, i.e., human granulocytes do not express Ly-6G, differences in FC receptor expression and granule content). This should be addressed and/or highlighted as a potential limitation of the study. In addition to studies proposed above, do the authors have any more results that highlight their findings are recapitulated in human tissues?

Addressed with revisions.

** Minor additional comments on revised figures and additional unsolicited data provided by the authors **:

• Revised Figure 2C: * You state that the myeloid cells (monocytes/granulocytes) show "enhanced activation". How did you determine in your flow analysis that macrophages and granulocytes exhibited "enhanced activation"? What markers did you look at by flow in these cells to draw that conclusion?

• The authors wrote: "To gain further insight into the changes occurring in cytokine production, we included the analysis of their serum levels in mice to match the measurements taken from the neuroblastoma TME in the Th-MYCN model (Fig 2b). This approach provides deeper understanding of the relationship between local and systemic immune responses, and emphasizes the role of IL-8/KC in stimulating neutrophil mobilization and trafficking to the tumor site."

* Do you mean IL-6 (IL-8 is not shown in figure 2b, but IL-6 is elevated in TME of TEPA treated animals)? KC is more commonly known as CXCL1 and not IL-8 (IL-8 is also known as CXCL8, which is also not depicted in Fig 2b, so did you mean KC/CXCL1? Or KC/IL-6?). By identifying increased KC/CXCL1 and/or IL-6 in the serum/TME does not guarantee that they stimulated or aided in neutrophil mobilization and trafficking to the tumor in vivo for you did not show this to be the case functionally (by blocking it, etc). I would reword that statement to be a less definitive relationship unless you can perform functional experiments to support that cause/effect.

• The authors write: "For further clinical validation, we performed a Kaplan-Meier analysis of patients with neuroblastoma using the copper exporter ATP7A gene in the R2: Genomics Analysis and Visualization Platform where elevated ATP7A expression was associated with improved survival. This is updated in Line 377: "In clinical support, elevated expression of the copper export protein ATP7A (encoded by ATP7A) was significantly associated with a T cell-infiltrated tumor microenvironment in a wide range of solid human pediatric malignancies, and was also associated with the improved survival of patients with neuroblastoma (Supp. Fig. 7c,d)."

* Is the Kaplan Meier curve of ATP7A expression and survival from gene expression/outcomes data of high-risk neuroblastoma tumors only? This graph is less relevant if it is taking in ALL risk categories of neuroblastoma tumors - as only HR tumors are immune cold and treated with anti-GD2. What gene expression dataset in R2 did you use? and did you validate it with another tumor dataset that has outcomes data (there are many in that database)? Did you use the bonferoni correction of the P value from that analysis of multiple patient data given the regular p value is not accurate in this setting?

Reviewer #2

(Remarks to the Author)

Thank you for addressing many of the questions that were raised. I understand the logic for using the NXS2 tumor model from the consideration of the role of neutrophil infiltration.

While the authors have shown other articles which show the expression of GD2 on NXS2 cells, the expression of GD2 on NXS2 cells notoriously varies, and it is often lost over time. For that reason, it would be helpful for the authors to show the expression level (perhaps in a supplementary figure) of GD2 on their NXS2 cells to confirm the expression is good.

Thank you for including the individual mouse curves for Figure 7. Would it also be possible (either in the text after the treatment name, or in a separate sub-figure graph) to show the cure rate for Teta+anti-GD2 in that same figure. For example, after the in the text after the treatment, you could maybe include "Teta+anti-GD2 (3/8 TF); Teta+anti-IgG (0/8); etc."

Reviewer #3

(Remarks to the Author)

I believe the edits the authors have made address the concerns I conveyed during initial review of this manuscript.

Reviewer #4

(Remarks to the Author)

Title: Copper chelation redirects neutrophil function to enhance anti-GD2 antibody therapy in neuroblastoma

Manuscript Type: Research Paper

Manuscript ID: 488552_1

Author: Jourdin R.C. Rouaen et al.

The AAs addressed almost all points raised during the first revision, and the manuscript can be now considered acceptable for publication.

Minor

In Figure 1, as requested, AAs added experimental controls. In Figure 1d, TETA should be however substituted with TEPA.

We would like to express our sincere gratitude to the reviewers for their invaluable feedback and constructive comments on our manuscript. All their insightful suggestions and guidance have significantly enhanced the quality and clarity of our paper, ultimately strengthening the overall message we aim to deliver. Each of your inputs has been instrumental in improving the manuscript, and we are grateful for the opportunity to address your comments. The amendments have been highlighted in blue in the revised manuscript.

Please find here our point-by-point responses to the reviewers' comments:

REVIEWER COMMENTS

Reviewer #1 anti-GD2 in cancer (Remarks to the Author):

Rouaen et al. present evidence that adjuvant copper chelation using trientine analogs potentiates efficacy of anti-GD2 antibody therapy in immunocompetent murine models of neuroblastoma through altered infiltration of lymphoid and myeloid cells. Specific focus is placed on the N1 (anti-tumor) neutrophil response, which is an understudied and likely critical aspect in the efficacy of anti-GD2 antibody therapy in patients. There is compelling evidence worthy of publication, and the overall project is highly translatable. The presentation of the manuscript is good, with a logical and well-written text. Some points for revision include:

Figure 1D, 2C – Images are somewhat blurry in my version; consider improving quality of Multiplex OPAL IHC images.

R: Thanks for this observation. In the updated version of the paper, we provided the figures as an additional stitched PDF which now feature higher resolution images.

Figure 1D, E - It would be nice to see the TEPA + IgG control for these studies. While these were separately studied in Figure 2 in a 7-day treatment experiment without the IgG, it is important to see them in context with the other treatments in the same experiment.

R: Thanks for this suggestion. In the revised version of the paper, we have provided the TEPA + IgG control arm for this panel (See Fig 1d,e).

FIGURE 1B, the authors should comment here or in the discussion why they do not see any anti-tumor/survival effect of the TEPA +IgG control, given their Cancer Research article shows a statistically significant improvement in survival in vivo comparing control to TEPA alone in same mouse model. Do they think the control IgG had some negative effect on TEPA treatment?

R: Thanks for highlighting this important point. To study potential synergy between TEPA and anti-GD2 therapy we optimised drug concentrations and conditions where the single agent activity was minimal. In our 2020 Cancer Research paper we started treatment with TEPA when tumour was very small, ≤ 2 mm. Whereas in this current study we started the treatment with when the tumour was ≥ 3 mm (as shown in the new added Fig. 1B). This made TEPA (at

that concentration) less effective in reducing tumour growth, even if it increased immune infiltration. We added this comment in the material and methods of the revised paper in Line 671: This recruitment size was selected to obtain adequate tumor material to study the synergy between copper chelation and immunotherapy.

Figure 1E- In the authors' previous Cancer Research article, TEPA treatment significantly increased CD8 T cell tumor infiltration but this is not recapitulated in the current model. Is there a reason for this inconsistency? Difference here is the inclusion of anti-GD2 antibody, which has a strong mono-agent effect on CD8 infiltration. See further comment below.

R: We fully agree with this comment. We have included the TEPA + IgG arm in Fig 1E to show that TEPA as a monotherapy increases CD8+ T cell infiltration (alongside Fig 2c,d). This is in line with the results obtained for Cancer Res paper (PMID: 32816860). Our results, clarifies that both TEPA and anti-GD2 act well as monotherapies to induce CD8+ T cell infiltration. However, they do not synergise to further increase infiltration when combined. We added this comment in the results section in Line 122: Nonetheless, the addition of copper chelation to anti-GD2 therapy substantially enhanced the infiltration of NK ($p=0.006$) and myeloid ($p=0.001$) cell subsets however did not synergize to cause further infiltration of CD8+ cytotoxic T cells.

In the text (lines 140-141), the authors state that levels of TGF-B are reduced in TEPA-treated tumors. However, the stats in Figure 2B do not support that this absolutely occurs and the reduction is marginal compared to the increases observed in other cytokines. It might be better to exclude this from the results text or modify to sound less conclusive.

R: We have a previous publication highlighting the relationship between TGF- β expression and copper chelation across in vitro and in vivo models of cancer (PMID: 37480151), and we do agree that this data in neuroblastoma were not as strong. However, as requested by reviewers, we performed a flow cytometric characterization of the myeloid compartment after one week of treatment. These animals were also used as an opportunity to obtain sera and tumor material for further TGF- β characterisation. This increase in power now provides statistical significance (from $n=4$ to $n=7$ /group) to demonstrate reduced serum and TME expression. We have updated this graph to include new data and cite our previous work linking copper homeostasis and TGF- β expression. This is reflected in Line 153: Of note, copper chelation significantly reduced levels of the immunosuppressive cytokine transforming growth factor-beta (TGF- β) which aligns with our recent finding linking its expression to copper levels in a variety of cancer types, including neuroblastoma (18).

In Figure 2C, TEPA treatment now significantly increases CD8 T cell tumor infiltration but this was not the case in Figure 1D,E. This result better reflects the authors' previous work. One

explanation is that the anti-GD2 antibody caused a dramatic increase on its own that cannot be furthered by TEPA? The authors should draw this dissimilarity out in the text regarding Figure 2C-D. Other than this, panels 2C and 2D are somewhat redundant with 1D and 1E (and previously published work). It might be advantageous to just show the TEPA only controls in Figure 1. Based on the arms of the in vivo study presented, a TEPA and IgG treatment arm was included. Are samples available for IF?

R: We appreciated this comment. Following your suggestion, we added the results from the mice treated with TEPA+IgG in Fig 1. We observed that both TEPA and anti-GD2, as single agent, caused increased CD8+ T cells infiltration. However, when combined together we couldn't see increased CD8+ infiltration compared to the one induced by anti-GD2 alone. We hypothesized that anti-GD2 causes already a substantial increase in CD8+ T cell infiltration which cannot be furthered by TEPA. We have amended the text to mention this in Line 118 as above. We have removed the OPAL data from this figure to remove redundancy with Fig. 1d,e.

Figure 3A, Smooth Muscle Actin appears greatly enhanced in TEPA-treated tumors compared to control. The authors do not touch on this marker in the text related to the figure. Is this change expected in the model (does it represent vasculature?) and how does it relate to the other findings in that figure?

R: The smooth muscle marker was recommended by Nanostring as part of the standard staining GeoMx panel. We did not include this marker explicitly to draw experimental conclusions. However, after your comment, we have examined all the cores of the TMA and we concluded that there is an increased number of vasculatures in the tumours of the mice treated with TEPA. To provide more transparency in our ROI selection, we have included a summary and characterization of the ROIs as part of the supplementary methods (see Supp. Methods 2). We have also described this angiogenic activity in the results section in Line 209: "We observed increased angiogenic signaling along with significant upregulation of the hemoglobin family genes Hbb-b2, Hba-a1, and Hbb-b1 (Fig. 3c), alluding to increased tumoral vascularity which may facilitate intra-tumoral immune infiltration (Fig. 3d)."

Figure 3B, why are days 3 & 7 combined for control tumors but separated for treated tumors? Should the reader assume tumors were identical at these points or was data combined for control purposes? Also, in Figure 3B, the True Infiltration signal seems robust even in a control tumor but in Figure 3A, almost no CD45 staining is observed – is that due to photo quality and/or magnification or sampling differences? Can the authors provide a subfigure with increased magnification in an area of infiltration for both control and treated tumors so that the results for Figure 3A/B are better aligned?

R: Thanks for helping to clarify this important part. Following your suggestion, we have changed 3b to reflect the separate treatment groups and have also

switched from false/true to low/high binary assignment for more intuitive comprehension. We have provided a subfigure magnification showing infiltration (or lack thereof) in the representative images in 3a, to complement with the data presented in 3b. We have also provided a Supplementary Methods Figure 2 which outlines the summary of ROIs analyzed, annotated with treatment group and CD45-infiltration status.

Figure 3C. The murine hemoglobin genes are the most highly upregulated in the + infiltrated tumor sections, along with smooth muscle actin (which is in keeping with figure 3A IF). Is there increased vascularity in those TEPA treated “infiltrated” cores that may help explain the increased WBC (as well as RBCs/Hb genes) in those sites? Was bleeding into the tissue noted by IHC at any of those “infiltrated” sites (to suggest necrosis or breakdown of the vascularity where WBC more easily “seep” into the tissues following TEPA treatment vs recruitment?). Or does smooth muscle actin represent something else?

R: Thanks for this comment. In the revised paper we specified in the materials & methods that, prior to construction of the tissue microarray, our pathologist examined tumor samples to exclude areas featuring necrosis. As a result, only non-necrotic areas were cored to comprise the tissue microarray. This is reflected in Line 764: Coring sites were chosen at random however areas exhibiting necrosis were excluded prior by an experienced histopathologist using a hematoxylin-eosin stain. As previously noted, smooth muscle actin was recommended by the technical team at Nanostring as part of the GeoMx standard panel. We have acknowledged your note on vascularity, and we have added a comment about in-text with the following in Line 209: “We observed increased angiogenic signaling along with significant upregulation of the hemoglobin family genes Hbb-b2, Hba-a1, and Hbb-b1 (Fig. 3c), alluding to increased tumoral vascularity which may facilitate intra-tumoral immune infiltration (Fig. 3d).”. Moreover, we have highlighted in the text that in the ROIs of the tumors treated with TEPA, we observed a reduction in the HALLMARK_MYC_TARGETS_V1 and upregulation of the REACTOME_NEUTROPHIL_DEGRANULATION, INTERFERON_GAMMA_RESPONSE, and TNFa signalling. This data suggests that the infiltration promoted by TEPA was associated to an anti-cancer immune response (see Fig. 3e).

Figure 4E, would these findings not refute the IF data from Figures 2 (and prior Ca Res work) where TEPA treatment increased tumor recruitment of CD8 T cells and NK cells? Do the authors have any explanation for why TEPA did not alter CD8/NK cell recruitment in sequencing data of tumors? It is also interesting that TEPA treatment results in a reduced CD4 T cell compartment (Figure 4E, F) but this is not addressed by authors. Please also compare Figure 4 results to Figure 1 TEPA + IgG alone arm (once you add it) and TEPA/Anti-GD2 results (which, the combination, is similar to antiGD2 alone in increasing CD8/NK recruitment).

R: Thanks for asking to clarify those points. In Fig 4e we had identified a cluster called NKT-like cells expressing CD8. This could be identified only by

transcriptomics analysis and not by IF. Given your question, we looked in literature and we found a paper stating that mouse NKT-like cells are less likely to express CD8 marker (PMID: 21196373). Therefore, we looked again at our single cells transcriptomics data and decided that is more appropriate to renominate the NKT-like cluster as CD8 effector T cells. It is important to note that this CD8 effector T cells cluster, contains about the double number of cells in the TEPA treated tumours compared to control. This is consistent with the findings shown in Figs. 1 and 2 and in our previous publication in *Cancer Res*, where we stained for CD8. Moreover, during our review, it was noted by another reviewer that it was best practice to normalize immune cell counts to tumor weight. We have included in the revised version the data normalized to tumors weight in Supp. Fig. 2a.

Regarding the number of NK cells obtained by the single cells RNAseq data, we did not see an increase in number of NK cells (the NK amount resulted to be very similar in both the control and TEPA group). However, in our previous *Cancer Res* paper we saw an increased number of NK cells by IHC in the tumors treated with TEPA. This discrepancy might be attributed to the different size of the tumors when we started treatment with TEPA. In our *Cancer Res* paper we started the treatment with TEPA in the THMYCN mouse immediately after weaning with a tumor typically ≤ 2 mm in size (3 weeks of age, as it was reported in the Material and Methods of that paper); in the mice shown here in Fig. 1 we started the treatment with TEPA when tumors were ≥ 3 mm, and the IF was performed after **two weeks** of TEPA treatment to make a comparison between single and combination treatment, as we had one week of copper chelator plus one week when it was combined with the anti-GD2 treatment; on the other side, in our single cells transcriptomics analysis we started treatment when the tumors were ≥ 3 mm, as we did for Fig 1 experiment, but to get the early effect of TEPA (as single agent) and also enough tumor material we collected the samples after 1 week of TEPA treatment. The larger tumour size combined with shorter drug treatment might be the reason for not observing an increase in NK cells. To better understand this we did additional experiments during the review time and we ran flow cytometry from $n=3$ tumors treated with TEPA for one week as well as we did for the single-cells transcriptomics. Data showed that we have an increased trend of NK cells (not significant) after TEPA treatment (please refer to Supp. Fig. 2b), whereas we still see a significant increase of neutrophils. Those results actually clarify that the copper chelation could act inducing first the neutrophils response, with the NK increase infiltration could be happening as a secondary immune effect. This data were added in the suppl material and discussed in the results section in Line 255: "Of note, copper chelation resulted in increased numbers of CD8+ effector T cells yet revealed a marginal decrease in NK cells, contrasting with our previous report (12). Given animals commenced copper chelation therapy immediately after weaning (tumor burden ≤ 2 mm) in the former study, we posit that the recruitment at a larger tumor size herein (tumor burden ≥ 3 -4mm) may account for this discrepancy. To address this, we performed flow cytometry to determine the frequency of infiltrating NK cells in Th-MYCN tumors and observed a

marked but non-significant increase with TEPA treatment (Supp. Fig. 2b). Taken together, this suggests that NK infiltration may occur as a secondary effect during the destabilization of the immunosuppressive tumor microenvironment.... Nonetheless, neutrophils still exhibited a profound five-fold increase with treatment which is consistent with our flow cytometric analysis (see Fig. 4e,f)."

We also added a comment about the decreased levels of CD4 T cells observed in our single cell data (Fig. 4e,f) in Line 264: "We also observed a decrease in the infiltration of CD4+ naive T cells with TEPA, which may lead to lower levels of immunosuppressive regulatory T cells induced by exposure to TGF- β (31)."

Figure 6D, not sure of model-specific levels but this percentage seems low for circulating neutrophils. Can the authors provide details on the blood processing and/or show percentages of other immune cells as a supplemental component?

R: Thanks for raising this question. While humans feature a high percentage of neutrophils in their blood (50-70%), mice exhibit a lower neutrophil frequency (10-30%) (PMID: 14978070; PMID: 38065997) which our presented data is aligned with. We have updated our text to include a supplemental figure (Line 166; Supp. Fig. 1) reporting on other circulating cell counts (/ μ L blood) as well as frequencies (%). We acknowledge these are slightly higher than routine values however these are likely influenced by the tumor-bearing context. Other key variables include strain, age, and gender. Details on blood processing have been provided in the methods.

Could the authors use the human derived NB in vitro models in Figure 6 to provide any human validation of the murine modeling? Some considerations... Does TEPA treatment enhance migration of human neutrophils towards neuroblastoma cells using a migration assay? Do human cells mimic the cytokine secretion observed in Figure 2B following TEPA treatment?

R: We appreciated your suggestion and performed additional experiments using human cells to understand the state of neutrophils. To examine the influence of copper chelation TEPA on the function capacity of neutrophils, we performed characterization studies using human neutrophils and human neuroblastoma cell lines.

a) Using neutrophils from two different donors, we performed qRT-PCR analysis for the expression of genes in human neutrophils associated with intracellular copper (MT1X), migration (S100A8) and activation (ISG15) obtained after 30min incubation in conditioned media (media from human neuroblastoma cells incubated with TEPA for 24h). Data in Fig. 6h showed, increased levels of the metallothioneine MT1X which correlates with increased copper uptake, increased expression of the migration gene S100A8 and the increased inflammatory activity indicated by ISG15.

b) We performed a migration assay which demonstrated increased migration towards BE2C cells when treated with TEPA, as shown in Fig. 6i.

c) We performed an ADCC killing assay which showed enhanced killing in the presence of TEPA/anti-GD2 antibody against GD2+/MYCN-amplified Kelly cells, as shown in Fig. 6j.

In lines 299-323, the authors allude that neutrophil uptake of copper following neuroblastoma TEPA treatment should enhance neutrophil functionality. To support this, do human neutrophils demonstrate a more N1-like phenotype and/or ADCC potential following co-culture with TEPA treated human neuroblastoma cells?

R: The additional experiments presented in Fig. 6, explained in question 12 are confirming that TEPA treatment enhanced migration and activation of neutrophils, improving the ADCC when combined with anti-GD2 therapy (Fig. 6i).

The overall meaning of the sentence in lines 321-323 is not clear to me. Also, could the authors speak more to the secreted soluble factors that are mentioned?

R: We apologise for the poor phrasing and have updated this in-text (Line 374) to simplify the notion: "This phenomenon did not occur in untreated neuroblastoma cells, indicating that copper chelation therapy modulates levels of copper in the tumor microenvironment to facilitate copper uptake by neutrophils."

In this experiment we designed the co-culture using conditioned media i.e. neutrophil/neuroblastoma cells are not in direct contact with each other (as per traditional co-culture assays). This implies that soluble factors are present to facilitate copper uptake however the nature of these is beyond the scope of this project and is currently being investigated.

For Figure 7B, should the Y-axis label be CD45+ leukocytes? I don't think the authors are looking for CD11b/Ly6G in the lymphocyte population as neutrophils are of myeloid lineage.

R: Thanks for this comment. This was a typo that we fixed in the edited paper.

Visually the infiltration of NK/CD8/myeloid cells in 1D, 2C and 7i does not appear drastically different (albeit images are blurry) but the axes are drastically different in scale for subsequent quantification. The models are different so this cannot be compared head-to-head but the quantitation seems inconsistent between figures.

R: We appreciated this comment. We provided better quality pictures and are more representative of the overall count in the different samples which approximate 1000 nuclei in the provided field of view.

In discussion, the authors should consider including clinical literature for trientine in pediatric patients as studies are available and document potential adverse events. It is likely that toxicity can be controlled through proper dosing. (Mayr et al J Pediat Gastroenterol Nutr 2021)

R: Thanks for raising this important point. We have added in the discussion this reference (Line 579) to read “Clinical data reports that TETA is efficacious in pediatric patients, features a low occurrence of documented side effects (often reversible after dose reduction or discontinuation), and is available as an oral formulation for ease of administration (74,75).”

Neutrophils are the predominant white blood cell subset in humans (~70% of circulating leukocytes) but are less common in mice (~30%). There are also several known phenotypic and functional differences between human and murine neutrophils (Reviewed by Eruslanov et al, Trends in Cancer 2017, i.e., human granulocytes do not express Ly-6G, differences in FC receptor expression and granule content). This should be addressed and/or highlighted as a potential limitation of the study. In addition to studies proposed above, do the authors have any more results that highlight their findings are recapitulated in human tissues?

R: We fully agree with your comment. Following your suggestions, we have provided three in vitro functional studies using human neutrophils to understand the influence of copper chelation therapy with results corroborating our murine findings. We have acknowledged the important difference between the murine and human models with the following in-text sentence and two citations (Line 566):“Although there are recognized distinctions between murine and human neutrophils, we have demonstrated that copper chelation can similarly promote neutrophil recruitment and activation across both models to enhance anti-GD2 antibody therapy efficacy in neuroblastoma (69,70).” We also added this citation: Eruslanov et al, Trends in Cancer 2017; Nauseef, Immunological Reviews 2023.

Reviewer #2 neuroblastoma, innate immune cells (Remarks to the Author):

Summary:

The authors present a novel therapeutic combination of a copper chelating agent (TETA or TEPA) together with anti-GD2 for treatment of neuroblastoma. Using preclinical modeling, they present high-throughput testing using spatial profiling (GeoMx), single cell RNA sequencing via BD Rhapsody, and OPAL multiplex IHC. While they show some in vivo response testing, more information regarding the findings from this data is needed. In addition, even though the high-throughput testing is valuable, simplified techniques as in vitro killing and flow cytometry to validate the high-throughput findings is necessary. Their data is compelling, but could be improved upon with revisions with the provision of a few additional details (with info that they likely already have) as detailed below:

R: We appreciated your suggestions. We added more flow cytometry data, as shown in Fig. 2c, d, Supp Fig. 2b, Fig. 4e, f, and Fig. 7b. Moreover, we performed additional experiments using human cells to understand the state of neutrophils. To examine the influence of the copper chelation agent TEPA on the functional capacity of neutrophils, we conducted characterization studies using human neutrophils and human neuroblastoma cell lines.

a) Using neutrophils from two different donors, we performed qRT-PCR analysis to measure the expression of genes in human neutrophils associated with intracellular copper (MT1X), migration (S100A8), and activation (ISG15) after a 30-minute incubation in conditioned media (media from human neuroblastoma cells incubated with TEPA for 24 hours). The data in Fig. 6h showed increased levels of the metallothionein MT1X, correlating with increased copper uptake, elevated expression of the migration gene S100A8, and heightened inflammatory activity indicated by ISG15.

b) We performed a migration assay, which demonstrated increased migration towards BE2C cells when treated with TEPA, as shown in Fig. 6i.

c) We performed an ADCC killing assay, which showed enhanced killing in the presence of TEPA/anti-GD2 antibody against GD2+/MYCN-amplified Kelly cells, as shown in Fig. 6j.

Minor Requests:

-Fig 3: More detail needs to be included regarding the method and results from this intriguing data. It's unclear what the ROIs were. For C-E, are those data from 3 or 7 days post treatment. Can you comment on the increased myogenesis and hypoxia seen in Fig E? Also, is increased angiogenesis helpful? Traditionally, wouldn't angiogenesis be associated with metastatic potential for the tumor and tumor growth?

R: Thanks for highlighting this point. To improve the clarity of the figure, we have changed 3b to reflect the separate treatment groups and have also relabelled infiltration from false/true to low/high binary assignment for more intuitive comprehension. The ROIs selected for analysis were considered TEPA-treated samples (3 & 7 days) with differences displayed low and high-infiltrated ROIs (stated in Line 200).

The link between myogenesis and neuroblastoma is not immediately clear on consult of the literature. Nonetheless, we found a recently published paper in Scientific Reports (2023) which report pathway enrichment with considerable overlap with Fig. 3e. These are associated with features of a low-risk clinical neuroblastoma cohort. We therefore posit that copper chelation therapy attenuates disease aggression, reflected in-text (Line 215): Notably, this enrichment analysis displayed a remarkable overlap with a recently published dataset comparing enriched pathways in low- versus high-risk neuroblastoma clinical cohorts (24).

Hypoxia is generally associated with more aggressive tumor phenotype in neuroblastoma (PMID: 29032465) however is also known to promote neutrophil survival and degranulation in pro-inflammatory contexts (PMID: 36761171; PMID: 35044899, PMID: 32053993), which is also featured as an upregulated pathway in Fig. 3e. Moreover, we show an increase in Hif1a (hypoxic-inducible factor 1-alpha) expression in our infiltrating neutrophils (Fig. 6c), demonstrating that neutrophils are indeed sensing and responding to this hypoxic environment. We reflect this in-text (Line 212): “Interestingly, pathways were also enriched for hypoxia and neutrophil degranulation activity, demonstrating an interplay known to occur in inflamed environments and further emphasizing the role of this immune population in the tumor microenvironment (Fig. 3d,e) (23).” Increased vascularisation is also associated to an increased infiltration which corresponds to a good prognosis in this case, since it helps the neutrophils and other immune components to reach the site of inflammation (Please also see response to question 9 from reviewer 1).

-Fig 6b: the authors claim that tumor-infiltrating neutrophils exhibited a pro-inflammatory N1 phenotype independent of copper chelation therapy...while there are some differences in gene expression in control vs. TEPA treated samples (Ifitm6, Hif1a, Isg25, Stat3), there aren't appreciable differences in “N2” gene expression signatures in control vs. TEPA treated samples. Can the authors provide a reason for the lack of changes in gene expression observed in the N2 phenotype genes identified?

-While grouping TINs into “N1” and “N2” categories might have been more acceptable in the neutrophil field in the past, recent neutrophil scRNA-seq studies published in the last two years have defined neutrophil heterogeneity more granularly (similar to M1/M2 descriptions). There may be even more differences between control vs. TEPA treated samples at various neutrophil subset levels if the authors expanded their neutrophil subclassifications beyond just “N1” and “N2”. Furthermore, it appears as though neutrophils make up a large amount of the total tumor (Fig 4e and 4f)--was the neutrophil cluster subsetted out for downstream analysis?

R: Thanks for this question. We realized that the calculation of the z-score was performed across all the different immune cells populations. This did not allow us to appreciate the gene expression changes specifically in neutrophils between TEPA and control. Therefore, we decided that using the Log2 fold changes, calculated during differential gene expression analysis, would be a better way to visualize some selected significant genes defying N1 and N2 subsets. Indeed, this visualization helped us to highlight the gene expression differences between TEPA and Control in the N2 subset. As validation, TGF-beta is the main driver of N2 polarization which is the most downregulated gene in Fig. 6d and is consistent with our cytokine profiling in Fig. 2b. We have reflected this in-text on Line 338: “This was unsurprising given N2 polarization is chiefly mediated by TGF-β which is significantly reduced in the tumor microenvironment with TEPA treatment (Fig. 2b) (44).”

Concerning the expression of the other genes shown in the N2 signature, since the magnitude of change was small (this is consistent with the low transcriptionally active nature of neutrophils), we couldn't visualize it when compared to the gene expression coming from the other cell types. After subsetting out the neutrophils and changing the metric into log2FC, we were able to better discriminate the significant effects of TEPA in this cluster.

We also considered a recent article in which the authors performed a single-neutrophils RNA-seq for different tumours with the scope of creating a neutrophils atlas (Wu et al., 2024) but we couldn't compare given the heterogeneity obtained with 225 samples and 17 cancer types to our dataset (2 conditions, 1 cancer type). Moreover, Xie et al. identified distinct transcriptional subtypes depending on the developmental stage of neutrophils. Given the enrichment of *Isg15*, *S100a8*, *Mmp8* and *Retnlg* with TEPA treatment, we believe our neutrophils are well-represented by the G4/G5b,c clusters identified by that study. These are associated with more mature neutrophils capable of carrying out migration and pro-inflammatory functions. Given that this study focused on human neutrophils, we could not fully distinguish these clusters in our murine dataset but we hypothesise that they similarly carry out these functions. We have acknowledged this classification in brief in Line 340: "Of note, treatment enriched the expression of the interferon response gene *Isg15*, a marker associated with a mature neutrophil phenotype that can be secreted to promote NK proliferation, DC maturation, and T-cell secretion of IFN- γ (46,47)."

Moreover, to further understand the effect of TEPA, we performed experiments co-culturing human neutrophils with neuroblastoma cells lines (see response to question 12 of reviewer 1). In particular, we incubated the neutrophils with condition media, which was coming from neuroblastoma cells treated for 24h with TEPA. After that we performed real-time PCR in the neutrophils: In Fig. 6h we showed, increased levels of the metallothioneine *MTX1* which correlates with increased copper uptake, increased expression of the migration gene *S100A8* and the activation gene *ISG15*. Additionally, we performed a migration assay which demonstrated increased neutrophils migration towards BE2C cells when treated with TEPA, as shown in Fig 6i. We also performed an ADCC killing assay which showed enhanced killing in the presence of TEPA/anti-GD2 antibody, as shown in Fig 6j. All these results are showing the neutrophils showed a more N1 phenotype and were able to migrate towards and kill cancer cells.

-Is Figure 4f normalized to tumor weight? The figure caption points out that 4f is showing cell proportions; however, it is showing cell counts. What size (area/volume/weight) were the tumors at the time of harvest for scRNAseq?

R: Thanks for this observation. This oversight has been updated to normalize to section weight that was dissociated for flow cytometric sorting. We have also provided individual cell counts in Supp. Fig. 2 for transparency.

-The authors stated in the methods section that treatment with saline or TEPA was started when mouse tumors were 3-4 mm³. This is a tiny tumor, even for neuroblastoma. It would be helpful to also provide tumor growth curves, as well as response data, for Figure 1.

R: We agree with this comment about the tumor size. We have to consider that TEPA is a “gentle” chelator, and at 400mg/kg showed low cytotoxicity as single agent. For this paper we spent a bit of time optimizing the dose and the starting tumor size to find synergy between TEPA and anti-GD2 therapy. In particular, we wanted to use the minimum dose of TEPA (400mg/kg) to see the effect on the tumor microenvironment, which was confirmed by our immune fluorescence data happening after one week of treatment. This is consistent with our previous work (Cancer Res 2020) where we showed that it takes one week of daily doses of TEPA (400mg/kg) to reduce the level of copper in the tumor and to modulate the tumor microenvironment increasing immune cells infiltration. Moreover, during the optimization of the dosing strategy, we initially commenced copper chelation therapy and anti-GD2 therapy simultaneously which conferred no survival advantage over anti-GD2 therapy alone. This emphasises the importance of the 7 days priming stage, and therefore the use of smaller tumors to give the necessary time to TEPA to achieve efficacious copper reduction and modulation of the tumor microenvironment prior to addition of immunotherapy. We have changed the manuscript text slightly to allude to this priming sequence in Line 107: “To stimulate immune activation in the Th-MYCN model, we primed the tumor microenvironment by administering the copper chelating agent TEPA daily for one week, followed by the addition of twice-weekly doses of anti-GD2 antibody (Fig. 1a). After four cycles, TEPA and anti-GD2 therapy were both reduced to twice weekly administrations until the ethical endpoint (tumor volume ≥1000mm³) or a maximum treatment duration of 180 days was met.” In the revised paper, we have updated the figure to provide the individual tumor growth curves as requested (see Fig. 1b).

-NXS2 is a notoriously “easy-to-cure” neuroblastoma model when grown in A/J mice. To my knowledge, the NXS2 model is not considered a model of high-risk neuroblastoma, and patients without high-risk disease often respond favorably to treatment (whereas those with high-risk disease have disease that is more difficult to treat).

R: Thanks for this comment. Initially, we planned to test TEPA in the 9464D model (MYCN-amplified), however we found out that the syngeneic C57BL/6J murine strain has been noted for defunct neutrophil trafficking (PMID: 27779193). This defect was evidenced inadvertently by collaborators using a preclinical model of mesothelioma wherein their AB1-HA/BALB/c model

exhibited superior tumoral neutrophil infiltration compared to their AE17/C57/Bl6 model when treated with TEPA. We added this data from the mesothelioma model to show that TEPA increased neutrophils infiltration in a different tumor type and also to justify our choice of using the NXS2 model (Supp. Fig. 8).

We understand your concern as NXS2 cells do not possess MYCN-amplification and exhibit a moderate tumor mutation burden which are indeed uncharacteristic of high-risk neuroblastoma in the clinic (PMID: 31810498). Nonetheless, the NXS2 model which has been widely recognized as an aggressive model of neuroblastoma and is suitable to assess GD2-directed therapies. When compared to 946D and Th-MYCN models, NXS2 tumors exhibit superior GD2 expression, faster growth kinetics, and similarly sparse immune infiltration (below figure from PMID: 33028899).

[figure redacted]

Other broadly used syngeneic models include Neuro2A and C1300 however neither exhibit MYCN amplification, and the latter is GD2-negative (PMID: 37835389; PMID: 31810498).

While Figure 7a does indeed show that TETA treatment slows tumor growth, it doesn't appear to demonstrate (if observed) tumor clearance. Fig 7a shows 4 mice/group, which is not a lot of power. Have the authors repeated that study, or was that just a study of 1 repeat? Can the authors provide response data for that study (tumor free vs tumor bearing)?

R: We appreciate this comment, and we added some clarifications in the material and methods about the choice of the TEPA concentration in Line 679: “This dose was selected based on prior optimization studies to ensure adequate tumor material was available to study the synergy between copper chelation and immunotherapy.” The experiment shown in Fig. 7a was performed to choose the concentration of TEPA to use in the NXS2 model. We specifically choose that concentration because we could see an effect in slowing tumor growth, but not clearance. This means that we still had room to improve the effect of TEPA to show synergy in combination with anti-GD2 immunotherapy. We note that a p-value was incorrectly used in-text however no statistical testing was performed and this has been removed. During review, we conducted a small experiment to increase sample size from n=4 to n=6-7 (addition of 2 mice to Control; 3 mice to TETA) which yielded no statistical significance at day 11 or day 14 of treatment. Tumor kinetics have been updated to illustrate in Fig. 7a.

Fig 7f shows that mice treated with TETA+anti-GD2 become tumor free around D32, but Fig 7g shows mice dying with that same treatment starting at D24. Could it be that most mice die, and only a handful of mice (<50%) become tumor free? Again, it would be helpful to show individual mouse curves here and response data. If less than 50% of mice become tumor free in this “easy-to-cure” NBL model system, is this worth pursuing in the SIOPEN-BEACON2 trial as the current standard of care therapy for patients with the difficult to treat High-Risk Neuroblastoma have a response rate >60%?

R: Thanks for this comment. We would like to clarify that NXS2 was the most appropriate model to use for the aim of this paper. It is well known that has a good expression of GD2 and it is established in A/J mouse strain which have fully functional myeloid component, whereas the 9464D is established in C57BL/6J murine strain which has a defunct neutrophil trafficking (PMID: 27779193). Moreover, we would like to highlight that the growth of tumor in the NXS2 model is comparable and, in many cases, more aggressive compared to other neuroblastoma models. Please see references and data in our response to your comment 6. We would like also to underline that TEPA + anti-GD2 showed a similar effect in eradicating the tumor in TH-MYCN mouse model (Fig. 1). Our data have shown strong evidence that copper chelation therapy increases the migration and activation of neutrophils which are the important effectors cells to induce ADCC in anti-GD2 therapy. We had shown that by using low dose of copper chelator (similar to what used in maintenance treatment of Wilson’s disease) we improved the efficacy of anti-GD2 in two mouse models of neuroblastoma. We agree that mouse models are different than human, but the ability of copper chelation to make the tumor microenvironment immunologically “hot”, and in particular its activity in increasing migration and activation of neutrophils, makes copper chelation therapy an ideal drug candidate to be tested in clinical trial in combination with anti-GD2 therapy. To improve the transparency of our data, we have provided growth kinetics for individual tumors as requested in Fig. 7f. To address your comment, we sought to investigate the response to TETA + anti-GD2 against the Saline + vehicle

treatment group in the NXS2 model. Plotted below, tumours respond to TETA + anti-GD2 treatment when compared to Saline + vehicle which grow aggressively. We split NXS2 tumor curves of the TETA + anti-GD2 group into responsive (more) (n=3/8) and responsive (less) (n=5/8) arms to understand how tumor control is induced. Prior to commencing TETA priming (d7), both arms started with similar tumour sizes (Rmore: 35mm³ vs Rless: 30mm³). After completion of TETA priming (d14), Rmore were determined to have an average tumor size of 36mm³, compared to Rless with an average of 349mm³. This indicates that tumors which are more responsive to TETA priming indeed benefit from the addition of anti-GD2 therapy to induce durable tumor clearance, most likely attributed to favorable modulation of the immunosuppressive tumor microenvironment. Moreover, we also demonstrate this in our Th-MYCN model utilizing TEPA with values plotted below.

Grey shading to indicate copper chelation therapy (TEPA or TETA) window; dotted lines indicate administration of anti-GD2.

	NXS2 Average tumor size (mm ³)			Th-MYCN Average tumor palpation (mm)		
	Saline + vehicle	TETA + anti-GD2 less	TETA + anti-GD2 more	Saline + vehicle	TEPA + anti-GD2 less	TEPA + anti-GD2 more
Starting size	50	30	35	3.5	3.4	3.6
After copper chelation priming	296	349	36	5.0	4.4	3.8
1wk of copper chelation + anti-GD2	1502	343	22	7.5	6.7	5.3

-In the discussion, on lines 461, the authors state that they reported on macrophages, DCs, granulocytes, etc. exhibit enhanced activation (etc.). Please mention here that this is Sup. Fig. 1, and that this was done in silico as no data in vitro or in vivo supports these findings.

R: Thanks for this suggestion we have added in the discussion that this data were done in silico. Moreover, we performed new experiments and had more data in the revised paper in Fig. 2c, where we showed the flow cytometric analysis of myeloid subset frequencies from Th-MYCN tumors after one week of TEPA treatment. We have reflected this in Line 539: "Herein we report the in silico and in vivo identification of major myeloid-associated cell subsets including monocytes (differentiating into macrophages, dendritic cells) and granulocytes (differentiating into neutrophils, basophils, and eosinophils) which exhibited enhanced activation consistent with anti-tumor immune response with short-term copper chelation therapy."

-In lines 485 to 486 of the Discussion, the authors overstate that neutrophils are the "most" potent effector population of anti-GD2 therapy, here referring to a paper from 1991 and 2000. The paper from 2000, in fact, shows that while neutrophils can effectors with anti-GD2 therapy (at the right E:T ratios), they can also encourage tumor growth. I don't know that either of these papers compared neutrophils to NK cells, but such study would be needed to make that statement that neutrophils are the "most" potent effector as NK cells are incredibly potent effectors of anti-GD2 treatment in vitro. This statement should be softened quite a bit, or even removed.

R: Thanks for this comment. We have amended this in Line 560 to specify the role neutrophil-mediated ADCC in neuroblastoma: "In neuroblastoma, neutrophils are recognized as a key effector of GD2 antibody therapy, mediating the eradication of opsonized cells via Fc gamma receptor IIa (FcγRIIa) binding (65). In vitro mechanistic studies have recently revealed that neutrophil-mediated ADCC occurs predominantly via trogocytosis with cytotoxic activity enhanced by stimulation with GM-CSF (65,66)."

Major Requests:

-The authors make the claim that "mobilization of copper enhances anti-tumor activity of neutrophils"; the evidence included in this manuscript to support this statement include increased neutrophil counts in tumors of TEPA treated mice and GSEA analysis showing "N1 anti-tumor neutrophils" in TEPA treated mice. The addition of in vitro killing assays would bolster the claim that the neutrophils are functionally "anti-tumoral" in addition to exhibiting gene signatures that suggest they are.

R: We fully agree with this comment, and we have performed the in vitro experiments using human neutrophils and human neuroblastoma cell lines (MYCN-amplified) to prove increased neutrophils migration and activation after

treatment with TEPA. Please see response to question 12 from reviewer 1. Please also see the new data added in Figure 6h-j.

-The authors use high-tech approaches to identify neutrophil infiltration with TETA + anti-GD2. Oftentimes, neutrophil infiltration is “assumed” via the presence of CD11b+ myeloid cells. Have the authors verified that these in fact are neutrophils via flow cytometry in either model system? CD11b can be on NK cells too (besides its expression on a number of myeloid populations, including macrophages). In the TH-MYCN model, macrophages have been shown to one of the major immune cell subsets present within tumors that develop in these mice as shown by Costa et al. (which the authors acknowledge may be due to the analytical technique and tumor stage). Besides spontaneously developing NBL in TH-MYCN mice, we have found in syngeneic mice bearing TH-MYCN tumors, 9464D, that both macrophages and NK cells play major roles in the antitumor response. Yet, the authors show in Fig 4 that the major population, even in untreated mice, is neutrophils and that the macrophage population is minimal compared to neutrophils. Especially since this differs from what has been reported previously (Costa et al.), it’s crucial to validate the findings observed in Fig 4 with flow cytometry.

Furthermore, as the authors point out, it’s possible that dependent upon the timing of the treatment, neutrophil-driven responses may seem more apparent than other immune cells (like NKs), but in the long term the neutrophils may not be, in fact, the major antitumor effector. The authors assess some of their neutrophil infiltrate at early time points (D7 for eg.). It would be good to confirm the actual number of neutrophils at this time point by flow, as mentioned above, but also to see if they persist at later time points.

R. We fully agree in the need to have more evidence about the neutrophils increase. To validate our data, we included in the revised papers the flow cytometry data showing the ability of copper chelation to increase neutrophils in both THMYCN and A/J mice (Fig. 2c,d,e, and Fig. 7b, respectively). We did observe a slight increase in macrophages and NK cells, but not of the magnitude of neutrophils. By comparing our IF flow cytometry data and single cells data, we noticed that we needed two weeks of TEPA treatment to see a significant increase in NK cells number. Whereas it was enough one week of TEPA treatment to see a high increase of neutrophils in the tumors. Moreover, we provided additional data showing the increase of neutrophils in response to TEPA in a mesothelioma mouse model, performed by our collaborator (Supp. Fig. 8). Additionally, we would like to highlight that the 9464D is established in C57BL/6J murine strain, which has been noted to be defective for neutrophil trafficking (PMID: 27779193).

-To determine if neutrophils truly are the key mediators of the antitumor response, and not NK cells, it would help to show this by depletion studies during treatment. While neutrophils may be hard to deplete, NK cells are not. It would strengthen the role for neutrophil-mediated response if the authors deplete NK cells in the NXS2/A/J model while giving full treatment. If mice respond, even in the face of NK cell (and possibly also CD8 T cell) depletion, this would strengthen the role that neutrophils may be playing.

R: We fully understand this comments. We have been thinking about doing some depletion experiments. However, after discussing with experts in neutrophils (A/Prof

Tatyana Chtanova) and NK biology (A/Prof Fernando Guimaraes), we decided that those experiments would be not informative for our case. This because, we would need to keep the cells depletion for at least 21 days to see the difference in tumor growth and survival by using TEPA and anti-GD2. Keeping cell depletion for long time (more than one week) is extremely toxic and it would cause a series of off target effects, making difficult the interpretation of the data. We posit there exists a relationship between neutrophils with NK and CD8+ effector cells given the results of our ligand-receptor analysis which demonstrates increased Lgal9 and TNFa interactions in Supp. Fig 6. We have updated in-text to highlight this in Line 307: "Given the increase in tumor-infiltrating neutrophils with TEPA treatment, we examined ligand-receptor expression occurring between this cluster and other subtypes and found this was largely driven by the Galectin-9 (Lgals9) and TNF (Tnf) axes which are known to support CD8+ T and NK cell-mediated cytotoxicity (Supp. Fig. 6) (36,37)." Moreover, we confirm the importance of neutrophils using functional in vitro assays using human neutrophils and human neuroblastoma cells line, as reported in Fig. 6h-i.

-It might be worthwhile to pursue this treatment in a separate model of high-risk disease (such as 9464D syngeneic to C57Bl/6 mice) to confirm efficacy, and justify pursuit of clinical trial testing in SIOPEN-BEACON2. This data is too premature to make that jump.

Please see response to comment 6 from reviewer 1.

We believe that the new data added in the revised paper, which shows increased functional activity of human neutrophils after incubation with TEPA, along with the increased number of neutrophils found in a mesothelioma model in response to copper chelation (fully performed by an independent laboratory), strengthens our hypothesis. This also confirms that C57Bl/6 mouse model cannot be used for studying neutrophils activation/migration as it is defective for neutrophil trafficking (PMID: 27779193). We also removed from the tet the mention about SIOPEN-BEACON2 trial.

Reviewer #3 neuroblastoma, neutrophils, early career researcher (Remarks to the Author):

Reviewer #4 neuroblastoma (Remarks to the Author):

Title: Copper chelation redirects neutrophil function to enhance anti-GD2 antibody therapy in neuroblastoma

Manuscript Type: Research Paper
Manuscript ID: 488552
Author: Jourdin R.C. Rouaen et al.

This MS aims at determining whether the combination of a copper chelator (TEPA or TETA) can potentiate the anti-tumor activity of anti-GD2 antibody (Ab) therapy in neuroblastoma (NB). The hypothesis of a possible improved efficacy of a treatment regimen, combining TEPA with anti-GD2, comes from their previously published results. In those cases, they demonstrated that copper intra-tumoral levels regulate the expression of PD-L1 and copper depletion, by means of a copper chelator, reduces the expression of PD-L1 in NB cells and definitively results in increased infiltration of cytotoxic T lymphocytes and NK cells.

In the present work, AAs claim to demonstrate that the copper chelation enhances anti-GD2 Ab efficacy through increasing neutrophils infiltration in NB. The topic of the study is interesting, also in view of future translational application, since the drug repurposing of TETA that demonstrates to be safe and easily administrable per OS, seems to be rationale e feasible. The study has been performed using highly advanced technology and overall is well performed and executed.

Nevertheless, some major concerns need to be addressed before the MS can be accepted for publication in a so prestigious journal like Nature Communications.

Major points

1) Why did the AAs decide to use different schedules of treatment for the two mouse models (Th-MYCN GEMM vs NXS2) used? In the first case, animal are subjected to a long-lasting treatment until the ethical end-point is reached, while in the second, treatment is interrupted at day 42.

R: Thank you for raising this question. As mentioned in the text, we halted treatment at 42 days in the NXS2 mouse model to evaluate the relapse rate, which is a significant concern for patients receiving anti-GD2 antibody therapy (PMID: 36854071). The NXS2 model allows us to measure tumor size accurately using calipers due to its subcutaneous nature. In contrast, the Th-MYCN model relies on palpation to assess tumor size, which is less precise and doesn't capture subtle changes as effectively. Therefore, the Th-MYCN model required a longer treatment duration until the ethical endpoint to ensure comprehensive evaluation.

2) Why did the AAs decide to use different drugs (TEPA and TETA) for the two mouse models, even if they are analogs?

R: Thank you for your question. Our lab has primarily worked with TEPA due to its lower cost and ready availability. Despite TEPA and TETA being chemically and structurally analogous copper chelators, we included TETA in our in vivo experiments because it has recently been reformulated by Orphalan as Cuprior, for improved stability and holds potential as a repurposed anti-cancer drug. We used the NXS2 model for testing TETA as it allowed for more precise measurement of tumor growth with calipers.

3) Figure 1 d and e lacks of an experimental control. The evaluation of immune cells infiltration in tumor mass of mice treated with TEPA + IgG needs to be included.

R: We fully agree with the reviewer about the lack of the control. In the revised manuscript we updated the panel in Fig 1 to provide the TEPA + IgG OPAL results.

Moreover, we are surprised of the total ineffectiveness of TEPA treatment in the Th-MYCN GEMM model, compared to the NXS2 one (Fig 7). Since TEPA determines a tumor microenvironment shift from immune suppressive to immune permissive, and due to the fact that treatment lasts until the ethical end-point, we would expect to see at least a slight effectiveness. How can the AAs explain this difference between the models used?

R: To explore potential synergy between TEPA and anti-GD2 therapy, we optimized drug concentrations and conditions to ensure minimal single-agent activity. In our 2020 Cancer Research paper, we initiated TEPA treatment when the tumor size was very small (≤ 2 mm), which resulted in increased survival in mice treated with TEPA at the same dose. However, in this study, we started treatment when the tumor size was ≥ 3 mm (as shown in the newly added Fig. 1B). This larger starting tumor size made TEPA (at this concentration) less effective in reducing tumor growth, even though it induced increased immune infiltration. We have included this detail in the Materials and Methods section of the revised paper on Line 671. For the NXS2 experiments, we used the same drug concentration as in the Th-MYCN model to facilitate comparison, though we did not have prior data on TETA's effect relative to the starting tumor size in NXS2. Fortunately, this choice proved effective as we observed improved efficacy of the TETA/anti-GD2 combination compared to single treatments. We also noted a slight effect of TETA alone in the NXS2 model, which may be due to the smaller tumor size at the start of treatment and/or the higher immunogenicity of this model.

4) Most of the evidences brought by the AAs are based on gene-set and transcriptomics analyses, which confirm the hypothesis of the AAs. However, in our opinion, a mechanistic demonstration of the real role of the infiltrated neutrophils is necessary. The AAs advise that recruited and activated neutrophils are responsible for the improved efficacy of the combination therapy, but a real demonstration is lacking.

R: We appreciated your suggestion and performed additional experiments using human cells to understand the state of neutrophils. To examine the influence of copper chelation TEPA on the function capacity of neutrophils, we performed characterization studies using human neutrophils and human neuroblastoma cell lines:

a) Using neutrophils from donors, we performed qRT-PCR analysis for the expression of genes in human neutrophils associated with intracellular copper

(MT1X), migration (S100A8) and activation (ISG15) obtained after 30min incubation in conditioned media (media from neuroblastoma cells incubated with TEPA for 24h). Data in Fig. 6h showed, increased levels of the metallothioneine MTX1 which correlates with increased copper uptake, increased expression of the migration gene S100A8 and the increased inflammatory activity indicated by ISG15.

b) We performed a migration assay which demonstrated increased migration towards BE2C cells when treated with TEPA, as shown in Fig. 6i.

c) We performed an ADCC killing assay which showed enhanced killing in the presence of TEPA/anti-GD2 antibody, as shown in Fig. 6j.

5) Further, did the AAs think about the possibility that an ADCP mechanism can be also involved? Please, address this point.

R: We really appreciated this question. As part of our functional assays, we conducted a killing assay and demonstrate enhanced cytotoxicity of neuroblastoma cells by healthy human neutrophils in the presence of anti-GD2 antibody. It has been reported that this predominantly occurs by trogoptosis (phagocytosis-inducing apoptosis). We are currently investigating the expedience of this mechanism in our own work as a separate project but allude to this in the discussion in Line 562: "In vitro mechanistic studies have recently revealed that neutrophil-mediated ADCC occurs predominantly via trogocytosis with cytotoxic activity enhanced by stimulation with GM-CSF (65,66)."

6) We also think that the cytofluorimetric analyses on peripheral blood, bone marrow and tumor masses collected from control and treated mice are mandatory and they need to be showed, to evaluate immune cell subpopulations, with particular attention for myeloid derived suppressor cell markers, which in part overlap with neutrophil and macrophages ones.

R: Thanks for this suggestion. We have added more data in the revised paper in Fig 2c, where we showed the flow cytometric analysis of myeloid subset frequencies from Th-MYCN tumors after one week of TEPA treatment. We have reflected this in Line 539: We report the in silico and in vivo identification of major myeloid-associated cell subsets including monocytes (differentiating into macrophages, dendritic cells) and granulocytes (differentiating into neutrophils, basophils, and eosinophils) which exhibited enhanced activation consistent with anti-tumor immune response with short-term copper chelation therapy.

Minor points

1) Line 93: this sentence looks redundant, since it has been already reported in the introduction section.

This sentence has been replaced with "To stimulate immune activation in Th-MYCN model, we primed the tumor microenvironment by administering the copper chelating agent TEPA daily for one week, followed by the addition of twice-weekly doses of anti-GD2 antibody (Fig. 1a)."

2) Line 361: the sentence “Similarly to TEPA, TETA slowed tumor growth.....” is not correct because, unless some AAs mistakes reported in Figure 1, TEPA is completely ineffective in the transgenic model. *We have removed “Similar to TEPA” from the sentence and updated this to: “Using the syngeneic NXS2 model, we commenced treatment on day seven post-inoculation (Fig. 7a, black arrow) and animals were randomly assigned to receive control (saline) or TETA daily for seven days with treatment observed to slow tumor growth as a single agent (Fig. 7a).”*

In addition to conducting the experiments suggested by the reviewers, we have added more data to enhance clarity and strengthen our hypothesis.

- 1. To gain further insight into the changes occurring in cytokine production, we included the analysis of their serum levels in mice to match the measurements taken from the neuroblastoma TME in the Th-MYCN model (Fig 2b). This approach provides deeper understanding of the relationship between local and systemic immune responses, and emphasizes the role of IL-8/KC in stimulating neutrophil mobilization and trafficking to the tumor site.*
- 2. To further support our copper distribution hypothesis, we plated neuroblastoma cell line SK-N-BE(2)-C and monitored growth and assayed copper media concentration over a 48hr period. We observed normal expansion of cells with a simultaneous decrease in media copper, demonstrating that tumor cells continue to uptake environmental copper as part of their growth. We reflect this work in Line 370: “Expanding on this observation, SK-N-BE(2)-C cells monitored over 48hr continued to sequester copper from media to support their growth (Supp. Fig. 7a,b).”*

- 3. For further clinical validation, we performed a Kaplan-Meier analysis of patients with neuroblastoma using the copper exporter ATP7A gene in the R2: Genomics Analysis and Visualization Platform where elevated ATP7A expression was associated with improved survival. This is updated in Line 377: “In clinical support, elevated expression of the copper export protein ATP7A (encoded by ATP7A) was significantly associated with a T cell-infiltrated tumor microenvironment in a wide range of solid human pediatric malignancies, and was also associated with the improved survival of patients with neuroblastoma (Supp. Fig. 7c,d).”*

REVIEWERS' COMMENTS

Reviewer #1 (Remarks to the Author):

Reviewer #1 anti-GD2 in cancer (Remarks to the Author):

Overall, the authors have addressed the majority of my concerns with the original manuscript with additional experiments and explanations. Please see below my comments to my revision requests (authors' detailed answers are taken out to abbreviate this review) with a few remaining points noted by [*] that still need to be addressed:

1. Figure 1D, 2C – Images are somewhat blurry in my version; consider improving quality of Multiplex OPAL IHC images.

- Addressed

2. Figure 1D, E - It would be nice to see the TEPA + IgG control for these studies. While these were separately studied in Figure 2 in a 7-day treatment experiment without the IgG, it is important to see them in context with the other treatments in the same experiment.

- The IgG control was appropriately added. * OF NOTE: Figure 1D contains TETA instead of TEPA in labeling – is this correct? If so, why (as the rest of the figure is TEPA)?

Thank you for noting this. Label text was copied over from Figure 7 in error and has been corrected.

3. FIGURE 1B, the authors should comment here or in the discussion why they do not see any anti-tumor/survival effect of the TEPA + IgG control, given their Cancer Research article shows a statistically significant improvement in survival in vivo comparing control to TEPA alone in same mouse model. Do they think the control IgG had some negative effect on TEPA treatment?

- Addressed. While it seems like a very small difference in tumor size, I can appreciate that this small size can make a large impact in the outcome of an in vivo study.

4. Figure 1E- In the authors' previous Cancer Research article, TEPA treatment significantly increased CD8 T cell tumor infiltration but this is not recapitulated in the current model. Is there a reason for this inconsistency? Difference here is the inclusion of anti-GD2 antibody, which has a strong mono-agent effect on CD8 infiltration. See further comment below.

- Comment thoroughly addressed.

5. In the text (lines 140-141), the authors state that levels of TGF- β are reduced in TEPA-treated tumors. However, the stats in Figure 2B do not support that this absolutely occurs and the reduction is marginal compared to the increases observed

in other cytokines. It might be better to exclude this from the results text or modify to sound less conclusive.

- Comment addressed by additional experimentation. * OF NOTE: the treatment schemas presented in Figure 2A of original manuscript and Figure 2A of revised manuscript differ by 1 additional day of treatment (mouse schema at top of figures) so not sure that it was appropriate to combine the results for statistical purposes if indeed the treatment durations differed. Please clarify if this was error in the cartoons and if all animals were treated with same duration of treatment regimen or justify why it is ok to combine?

The original mouse schema submitted was incorrect and was corrected during address of reviewer's comments. We apologize for not noting this in our changes.

6. In Figure 2C, TEPA treatment now significantly increases CD8 T cell tumor infiltration but this was not the case in Figure 1D,E. This result better reflects the authors' previous work. One explanation is that the anti-GD2 antibody caused a dramatic increase on its own that cannot be furthered by TEPA? The authors should draw this dissimilarity out in the text regarding Figure 2C- D. Other than this, panels 2C and 2D are somewhat redundant with 1D and 1E (and previously published work). It might be advantageous to just show the TEPA only controls in Figure 1. Based on the arms of the in vivo study presented, a TEPA and IgG treatment arm was included. Are samples available for IF?

- Comment thoroughly addressed with additional data to support conclusions. redundant figures removed.

7. Figure 3A, Smooth Muscle Actin appears greatly enhanced in TEPA-treated tumors compared to control. The authors do not touch on this marker in the text related to the figure. Is this change expected in the model (does it represent vasculature?) and how does it relate to the other findings in that figure?

- * Partially addressed. I still think it would be useful to the manuscript reader to address the changes in SMA in the main text. It is still not entirely clear what SMA is meant to represent in the TMA and why it was recommended by nano string as part of the standard GeoMx panel.

As this did not form one of our original hypotheses, we do not have sufficient data and would prefer not to draw conclusions with proper and robust evidence and inferences. We will seek to investigate this aspect more thoroughly in our future studies.

8. Figure 3B, why are days 3 & 7 combined for control tumors but separated for treated tumors? Should the reader assume tumors were identical at these points or was data combined for control purposes? Also, in Figure 3B, the True Infiltration signal seems robust even in a control tumor but in Figure 3A, almost no CD45 staining is observed – is that due to photo quality and/or magnification or sampling differences? Can the authors provide a subfigure with increased magnification in an area of infiltration for both control and treated tumors so that the results for Figure 3A/B are better aligned?

- Addressed.

9. Figure 3C. The murine hemoglobin genes are the most highly upregulated in the + infiltrated tumor sections, along with smooth muscle actin (which is in keeping with figure 3A IF). Is there increased vascularity in those TEPA treated “infiltrated” cores that may help explain the increased WBC (as well as RBCs/Hb genes) in those sites? Was bleeding into the tissue noted by IHC at any of those “infiltrated” sites (to suggest necrosis or breakdown of the vascularity where WBC more easily “seep” into the tissues following TEPA treatment vs recruitment?). Or does smooth muscle actin represent something else?

- Addressed. See above regarding SMA.

10. Figure 4E, would these findings not refute the IF data from Figures 2 (and prior Ca Res work) where TEPA treatment increased tumor recruitment of CD8 T cells and NK cells? Do the authors have any explanation for why TEPA did not alter CD8/NK cell recruitment in sequencing data of tumors? It is also interesting that TEPA treatment results in a reduced CD4 T cell compartment (Figure 4E, F) but this is not addressed by authors. Please also compare Figure 4 results to Figure 1 TEPA + IgG alone arm (once you add it) and TEPA/Anti-GD2 results (which, the combination, is similar to antiGD2 alone in increasing CD8/NK recruitment).

- Addressed. Authors acknowledged discrepancies in findings and presented logic for differences with experiments to address. Adjusted conclusion to focus on neutrophils due to caveats in experiments.

11. Figure 6D, not sure of model-specific levels but this percentage seems low for circulating neutrophils. Can the authors provide details on the blood processing and/or show percentages of other immune cells as a supplemental component?

- Addressed.

12. Could the authors use the human derived NB in vitro models in Figure 6 to provide any human validation of the murine modeling? Some considerations... Does TEPA treatment enhance migration of human neutrophils towards neuroblastoma cells using a migration assay? Do human cells mimic the cytokine secretion observed in Figure 2B following TEPA treatment?

* Would be ideal to have an additional biological (and technical replicates) here but the authors explicitly state the low n's in the legend. Maybe move to supplemental if not going to perform a more robust characterization.

Human neutrophils are difficult to isolate, and maintain given their short half-life. Moreover, they possess an incredibly low mRNA count per cell making which necessitate a high starting number of cells (approx. 10×10^6) required for qPCR analyses. We thank you for your suggestion and have exhausted our neutrophil stocks for performing these characterization experiments.

b) We performed a migration assay which demonstrated increased migration

towards BE2C cells when treated with TEPA, as shown in Fig. 6i.

* Y-axis of figure has a typo – should be transmigration.

We thank you for noting this error which has now been corrected to “transmigration”.

c) We performed an ADCC killing assay which showed enhanced killing in the presence of TEPA/anti-GD2 antibody against GD2+/MYCN-amplified Kelly cells, as shown in Fig. 6j.

- Comment was addressed using human derived in vitro models with similar outcomes.

13. In lines 299-323, the authors allude that neutrophil uptake of copper following neuroblastoma TEPA treatment should enhance neutrophil functionality. To support this, do human neutrophils demonstrate a more N1-like phenotype and/or ADCC potential following co-culture with TEPA treated human neuroblastoma cells?

- Addressed. * On the experiment design, can the authors confirm that the increased caspase-3/7 signal was generated by dying tumor cells and not dying neutrophils. The results fit within the story as a whole, which is great, but was the experiment controlled by assaying each of the components separately since caspase-3/7 dye is not specific to tumors or neutrophils.

We confirm that these neutrophil only wells were subtracted from the matched neutrophil + SK-N-BE2-C wells. To clarify this, we have added in Line 404: Importantly, caspase activity was not detected in treated neutrophils in the absence of the neuroblastoma cells, confirming assay specificity and demonstrating the therapeutic safety of this combination strategy.

14. The overall meaning of the sentence in lines 321-323 is not clear to me. Also, could the authors speak more to the secreted soluble factors that are mentioned?

- Addressed with revised wording

15. For Figure 7B, should the Y-axis label be CD45+ leukocytes? I don't think the authors are looking for CD11b/Ly6G in the lymphocyte population as neutrophils are of myeloid lineage.

- Addressed

16. Visually the infiltration of NK/CD8/myeloid cells in 1D, 2C and 7i does not appear drastically different (albeit images are blurry) but the axes are drastically different in scale for subsequent quantification. The models are different so this cannot be compared head-to-head but the quantitation seems inconsistent between figures.

- Addressed.

17. In discussion, the authors should consider including clinical literature for trientine

in pediatric patients as studies are available and document potential adverse events. It is likely that toxicity can be controlled through proper dosing. (Mayr et al J Pediat Gastroenterol Nutr 2021)

- Addressed.

18. Neutrophils are the predominant white blood cell subset in humans (~70% of circulating leukocytes) but are less common in mice (~30%). There are also several known phenotypic and functional differences between human and murine neutrophils (Reviewed by Eruslanov et al, Trends in Cancer 2017, i.e., human granulocytes do not express Ly-6G, differences in FC receptor expression and granule content). This should be addressed and/or highlighted as a potential limitation of the study. In addition to studies proposed above, do the authors have any more results that highlight their findings are recapitulated in human tissues?

- Addressed with revisions.

** Minor additional comments on revised figures and additional unsolicited data provided by the authors **:

- Revised Figure 2C: * You state that the myeloid cells (monocytes/granulocytes) show "enhanced activation". How did you determine in your flow analysis that macrophages and granulocytes exhibited "enhanced activation"? What markers did you look at by flow in these cells to draw that conclusion?

In Line 580, we wrote "Herein we report the *in silico* and *in vivo* identification of major myeloid-associated cell subsets" to encompass both types of analyses. We refer to enhanced activation observed in Fig. 5a and Line 319: "myeloid migration, differentiation, and activation". We did not perform flow cytometric analyses of the myeloid compartment as we were primarily interested in neutrophils.

- The authors wrote: "To gain further insight into the changes occurring in cytokine production, we included the analysis of their serum levels in mice to match the measurements taken from the neuroblastoma TME in the Th-MYCN model (Fig 2b). This approach provides deeper understanding of the relationship between local and systemic immune responses, and emphasizes the role of IL-8/KC in stimulating neutrophil mobilization and trafficking to the tumor site."

* Do you mean IL-6 (IL-8 is not shown in figure 2b, but IL-6 is elevated in TME of TEPA treated animals)? KC is more commonly known as CXCL1 and not IL-8 (IL-8 is also known as CXCL8, which is also not depicted in Fig 2b, so did you mean KC/CXCL1? Or KC/IL-6?). By identifying increased KC/CXCL1 and/or IL-6 in the serum/TME does not guarantee that they stimulated or aided in neutrophil mobilization and trafficking to the tumor *in vivo* for you did not show this to be the case functionally (by blocking it, etc). I would reword that statement to be a less definitive relationship unless you can perform functional experiments to support that cause/effect.

Thank you for bringing this to our attention. Although KC/CXCL1 has been recognised as murine homolog of human IL-8 (PMID: 20007247; PMID: 16020260),

a consult with more recent literature cites that a true rodent homolog for CXCL8 has not been found (ref). We have reflected this and updated Line 167 from "KC/CXCL1/IL-8" to KC/CXCL1" to avoid confusion. Although included in our response to reviewer's, the statement above does not appear in the actual manuscript, however, we allude to this role in Line 178: "Given the systemic and local upregulation of KC (a known neutrophil chemoattractant) with copper chelation therapy, we postulated that treatment could stimulate neutrophil trafficking towards the tumor site (19)."

•The authors write: "For further clinical validation, we performed a Kaplan-Meier analysis of patients with neuroblastoma using the copper exporter ATP7A gene in the R2: Genomics Analysis and Visualization Platform where elevated ATP7A expression was associated with improved survival. This is updated in Line 377: "In clinical support, elevated expression of the copper export protein ATP7A (encoded by ATP7A) was significantly associated with a T cell-infiltrated tumor microenvironment in a wide range of solid human pediatric malignancies, and was also associated with the improved survival of patients with neuroblastoma (Supp. Fig. 7c,d)."

* Is the Kaplan Meier curve of ATP7A expression and survival from gene expression/outcomes data of high-risk neuroblastoma tumors only? This graph is less relevant if it is taking in ALL risk categories of neuroblastoma tumors - as only HR tumors are immune cold and treated with anti-GD2. What gene expression dataset in R2 did you use? and did you validate it with another tumor dataset that has outcomes data (there are many in that database)? Did you use the bonferoni correction of the P value from that analysis of multiple patient data given the regular p value is not accurate in this setting?

We sought to include all risk groups of neuroblastomas as the copper transfer phenomenon is likely present and as such contribute to immune effector activity. For example, the 4S group experience spontaneous tumor regression with immune infiltration thought to play a role in tumour clearance alongside neuronal differentiation (PMID: 37709978). It is therefore likely prognostic for all NB risk groups to have high ATP7A expression (as we also show in Supp. Fig. 6c for many types of solid pediatric malignancies) to facilitate this copper transfer and we argue that copper chelation therapy expedites copper redistribution. The Bonferroni correction is only used in multiple comparison testing (i.e. three or more groups) in a Kaplan-Meier survival analysis (PMID: 29768911). Given we are only comparing two groups (high, low), we can proceed with the noted raw p-value.

Reviewer #2 (Remarks to the Author):

Thank you for addressing many of the questions that were raised. I understand the logic for using the NXS2 tumor model from the consideration of the role of neutrophil infiltration.

While the authors have shown other articles which show the expression of GD2 on NXS2 cells, the expression of GD2 on NXS2 cells notoriously varies, and it is often lost over time. For that reason, it would be helpful for the authors to show the expression level (perhaps in a supplementary figure) of GD2 on their NXS2 cells to confirm the expression is good.

We confirm we have positive GD2 expression in our NXS2 cells and provide flow cytometric staining as part of our supplementary materials. We have updated to include this at Line 479: "To validate model sensitivity, we confirmed NXS2 as GD2+ cells by flow cytometry prior to inoculation (Supp. Fig. 8d)."

Thank you for including the individual mouse curves for Figure 7. Would it also be possible (either in the text after the treatment name, or in a separate sub-figure graph) to show the cure rate for Teta+anti-GD2 in that same figure. For example, after the in the text after the treatment, you could maybe include "Teta+anti-GD2 (3/8 TF); Teta+anti-IgG (0/8); etc."

Thank you for your constructive advice and are in agreement as we note in-text this schedule was designed to evaluate relapse rates after treatment cessation at 42 days. We have provided the cure rates for each group in Fig. 7e.

Reviewer #3 (Remarks to the Author):

I believe the edits the authors have made address the concerns I conveyed during initial review of this manuscript.

Reviewer #4 (Remarks to the Author):

Title: Copper chelation redirects neutrophil function to enhance anti-GD2 antibody therapy in neuroblastoma

Manuscript Type: Research Paper

Manuscript ID: 488552_1

Author: Jourdin R.C. Rouaen et al.

The AAs addressed almost all points raised during the first revision, and the manuscript can be now considered acceptable for publication.

Minor

In Figure 1, as requested, AAs added experimental controls. In Figure 1d, TETA should be however substituted with TEPA.

Thank you for noting this. Label text was copied over from Figure 7 in error and has been corrected.